# *AQuA*: A Benchmarking Tool for Label Quality Assessment

**Mononito Goswami**[*], **Vedant Sanil**[*], **Arjun Choudhry**[†],
**Arvind Srinivasan**[†], **Chalisa Udompanyawit**, **Artur Dubrawski**
Auton Lab, School of Computer Science
Carnegie Mellon University
{mgoswami, vsanil, arjuncho, arvindsr, cudompan, awd}@cs.cmu.edu
www.github.com/autonlab/aqua

## Abstract

Machine learning (ML) models are only as good as the data they are trained on. But recent studies have found datasets widely used to train and evaluate ML models, e.g. *ImageNet*, to have pervasive labeling errors. Erroneous labels on the train set hurt ML models' ability to generalize, and they impact evaluation and model selection using the test set. Consequently, learning in the presence of labeling errors is an active area of research, yet this field lacks a comprehensive benchmark to evaluate these methods. Most of these methods are evaluated on a few computer vision datasets with significant variance in the experimental protocols. With such a large pool of methods and inconsistent evaluation, it is also unclear how ML practitioners can choose the right models to assess label quality in their data. To this end, we propose a benchmarking environment AQuA to rigorously evaluate methods that enable machine learning in the presence of label noise. We also introduce a design space to delineate concrete design choices of label error detection models. We hope that our proposed design space and benchmark enable practitioners to choose the right tools to improve their label quality and that our benchmark enables objective and rigorous evaluation of machine learning tools facing mislabeled data.

## 1   Introduction

A lot of machine learning (ML) research is devoted to making efficient and effective use of available data to learn accurate, high-fidelity, and interpretable models, with little to no focus on the quality of the data they are trained and evaluated on. Nonetheless, it is widely recognized that ML models are only as good as the data they rely on, i.e., the quality of data imposes practical limits to what ML models can achieve. Not only are datasets used to train ML models; they also serve as benchmarks to measure the state-of-the-art and validate theoretical findings. Thus, high quality large labeled datasets are the cornerstone of progress in supervised machine learning. However, the data is rarely free of noise, which can both manifest in the features of the data (feature noise) and in labels that categorize them (label noise). Between feature and label noise, the former has been found to be much more harmful to machine learning models [1, 2, 3]. To make matters worse, label noise is prevalent in popular ML benchmarks. A recent study estimated an average of at least 3.3% label errors across 10 datasets commonly used for benchmarking computer vision, natural language, and audio classification algorithms [4]. Consequently, a growing body of research is devoted to understanding the harms of label noise and to developing techniques to identify and mitigate labeling errors.

---

[*]MG and VS contributed equally. MG is the corresponding author.
[†]AC and AS have equal contribution.

37th Conference on Neural Information Processing Systems (NeurIPS 2023) Track on Datasets and Benchmarks.

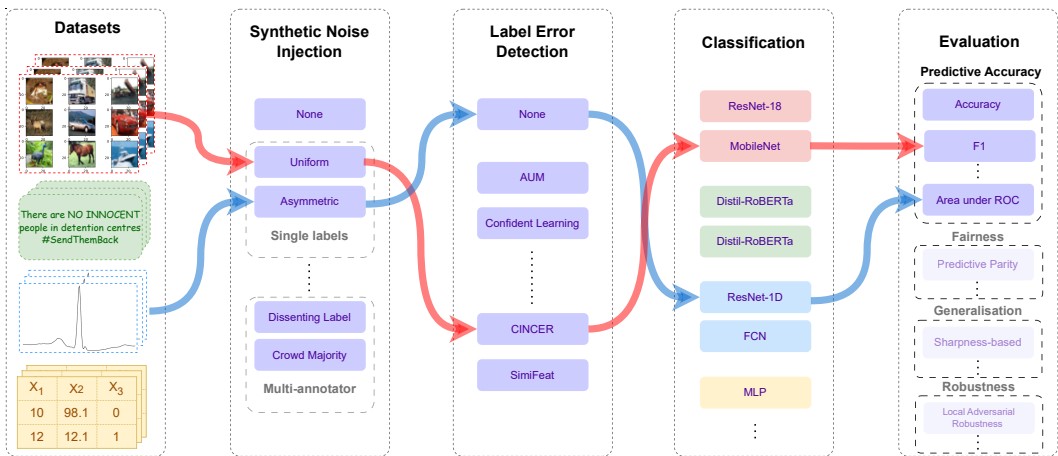

Figure 1: *Overview of the* `AQuA` *benchmark framework.* `AQuA` comprises of datasets from **4** modalities, **4** single-label and **3** multi-annotator label noise injection methods, **4** state-of-the-art label error detection models, classification models, and several evaluation metrics beyond metrics of predictive accuracy. We are in the process of integrating several fairness, generalization, and robustness metrics into `AQuA`. The red and blue arrows show two example experimental pipelines for image data and time-series data, respectively.

In recent years, over **50** papers have been written on this topic, including **6** surveys, yet the literature lacks a comprehensive benchmark to evaluate the available methods. The evaluation of existing methods is lacking along the following dimensions:

**Arbitrary choice of datasets and limited data modalities.** To the best of our knowledge, relevant studies have used over **40** datasets (e.g., ImageNet [5]) and their variations (e.g., Imagenette [6], ImageNet-100 [7]) for evaluation, but mostly on computer vision related tasks, with less than **15** studies using text data, **7** using tabular data and only **1** paper using time-series data.

**Arbitrary choice of classification models.** The ultimate goal of identifying labeling errors is to learn a classification model using training data with clean labels. Much like the datasets, relevant studies have used over **47** different classification architectures (e.g., ResNet [8], MobileNet [9], ResNeXt [10], BERT [11], XLM-RoBERTa [12], etc) to measure the impact of label cleaning.

**Inconsistent evaluation protocols and metrics.** Different studies conduct different experiments to measure the efficacy of their proposed methods (e.g., the accuracy of the label cleaning method, or performance of the downstream model before and after label cleaning, etc.) and use various measures of success (e.g., high accuracy, $F_1$-score, or low error rate).

With such diversity and inconsistency in the way in which these methods are evaluated, it is hard to measure the state of the art. To bridge this gap, we propose the Annotation Quality Assessment, `AQuA`, the *first* benchmark framework to evaluate machine learning methods in the presence of label noise (Fig. 1). We also elucidate the design space for such models, with the hope that it will not only foster future research on detecting labeling errors, but also enable ML practitioners to choose the appropriate label cleaning tools for their specific data and tasks. We run a large-scale experiment (**> 1000** unique experiments) and make several interesting observations, demonstrating `AQuA`'s efficacy in benchmarking machine learning models in the presence of label noise.

## 2 Background and Problem Formulation

**Sources of labeling errors.** Labeling errors can arise from automated labeling processes such as crowd-sourcing [13], programmatic weak supervision [14, 15], and human error (e.g., due to lack of expertise or low confidence in expert assessment) [16][3]. Errors may also stem from idiosyncrasies

---

[3]The root cause of labeling errors in crowd-sourcing is different from human expert annotation. For instance, errors during crowd-sourcing have been shown to arise from other factors such as gaming the system to maximize monetary gains [13].

of the annotation procedure and the corresponding guidelines themselves [17]. Finally, existing labels may also become inconsistent with prevailing knowledge due to constantly evolving problem definitions and domain knowledge leading to concept drift[4].

**Impact of labeling errors.** At training time, labeling errors can cripple an ML model's ability to generalize and introduce undesirable biases in its hypothesis space [19, 20]. Mislabeled training data is especially problematic for over-parameterized deep neural networks, which can achieve zero training error even on randomly-assigned labels [20]. At test time, labeling errors can lead to noisy model evaluations and invalidate common model selection strategies. In safety-critical settings, models trained, evaluated, and selected using mislabeled data can be

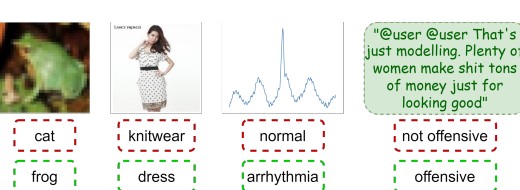

Figure 2: Labeling errors in widely used benchmarks: CIFAR-10, Clothing-100K, MIT-BIH, and TweetEval Hate Speech datasets. Observed labels are in red and true labels are in green.

ineffective at best and can lead to disastrous outcomes at worst. Finally, recent studies in the context of fairness have shown that naively enforcing parity constraints based on noisy labels can harm groups that are unaffected by label noise [21, 22].

**Problem formulation.** Due to the far-reaching consequences that labeling errors can have on model training and evaluation, the literature has attacked multiple different but related problems, for example: (1) *label error detection*, identify which data points have erroneous labels [23, 24], (2) *label noise estimation*, estimate the proportion of data with noisy labels [25], (3) *label noise robust learning*, learn models robust to label noise [26, 27], and (4) *noise transition matrix estimation*, estimate the parameters of the noisy label generation process [28].

In this work, we focus on the **label error detection problem**, because (a) it is the most *general* of the above problem types, i.e., with knowledge of labeling errors, we can estimate the noise rate, parameters of the noise generation process and train ML models free from label noise, (b) it provides practitioners greater visibility of issues that plague their data, and (c) allows them to directly rectify these errors.

> **Label error detection problem:** Assume a dataset $\mathcal{D}^* = \{(\mathbf{x}_i, y_i^*)\}_{i=1}^N \in (\mathcal{X}, \mathcal{Y})$, where $\mathbf{x}_i$ and $y_i^*$ denote the features and labels, respectively. In practice, we do not have access to $\mathcal{D}^*$, but instead observe a noisy dataset $\mathcal{D} = \{(\mathbf{x}_i, y_i)\}_{i=1}^N \in (\mathcal{X}, \mathcal{Y})$[a]. We call $y_i$ a *labeling error*[b] if $y_i \neq y_i^*$, and *correctly labeled*, otherwise. Our goal is to identify all labeling errors in $\mathcal{D}$.
>
> ___________
> [a]We assume that we observe the true features since we are interested in identifying labeling errors and isolating their impact on downstream model performance.
> [b]*A note on terminology:* In this paper, we will sometimes refer to *labeling errors* as *noisy labels* or *label noise*, and the process of identifying them as *label error detection*, or loosely as *label cleaning*.

## 3 A Design Space of Labeling Error Detection Models

In this section, we seek to align the dimensions along which label error detection models vary, with dimensions that can facilitate model selection for ML practitioners. We provide a brief overview of these dimensions below and defer detailed discussions to Appendix A.1.

**What inputs do you have?** All label error detection models take features and noisy labels as input. In most datasets, data points are labeled by multiple experts, but their individual annotations are seldom available. When available, *multi-annotator labels* can be used to identify data points that are inherently ambiguous [29], or to model individual annotators to estimate their expertise and propensity for mislabeling examples [7], and using these to identify likely labeling errors. While most methods identify labeling errors and automatically remove or correct them, a few rely on a *human*

___________
[4]For example, sepsis is one of the most sought-after clinical conditions to predict. However, with the constantly evolving definition of sepsis, the labeling process is frequently affected, causing many annotations in legacy benchmark data to become inconsistent with the latest guidelines [18], a very dangerous risk to take in the particular type of application area

*expert* who can be queried to relabel suspicious data points [29, 30]. Some other methods assume access to data points called *anchor points*, which most certainly belong to a particular class [31, 32]. The number of anchor points required is generally proportional to the number of classes, and quickly becomes prohibitive for multi-class classification problems, and in more complicated noise settings [33]. Finally, a vast majority of methods assume *access to classification models*, and primarily differ in their *number* (model-free [34], one or multiple models [35, 36, 37, 38]), *nature of access* (prediction-only [23] versus access to logits [24], gradients [30] etc.), and *extent of pre-training* (no pre-training [23, 24] versus large-scale pre-training e.g. large language models [39]).

**What modeling assumptions can you make?**
Different studies use different assumptions on *data* (noise structure and clusterability), *heuristics* (model self-confidence and perceptual uncertainty), and *modeling decisions* (whether to explicitly model the transition matrix and multi-network training). Most studies in the literature explicitly assume some form of structure in the noise present in the data [23, 34, 40, 28]. Most early studies assumed class-dependent noise, i.e., the likelihood of error is only dependent on the latent true class, not on the data [23, 28]. There is growing interest in more realistic forms of noise where the probability of error also depends on the features of a data point (instance-dependent noise) [41, 42]. To this end, some recent studies have shown promising results by leveraging natural notions of similarity between data points and their labels. For example, Zhu et al. [34] assume that examples with similar features should have similar labels.

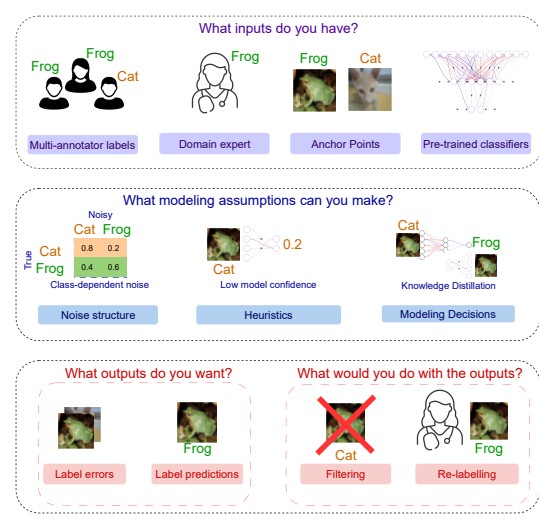

Figure 3: *Design space of labeling error detection models to delineate concrete design choices.*

Many studies treat a trained model's low confidence that a data point belongs to its observed label as a heuristic likelihood to identify labeling errors [24, 23, 43]. In a similar vein, a recent study used the loss of a pre-trained large language model on each data point to identify mislabeled examples [39]. When multi-annotator labels are available, as discussed before, some studies have also used them to model the perceptual uncertainty in the annotators to identify labeling errors.

Finally, studies differ in their modeling decisions. While some explicitly estimate a data structure called the noise transition matrix, which encodes the joint probability of latent true and observed noisy labels [23, 33, 27], others do not [24, 30, 14]. Finally, there is a body of work on label noise robust learning using multiple model instances either using knowledge distillation [35, 36, 44] or meta-learning [38, 37]. The key idea is to use a cooperative game between models to identify labeling errors and ensure that the eventually deployed model only learns from clean data.

**What outputs do you want, and what would you do with them?** All labeling error detection models identify data points that are likely to be labeling errors. With knowledge of the potentially mislabeled data points, most studies simply remove them from consideration[23, 24, 45, 42]. This strategy may be practical for large datasets, where only a small fraction of data is found to be mislabeled and domain experts are unavailable for supervision. We use this strategy by default in AQuA. A smaller number of methods predict the alternate class that the data point is most likely to belong to [38, 34] and even provide explanations for their predictions [30, 46]. CINCER [30] is one of the few methods which not only finds labeling errors but also identifies counter-examples in the training data to serve as explanations for its suspicion. Some studies use the label predicted by these models and perform loss re-weighting or correction to learn robust classification models [27, 47, 48]. When domain experts are available, some studies also leverage their insight to re-label mislabeled data point [30, 29].

| Modality | Dataset | # Train / Test | # Annotators/sample | Label Source | Classification Task | Sample Size | Usage |
|---|---|---|---|---|---|---|---|
| Image | CIFAR-10N[49] | 50K / 10K | 3 | Human annotation | Object | $32 \times 32 \times 3$ | [50, 34] |
| | CIFAR-10H[16] | 0 / 10K | 47–63 | Human annotation | Object | $32 \times 32 \times 3$ | [29] |
| | Clothing100K[51] | 100K | 1 | Web-labeled | Image | $256 \times 256 \times 3$ | [24, 4, 34] |
| | NoisyCXR[52] | 26K / 3K | 1–XX | Human expert annotation | Pneumonia | $1024 \times 1024 \times 1$ | [29] |
| Text | IMDb$^\beta$[53] | 25K / 25K | 1 | Human annotation | Sentiment | - | [27, 47, 4] |
| | TweetEval[54] | 10K | 1 | Human annotation | Hate speech | - | - |
| Tabular | Credit Card Fraud$^\beta$[55] | 284K | 1 | Human annotation | Credit card fraud | 28 | [56, 57] |
| | Adult$^\beta$[58] | 48K | 1 | Rule-based extraction | Salary | 14 | [30, 21, 22] |
| | Dry Bean[59] | 13K | 1 | Vision system-based annotation | Bean variety | 17 | - |
| | Car Evaluation[60] | 1K | 1 | Hierarchical decision model [60] | Car condition | 6 | [61] |
| | Mushroom$^\beta$[62] | 8K | 1 | - | Mushroom edibility | 22 | [56] |
| | COMPAS$^\beta$[63] | 6K | 1 | - | Recidivism | 28 | [21] |
| Time Series | Crop[64] | 7K / 16K | 1 | Hierarchical k-means tree with dynamic time warping [64] | Crop cover | $46 \times 1$ | - |
| | ElectricDevices[65] | 9K / 7K | 1 | Human annotation | Appliance-type | $96 \times 1$ | - |
| | MIT-BIH[66] | 23K / 4K | 1 | Human expert annotation | Arrhythmia | $256 \times 2$ | - |
| | PenDigits[67] | 7K / 3K | 1 | Human annotations | Handwritten digit | $16 \times 1$ | - |
| | WhaleCalls$^\beta$[68] | 11K / 2K | 1 | - | Whale call | $4,000 \times 1$ | - |

Table 1: **Summary of datasets.** AQuA currently includes a variety of datasets for different classification problems, varying in the number of classes, sources of annotations, and data modalities. All datasets except those marked with $\beta$ are multi-class.

# 4 Benchmark Design

## 4.1 Real-world, Popular Datasets, and Downstream Classification Models

**Datasets.** AQuA currently comprises of a collection of **17** popular real-world public datasets from **4** prevalent data modalities: *image*, *text*, *time-series* and *tabular*. To evaluate label error detection models across various practical scenarios, we carefully choose datasets with diversity in the following characteristics: (1) *classification problems* (*e.g.*, sentiment classification vs. hate speech detection), (2) *number of classes* (binary vs multi-class classification), (3) *relative prevalence of classes* (e.g., skewed datasets like Credit Card Fraud [55] and balanced ones like IMDb [53]), (4) *sources of annotations* (e.g., human vs rule-based annotation), and (5) *number of annotations per example* (e.g., CIFAR-10N labeled by 3 annotators). Table 1 summarizes the key characteristics of datasets included as a part of AQuA. In particular, to make comparison with prior work easier while maintaining diversity across practical scenarios, we try to include datasets that have been used frequently by prior work (see usage in Table 1) and preprocess them in a manner consistent with those works. We do not use any data augmentation during training. App A.3 provides detailed descriptions of the datasets.

**Classification models.** The ultimate goal of label cleaning is to train accurate downstream classifiers, but different studies use different classification models to measure the efficacy of their proposed label cleaning methods. To provide a level playing field for all cleaning methods, we include multiple classification model architectures for each data modality. Specifically, we include ResNet-18 [8], MobileNet [9] and FastViT-T8 [69] for image datasets, all-distilroberta-v1 [70, 71] and all-MiniLM-L6-v2 [72] for text datasets, ResNet-1D, PatchTST [73] and LSTM Fully Convolutional Network [74] for time-series datasets, and TabTransformer [75] and a Multi-Layer Perceptron for tabular datasets. While choosing classification models we prioritized *performant* methods with (1) *different architectures* and *inductive biases*, (2) ideally *pre-trained* using different strategies, and (3) *previously-used* either by label cleaning methods or task-relevant papers. App. A.4 and App. A.5 provide a detailed description of classification models and their hyperparameters, respectively.

## 4.2 Advanced Label Error Detection Methods

AQuA provides easy-to-use Application Programmer Interfaces (Fig. 4) for **4** state-of-the-art label error detection methods, namely Area Under Margin ranking (AUM) [24], Confident Learning [23], Contrastive and Influent Counter Example Strategy (CINCER) [30], and Model-free Label Error Detection (SimiFeat) [34]. Below, we provide a brief overview of these methods and their key ideas.

**Area Under the Margin Ranking (AUM) [24].** Given noisy data and access to the logits of a deep learning model, AUM exploits differences in training dynamics of clean and mislabeled samples to identify labeling errors. The key idea is to identify data points that do not contribute to the generalization of a model as labeling errors by leveraging the delicate tension between the label of a data point (via memorization) and its predicted label (via gradient updates), measured as the margin between the logits of a sample's assigned class and its highest unassigned class.

**Confident Learning** (`CON`) [23]. Given noisy data, confident learning estimates a data structure called *confident joint*, which is the joint probability distribution of observed noisy and latent true labels. The key idea is to leverage a model trained on held-out data drawn from the same (or similar) distribution to predict the probability that an example $\mathbf{x}_i$ belongs to its observed label $\mathbf{y}_i$. A low probability is then used as a heuristic-likelihood of $\mathbf{y}_i$ being a label error. The confident joint can then be used to identify labeling errors and estimate the noise rate.

**Contrastive and Influent Counter Example Strategy** (`CINCER or CIN`) [30]. CINCER treats the problem of identifying labeling errors as a sequential decision making problem where a domain expert can be queried to relabel suspicious examples. CINCER uses the same heuristic as AUM to identify labeling errors, but also identifies counter-examples in the data to serve as explanations of the model's suspicion.

```python
from aqua.models import TrainAqModel, ConvNet
from aqua.data import Aqdata, load_cifar
from aqua.reports import generate_report

# Load CIFAR-10 and ResNet-18
clf = ConvNet('resnet18')
data = load_cifar()
data.add_noise(noise_rate=0.2) # Add uniform
↪   noise

# Instantiate a cleaning method and classifier
cleaner = TrainAqModel(clf, method='aum')
label_errors = cleaner.find_label_issues(data)

# Remove data with label issues
data.clean_data(label_issues)
# Train a downstream model on cleaned data
y_preds = TrainAqModel(clf).fit_predict(data)
```

Figure 4: `AQuA` makes identifying label issues, and evaluating new and existing label error detection models simple.

**Model-free Label Error Detection (SimiFeat)** [34]. Unlike other methods, SimiFeat does not need a (pre-)trained model to identify labeling errors. Instead, it utilizes labels of the $k$ nearest neighbors to identify labeling errors based on the *clusterability* assumption, *i.e.* data points with similar features should have the same true label with high probability.

There are many methods to detect labeling errors, but we choose these methods as a *starting point* because they are recent, state-of-the-art, and have different inputs and core assumptions. While all these methods have existing public implementations, through `AQuA`, our goal is to create a one-stop shop for using and evaluating open-source label error detection models.

## 4.3 Evaluation

There is significant variance in the ways that label cleaning methods are evaluated. To rigorously, fairly, and systematically assess these models, we unify the breadth of experimental settings through the following three dimensions of evaluation.

**Supervision.** Identifying labeling errors in practice is an *unsupervised* problem since we do not know which data points are mislabeled. Hence, evaluating these methods is a challenging endeavor. Most studies in the literature gather noise labels either from human experts (*human-in-the-loop evaluation*) or by introducing synthetic label noise by design (*synthetic label noise*).

In human-in-the-loop evaluation, one or more human experts are asked to independently assess the true labels of data points identified as having erroneous labels [39, 4]. While this is a straightforward and precise evaluation method, it is in general unscalable, expensive, time-consuming, and limited to only measuring the *precision* of models (and not *recall*), because the experts are typically only shown data points which a model considers erroneous.

A much more common and scalable way of evaluating these methods is to introduce various kinds of synthetic label noise and measure a model's ability to detect them. There are many ways of introducing label noise, but injected noise may not always be reflective of the true noise that occurs in natural datasets, and hence identifying realistic noise injection strategies is an active area of research [33, 34, 39, 41, 76, 77]. Moreover, model evaluation may still be noisy because there may be mislabeled examples for which our pseudo-noise labels are negative (or *correctly labeled*).

**Hypotheses.** In general, existing studies evaluate two hypotheses: (1) *cleaning labels on the train set improves the performance of the downstream classifier on the test set*, and (2) *cleaning methods can accurately identify mislabeled data on the train set*. Hypothesis 1 is practical since the primary goal of identifying labeling errors is to train accurate and unbiased classifiers. However, appropriately regularized deep learning models are known to be naturally robust to some label noise. Hence, hypothesis 2 allows researchers to directly measure the efficacy of label cleaning techniques.

**Measures of goodness.** Different studies use different measures of predictive accuracy. While some measure error rate [24], others report the accuracy [33] or ROC-AUC [29] of their classification models. Similarly, for their cleaning methods, some studies report the $F_1$ score while others report the precision or recall [23, 24].

**More gaps in evaluation.** In addition to the lack of consistency, we believe that the experimental settings in many studies are occasionally (1) *unrealistic*, e.g., adding label noise to more than half (sometimes up to $80\%$) of the data points [24, 23]; and (2) *uni-dimensional*, e.g., reporting only one metric of predictive performance.

`AQuA`**'s design.** To enable a realistic, multi-faceted and holistic evaluation of label error detection models, we implement **7** popular label noise injection techniques and multiple metrics of predictive performance. Specifically, for single-label datasets, we implement asymmetric [34], class-dependent [76], instance-dependent [33], and uniform [76] noise, and for datasets with labels from multiple annotators, we implement dissenting label, dissenting worker, and crowd majority [39]. In terms of metrics of predictive accuracy, we implement $F_1$, accuracy, (*weighted*) precision, recall, area under ROC curve (ROC-AUC), average precision (PR-AUC), and error rate. We are in the process of implementing some other metrics beyond predictive accuracy, such as generalization [78] and robustness [79] of models. Our hope is that `AQuA`'s *config-driven* design will allow non-technical users to integrate it into their labeling workflows and researchers to add new models, datasets, and evaluation pipelines seamlessly. Our choice of datasets and downstream classifiers ensures that the computational complexity of running experiments is not prohibitive. Finally, we make all code, pre-trained models, and experimental logs open-source to enable rigorous and fair evaluation of models.

# 5 Experiments, Results and Discussion

| **Datasets** | **Uniform** | | | | **Asymmetric** | | | | **Class-dependent** | | | | **Instance-dependent** | | | |
|---|---|---|---|---|---|---|---|---|---|---|---|---|---|---|---|---|
| | AUM | CIN | CON | SIM | AUM | CIN | CON | SIM | AUM | CIN | CON | SIM | AUM | CIN | CON | SIM |
| CIFAR-10 | 73.3 | 74.1 | 45.6 | **76.7** | 74.3 | 70.8 | 47.7 | **75.5** | 93.5 | 80.5 | 42.6 | **93.6** | 68.0 | 69.9 | 44.8 | **70.9** |
| Clothing-100K | 75.0 | 70.0 | **76.6** | 76.5 | 74.2 | 68.4 | 73.6 | **75.7** | 76.3 | 71.2 | 74.0 | **81.2** | 69.4 | 65.1 | **72.9** | 71.6 |
| NoisyCXR | **75.2** | 74.4 | 43.2 | 74.5 | **73.7** | 71.5 | 39.5 | 73.5 | 84.7 | 78.7 | 31.4 | **88.4** | 68.0 | 69.8 | 43.3 | **72.1** |
| IMDb | 75.6 | 73.3 | 58.4 | **78.5** | 75.7 | 74.3 | 59.5 | **78.7** | 92.1 | 91.0 | 62.8 | **95.0** | 69.7 | 70.2 | 56.4 | **74.5** |
| TweetEval | 75.3 | 75.2 | 58.9 | **77.7** | 75.8 | 76.0 | 57.4 | **77.6** | 69.2 | 67.9 | 52.4 | **70.2** | 69.6 | 69.6 | 62.4 | **73.2** |
| Credit Fraud | 75.8 | 75.8 | 73.3 | **78.1** | 75.7 | 75.8 | **80.0** | 76.7 | 63.3 | 63.0 | **87.2** | 72.0 | 69.5 | 69.4 | **74.3** | 73.5 |
| Adult | 75.7 | 75.8 | 72.9 | **78.5** | 75.8 | 75.8 | 66.9 | **77.5** | 63.6 | 64.6 | 61.2 | **64.9** | 69.6 | 70.2 | 68.7 | **72.4** |
| Dry Bean | 75.7 | **91.6** | 42.1 | 82.2 | 75.7 | **84.9** | 39.0 | 80.3 | 87.2 | **95.0** | 35.4 | 92.1 | 69.5 | **83.1** | 35.8 | 77.5 |
| Car Evaluation | 75.3 | 83.5 | 77.4 | **84.1** | 75.6 | 80.2 | 75.7 | **81.6** | 77.3 | **87.5** | 83.2 | 81.2 | 70.1 | **78.8** | 78.5 | 77.0 |
| Mushrooms | 76.0 | 82.5 | 62.7 | **85.2** | 75.7 | 80.7 | 66.3 | **83.0** | 99.3 | **100** | 75.5 | 99.8 | 69.5 | **75.4** | 64.1 | 74.3 |
| COMPAS | 75.8 | 74.9 | 63.2 | **75.9** | 75.8 | 74.8 | 64.6 | **76.5** | 55.5 | 57.1 | 52.9 | **57.7** | 69.5 | 69.4 | 61.0 | **73.1** |
| Crop | 76.0 | **79.0** | 16.3 | 73.1 | **75.8** | 73.6 | 16.2 | 70.1 | 29.1 | 40.8 | 51.2 | **63.7** | **69.5** | 63.2 | 16.3 | 63.8 |
| Electric Devices | 75.8 | **82.2** | 35.0 | 79.3 | 75.7 | **78.6** | 35.3 | 75.8 | 37.8 | 50.5 | 55.9 | **68.3** | 69.9 | **71.5** | 32.7 | 69.2 |
| MIT-BIH | 75.6 | **88.4** | 49.7 | 83.3 | 75.7 | **83.0** | 51.3 | 78.4 | 68.2 | 75.7 | 45.4 | **80.6** | 69.6 | **78.4** | 48.1 | 75.2 |
| PenDigits | 75.8 | **89.0** | 23.1 | 73.4 | 75.7 | **83.1** | 23.4 | 72.7 | 46.7 | 44.9 | 53.5 | **78.4** | 69.9 | **76.0** | 19.8 | 68.1 |
| WhaleCalls | 75.6 | 74.9 | 60.3 | **77.3** | 75.7 | 75.5 | 61.8 | **77.2** | 42.3 | 44.7 | **52.4** | 47.1 | 69.6 | 69.1 | 59.2 | **71.2** |

Table 2: Performance evaluation of cleaning methods to detect erroneous labels across different types of synthetic noise added to the train set in terms of weighted $F_1$, averaged across noise rates and downstream models.

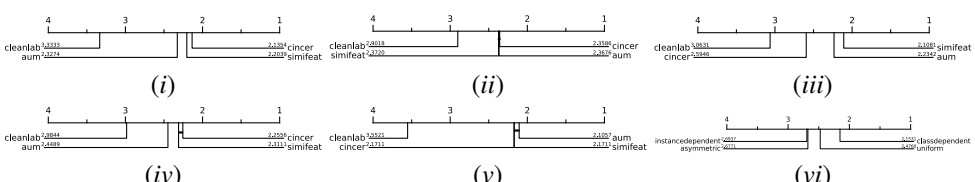

*(i)*      *(ii)*      *(iii)*

*(iv)*      *(v)*      *(vi)*

Figure 5: *Critical difference diagrams representing rankings of cleaning methods* across: (*i*) all datasets, (*iii*) only image or (*iv*) only text datasets. (*v*) also shows the ranking of cleaning methods across all datasets when accuracy is measured instead of weighted $F_1$ (c.f. *i*). Finally, (*ii*) represents the performance of *downstream models* trained using cleaned labels, and (*vi*) performance of all cleaning methods disaggregated by noise type.

| Datasets | No Noise Injected | | | | | Uniform | | | | | Asymmetric | | | | | Class-dependent | | | | | Instance-dependent | | | | |
|---|---|---|---|---|---|---|---|---|---|---|---|---|---|---|---|---|---|---|---|---|---|---|---|---|---|
| | NON | AUM | CIN | CON | SIM | NON | AUM | CIN | CON | SIM | NON | AUM | CIN | CON | SIM | NON | AUM | CIN | CON | SIM | NON | AUM | CIN | CON | SIM |
| CIFAR-10 | **74.3** | 74.1 | 73.0 | 46.0 | 73.5 | 63.2 | 62.6 | **65.0** | 36.9 | 63.4 | 58.1 | **63.1** | 62.2 | 38.3 | **63.1** | 71.0 | **71.7** | 70.5 | 46.2 | 67.5 | 57.7 | 60.2 | **62.1** | 34.0 | 56.8 |
| Clothing-100K | **90.9** | 90.7 | 90.5 | 90.8 | 90.8 | 82.5 | 79.5 | 83.2 | **85.4** | 83.2 | 82.8 | 81.4 | 79.2 | 79.4 | **83.1** | 80.8 | 83.3 | 82.1 | 83.8 | **86.1** | 78.9 | 74.6 | 71.7 | **81.6** | **81.6** |
| NoisyCXR | 56.0 | 56.5 | 56.7 | 25.2 | **57.0** | 49.6 | 49.4 | **52.2** | 19.1 | 48.2 | 49.6 | 48.8 | **50.6** | 18.0 | 47.9 | 54.2 | 54.8 | **55.8** | 18.6 | 53.8 | 46.4 | 46.7 | **48.3** | 18.9 | 46.7 |
| IMDb | 84.9 | 87.5 | 89.2 | 69.6 | **90.3** | 69.3 | 65.6 | 68.2 | 64.5 | **70.8** | 74.9 | 67.7 | 76.8 | 66.4 | **80.6** | 87.1 | 85.5 | **89.1** | 84.4 | 87.4 | 65.3 | 62.3 | **69.2** | 64.4 | 64.5 |
| TweetEval | 73.6 | 73.6 | **77.1** | 65.1 | 76.8 | 71.3 | 72.4 | **74.2** | 53.8 | 73.2 | 71.3 | 68.4 | 71.9 | 61.8 | **72.4** | **77.7** | 74.8 | 70.6 | 49.8 | 67.4 | 68.5 | 69.4 | **71.7** | 71.5 | 63.1 |
| Credit Fraud | **100** | 99.9 | 99.9 | 99.9 | 99.9 | **99.9** | **99.9** | **99.9** | 88.8 | **99.9** | **99.9** | **99.9** | **99.9** | 99.8 | **99.9** | 99.6 | 66.6 | **99.8** | **99.9** | 66.6 | **99.9** | 99.8 | **99.9** | 88.7 | 99.8 |
| Adult | **84.0** | **84.0** | **84.0** | 79.9 | **84.0** | 82.1 | 82.1 | **83.0** | 75.0 | 81.9 | **83.3** | 83.0 | 83.0 | 77.9 | 83.0 | 81.8 | 82.3 | **83.1** | 81.9 | 80.4 | **82.5** | 80.6 | 81.5 | 73.1 | 81.1 |
| Dry Bean | **91.9** | 90.9 | 91.2 | 57.2 | 90.6 | 89.5 | 90.9 | **91.4** | 60.6 | 78.6 | 85.6 | 87.0 | **89.3** | 55.1 | 87.4 | **91.4** | 91.0 | 90.5 | 29.8 | 90.4 | 87.3 | 83.6 | 84.5 | 40.0 | **87.5** |
| Car Evaluation | **93.4** | 92.5 | 85.9 | 57.6 | 92.0 | **83.8** | 82.5 | 80.0 | 66.5 | 81.1 | **86.1** | 85.2 | 74.1 | 62.7 | 82.6 | **86.9** | 83.2 | 78.0 | 57.6 | 83.9 | **82.6** | 81.5 | 75.3 | 60.2 | 80.5 |
| Mushrooms | 99.7 | **99.9** | 99.6 | 99.7 | **99.9** | 98.3 | 98.2 | 97.8 | 89.0 | **98.7** | 97.9 | 97.5 | **98.4** | 87.3 | 96.5 | 99.5 | **100** | 99.3 | 99.1 | **99.9** | 95.3 | **96.9** | 96.4 | 81.1 | 95.8 |
| COMPAS | 67.2 | **67.3** | 66.2 | 63.7 | 66.6 | 65.6 | 62.1 | **65.9** | 58.6 | 65.3 | 66.1 | 64.4 | **66.2** | 46.9 | 65.6 | 54.3 | 65.1 | 63.8 | 35.5 | **66.2** | 61.6 | 63.0 | 61.8 | 48.6 | **63.7** |
| Crop | **39.1** | 38.7 | 35.5 | 8.4 | 37.8 | 33.1 | **37.2** | 36.2 | 7.3 | **37.9** | **34.1** | 31.5 | 32.8 | 7.2 | 33.4 | **32.3** | 31.2 | 29.5 | 7.3 | 28.9 | 27.7 | 27.8 | 29.9 | 5.7 | **34.5** |
| Electric Devices | 45.3 | **48.0** | **48.0** | 29.8 | 46.7 | 41.8 | 42.6 | **44.8** | 27.3 | 42.1 | 42.5 | 41.3 | 41.3 | 26.9 | **42.7** | 30.9 | 30.9 | **32.1** | 24.2 | 31.7 | 39.3 | 36.6 | 38.3 | 23.1 | **40.4** |
| MIT-BIH | 72.7 | 65.1 | **81.2** | 55.7 | 72.5 | 73.2 | 70.1 | **80.1** | 61.7 | 74.7 | 71.3 | 68.4 | 69.2 | 46.3 | 69.6 | 72.6 | 73.9 | 74.4 | 56.9 | **78.0** | 63.6 | 68.1 | 70.9 | 52.2 | **71.5** |
| PenDigits | 64.8 | **65.2** | 64.3 | 39.5 | 64.5 | 62.6 | **64.7** | 64.4 | 24.6 | 64.3 | 58.1 | **59.1** | 57.8 | 22.9 | 59.0 | 43.9 | **46.5** | 46.4 | 15.3 | 45.3 | 59.2 | 56.4 | 57.7 | 14.8 | **59.7** |
| WhaleCalls | **68.2** | 34.3 | 50.9 | 52.7 | 53.0 | 48.7 | 44.5 | **51.0** | 43.7 | 50.4 | 48.8 | **53.6** | 47.4 | 45.3 | 47.2 | 42.5 | 43.3 | **47.1** | 41.6 | 42.4 | 48.5 | 50.9 | **58.5** | 44.5 | 47.5 |

Table 3: Impact of label noise and each cleaning method on weighted $F_1$ score of a downstream model for each modality on the test set, averaged across noise rates and downstream models. Highlighted cells indicate better performance than that obtained without label cleaning (NON).

We conduct several experiments to support AQuA's design choices and demonstrate its utility in providing a comprehensive and holistic evaluation of machine learning models in the presence of label noise.

**Experimental Setup and Hyper-Parameter Tuning.** We run experiments for all combinations of cleaning methods (AUM (AUM), confident learning (CON), CINCER (CIN) and SimiFeat (SIM), including no label cleaning (NON), noise types (*asymmetric*, *class-dependent*, *instance-dependent* and *uniform*); for four different noise rates (0%, 2%, 10% and 40%), for a total of **2400 unique experiments**. We conduct experiments using three distinct classification architectures for image and time-series data, and two different architectures for text and tabular data. To account for class imbalance in some datasets, we report the $F_1$ weighted by the support of each class. Results for all other evaluation metrics can be found in App. A.8. We also adopt critical difference diagrams [80] to succinctly represent comparisons between multiple cleaning methods and other independent variables (e.g., data modality and noise type) on multiple datasets. These diagrams represent the average ranks of methods across datasets while grouping those with insignificant difference[5]. We tuned hyper-parameters of all the classification and cleaning methods till they performed reasonably well on average on all the datasets using hyper-parameter grids used by prior work and reported in App. A.5[6]. Finally, all our experiments were carried out on a computing cluster, with a typical machine having 128 AMD EPYC 7502 CPUs, 503 GB of RAM, and 8 NVIDIA RTX A6000 GPUs.

**Research Questions.** We aim to answer the following research questions through our experiments:

- *Which is the best cleaning method in terms of (i) its ability to identify synthetically injected label noise, and (ii) performance of the downstream classifier trained its cleaned labels?*
- *Do the rankings of cleaning methods differ across different (i) types of synthetic label noise, (ii) data modalities, and (iii) evaluation metrics (weighted $F_1$ versus accuracy)?*

## 5.1 Insights from Large-scale Experiments using AQuA

Tables 3, 2, and Fig. 5 report results from all our experiments aggregated by noise rate, and downstream classification models. Below we highlight some of our key findings. Due to lack of space, we defer finer grained results to App. A.8.

**Best cleaning method.** Overall, we found SimiFeat (SIM) [34] to be the best cleaning method in terms of its ability to identify synthetically injected label noise, closely followed by CINCER (CIN) [30] (Fig. 5(*i*)). However, these differences shrink when evaluating cleaning methods using the performance of the downstream model trained using their cleaned labels (Fig. 5(*ii*)). Confident learning (CON) [23] consistently performed the worst among all the evaluated methods.

---

[5]To form cliques, we abandon the posthoc test in favor of pairwise tests with Holm's correction for multiple testing based on prior work [81, 82]

[6]We deliberately did not perform extensive hyper-parameter tuning to not overfit to already existing label noise in the original datasets. Also, in practice it is unclear how to tune these cleaning methods well, without explicit knowledge of where the label errors are.

**Deep learning models are inherently robust to label noise.** Perhaps unsurprisingly, we found that most downstream classifiers were reasonably robust to synthetic label noise, as can be seen from the insignificant difference between the setting where datasets were not explicitly cleaned (`NON`), compared to when they were cleaned using `SIM`, `CIN` and `AUM`. These results also illustrate the importance of measuring both hypotheses (performance of cleaning methods versus downstream models) when evaluating the performance of ML models in the presence of label noise.

**Adding label noise can sometimes improve model performance.** In the context of class-dependent or uniform noise, label noise serves as regularization to prevent models from overfitting. This phenomenon is not specific any one modality, but happens for multiple modalities, datasets, and noise types too, for example Electric Devices (time-series) under uniform noise, MIT-BIH (time-series), and Dry Bean (tabular) for class-dependent noise, in Table 16. Moreover, deep learning optimization is highly non-convex, so adding some noise might help the model reach the global minima by traversing an alternative path within the loss landscape.

**Impact of `AQuA`'s design choices.** We found that cleaning methods perform differently for different data modalities. For instance, all cleaning methods barring `CON` perform on par on image datasets *(iii)*, but on tabular data *(iv)*, `AUM` performs significantly worse than `CIN` and `SIM`. This may be due to a variety of reasons beyond cleaning methods: size and nature of datasets, inductive biases of downstream classifier, and the quality of feature representations [34]. We also observed that some types of label noise are easier to detect than others. For example, uniform noise and asymmetric noise were the easiest to detect, cleaning methods found it much hard to detect instance and class-dependent noise *(vi)*. Finally, we noticed differences in model rankings when measuring different evaluation metrics. As an example, the difference between `CIN` and `AUM` vanishes when we measure the accuracy *(v)* of the cleaning methods instead of their weighted $F_1$ *(i)*. These findings highlight the need to evaluate label error detection methods across multiple datasets from different modalities, noise types and evaluation metrics.

## 6 Conclusion and Future Work

We propose the first benchmark designed to rigorously evaluate machine learning models in the presence of label noise. We also elucidate the design space of these methods to not only enable ML practitioners to choose the right label cleaning tool for their data, but also foster academic research on the label noise problem. We demonstrate `AQuA`'s utility by running large-scale experiments to glean several interesting findings. We believe that, as a benchmarking toolkit, `AQuA` would benefit from more cleaning methods, datasets, synthetic label noise injection strategies, and evaluation metrics.

Our short-term goals include experimenting with multi-annotator label noise, measuring the impact of feature noise on time-series and image data in comparison to label noise, incorporating several metrics for model generalization, robustness and fairness, and including audio datasets. While other types of noise are beyond the scope of this work, we believe that multi-annotator, multi-class multi-label, and noise in regression problems are exciting avenues of future work, and AQuA's modular design will enable researchers to experiment with both multi-annotator and multi-class multi-label classification problems easily. We restrict ourselves to multi-class but single-label classification (as opposed to multi-label classification).

We believe that future work on label error detection should address label issues in the multi-label classification and regression settings. We believe that our work on AQuA can both harness and facilitate the development of foundation models in the two ways: (1) foundation models can be used to identify labeling errors, without explicit supervision, and (2) methods within AQuA can be use to identify labeling errors which can affect foundation model pre-training and fine-tuning. We also believe that future work shoul

## 7 Limitations, Biases, and Social Impacts

We acknowledge the potential adverse impact of large-scale experimentation on the environment, but believe that our publicly accessible code and experimental findings can significantly reduce resource consumption for ML practitioners in this field. Label error detection models might perpetuate existing biases and impact the fairness of models. We included the Adult dataset, that is frequently used in the fairness literature, in AQuA, to evaluate the impact of label errors on the fairness of models. We

would also like to acknowledge that our experiments were carried without extensive hyper-parameter tuning. Moreover, hyper-parameters for cleaning methods and downstream classifiers were chosen based on model performance on the observed training set and fixed throughout the training process. We futher discuss these design choices and their limitations in Appendix A.6.

## Acknowledgments and Disclosure of Funding

We would like to thank Cherie Ho and Jack H. Good for their useful comments on initial drafts of the paper. This work was partially supported by the National Institutes of Health under awards R01HL141916, 1R01NS124642-01, and 1R01DK131586-01, and by the U.S. Army Research Office and the U.S. Army Futures Command under Contract No. W911NF-20-D-0002. The content of the information does not necessarily reflect the position or the policy of the government and no official endorsement should be inferred.

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
