# A  Appendix

## A.1  A Design Space of Labeling Error Detection Models

In this section, we provide some more details on some of the key design decisions of various popular methods which enable machine learning in the presence of label noise.

**Noise Transition Matrix.**   Many studies [83, 41, 33] explicitly estimate a probabilistic data structure called the noise transition matrix. A noise transition matrix $\mathbf{T}$ encodes the joint [23], or more frequently the conditional probability [83, 33] of distribution of latent labels $y_i^*$ and observed noisy labels $y_i$, such that $\mathbf{T}_{ij} \triangleq \mathbb{P}(y = j \mid y^* = i; \mathbf{x})$. The noise transition matrix can be estimated in many different ways, e.g. (using *anchor points*, labels of nearest neighbors (*clusterability*), and pre-trained models). Similarly, the matrix can be either used to identify labeling errors explicitly [23], or train robust machine learning models using modified loss functions. We note two key assumptions that a lot of these studies make, which might be violated in practice: (1) noise transition matrix is independent of the features of the data points, and (2) only a small fraction of the labels are noisy. To this end, recent studies have focused on designing novel techniques to estimate noise transition matrix while relax some of these assumptions (e.g., [42, 41]). Below we briefly discuss three ways in which a noise transition matrix can be estimated, namely using *anchor points*, *nearest neighbours* and *pre-trained models*, and one technique to use these matrices to train robust ML models.

**Estimating T using Anchor Points.**   Intuitively, anchor points are samples in the training data which are highly likely to belong to a certain class. In particular, a data point $\mathbf{x}$ is an anchor for a class $i \in C$ if $\mathbb{P}(y^* = i \mid \mathbf{x}) = 1 - \epsilon$, where $\epsilon \to 0$. If $\epsilon = 0$, then $\mathbb{P}(Y = j \mid \mathbf{x}) = \sum_{k=1}^{C} \mathbf{T}_{kj} \mathbb{P}(Y = k \mid \mathbf{x}) = \mathbf{T}_{ij}$. Hence, $\mathbf{T}$ can be derived by evaluating the posterior probability that a anchor point belongs to noisy classes [27, 31]. While intuitive, using anchor points to estimate the transition matrix is not scalable, especially in scenarios where the number of classes is high and training data points is small since training a model which predicts the probability of noisy labels is challenging. Moreover, unavailability and identifiability of anchor points can limit the efficacy of these approaches, even if the posterior distribution can be learned accurately. Lastly, these methods lack the flexibility to extend to more complicated noise settings.

**Estimating T using Clusterability.**   These methods assume that data points with similar features should have the same class labels. Unlike previous methods based on anchor points, if good features are available off the shelf, then methods can be considered *model-free*. Otherwise, reasonable features can automatically derived from intermediate-layer representations of deep learning models [33, 45]. While these methods are intuitive, they rely on finding a good distance metric between the features. Moreover, these models might identify outliers as label noise, preventing the downstream classifier from learning meaningful data points.

**Estimating T using pre-trained models.**   The key idea is to leverage a model trained on held-out data drawn from the same (or similar) distribution to predict the probability that an example $\mathbf{x}_i$ belongs to its observed label $\mathbf{y}_i$. A low probability is then used as a heuristic-likelihood of $\mathbf{y}_i$ being a label error. A careful count of these data points can then be use to estimate $\mathbf{T}$ [23].

But not all studies use pre-trained models to estimate $\mathbf{T}$. With the advent of pre-trained large language models, exploring their utility in detecting labeling errors [39] and studying their performance in the presence of label noise [84] is an active area of research. Recently, [39] used the loss of a large language model to identify labeling errors, under the assumption that these models will exhibit large losses for erroneous data points. Another study demonstrated that unlike classical machine learning models, large language models may already be robust to label noise [84].

**Using T to train robust ML models.**   We previously discussed how $\mathbf{T}$ can be used to identify labeling errors. There's another body of work which relies on the noise transition matrix to modify loss functions to make train machine learning models robust to label noise [48, 47, 85]. For example, given the noise transition matrix, Patrini et al. [27] introduced forward and backward loss corrections, involving simple operations like matrix inversion and multiplication to make existing loss functions robust to noisy labels.

Next, we provide a brief overview of techniques which do not explicitly estimate the noise transition matrix. We categorize these approches into three categories, primarily based on their key ideas: (1) approaches relying on the *training dynamics* of ML models, (2) *multi-network* approaches, and (3) approaches which leverage labels from *multiple annotators*.

**Approaches based on Training Dynamics.** These approaches exploit differences in training dynamics of clean and mislabeled samples to identify labeling errors. For example, Area under Margin Ranking [24] identifies data points that do not contribute to the generalization of a model as labeling errors by leveraging the delicate tension between the label of a data point (via memorization) and its predicted label (via gradient updates), measured as the margin between the logits of a sample's assigned class and its highest unassigned class. On the other hand, Yue and Jha [56] obtain the loss curves for each instance in a dataset from a neural network trained on a noisy training set, and apply clustering on these losses to separate clean and noisy samples.

**Multi-network approaches.** All methods we have discussed thus far use one model to identify labeling errors. But a few studies have leveraged two models to identify labeling errors, using either knowledge distillation [35, 36], or meta-learning [37, 38]. These methods are expected to better identify different types of label errors as they rely on different models of different sizes and inductive biases.

The key idea of methods based on knowledge-distillation is to use a larger teacher network to supervise the training of a smaller student network. The teacher model identifies correctly labeled data points, and trains the student network on these samples only [35]. Instead of training the student and teacher models sequentially, some other studies propose to train the models simultaneously [36, 44].

A few studies utilize similar ideas to knowledge-distillation, instead using meta-learning to train robust machine learning models. For example, Zheng et al. [38] propose a Meta Label Correction framework, where a label correction network acts as a meta-model to correct noisy labels, while the main model leverages these corrected labels. Some other methods re-weight training samples based on their gradient directions. These approaches generally comprise of a target and a meta-deep neural network, where the latter is trained on a clean validation set, and guides the training of the target network via sample re-weighting[37].

**Multi-annotator labels.** These approaches are based on the premise that certain annotation tasks are inherently ambiguous, and even domain experts find it difficult to correctly label such instances. These methods aim to use multiple annotator labels to better model the noise transition matrix using the correlation between labels from different annotators to better estimate ground-truth consensus. These approaches are particularly useful for the healthcare domain due to the limited number of annotators but high variability of annotations[86]. Bernhardt et al. [29] introduce active label cleaning based on "re-active learning", where they allow for re-annotation of already labeled instances in an active learning training scheme. Their proposed framework determines relabelling priority on the basis of the predicted posteriors from a classification model. Label cleaning is done over multiple iterations, and within each iteration, samples are initially ranked according to label prediction correctness and annotation difficulty. Each prioritized label is reviewed by multiple annotators until a consensus is formed using all generated labels. Drawing a leaf out of the crowd-sourcing literature, some other studies explicitly model the confusion matrix of each annotator to identify mislabeled data [7].

### A.2 Relation with Weakly Supervised Learning

AQuA serves two purposes: (1) as a benchmarking tool to evaluate methods that identify labeling errors, (2) and generally as a tool to identify labeling errors in a dataset and choose an appropriate cleaning method. Weakly supervised learning is a class of methods that learn from imperfect and weak sources of supervision to label datasets (see Zhang et al. [87] and Goswami et al. [88] as examples). The labels arising from these methods are indeed noisy. Methods in AQuA can therefore be used to clean datasets labeled using weakly supervised methods.

### A.3 Datasets and their characteristics

AQuA currently comprises of a collection of **17** popular real-world public datasets from **4** prevalent data modalities: *image*, *text*, *time-series* and *tabular*. To evaluate label error detection models across

| Modality | Dataset | # Train / Test | # Annotators/sample | Label Source | Classification Task | Sample Size | Usage |
|---|---|---|---|---|---|---|---|
| Image | CIFAR-10N[49] | 50K / 10K | 3 | Human annotation | Object | $32 \times 32 \times 3$ | [50, 34] |
| | CIFAR-10H[16] | 0 / 10K | 47–63 | Human annotation | Object | $32 \times 32 \times 3$ | [29] |
| | Clothing100K[51] | 100K | 1 | Web-labeled | Image | $256 \times 256 \times 3$ | [24, 4, 34] |
| | NoisyCXR[52] | 26K / 3K | 1–XX | Human expert annotation | Pneumonia | $1024 \times 1024 \times 1$ | [29] |
| Text | IMDb$^\beta$[53] | 25K / 25K | 1 | Human annotation | Sentiment | - | [27, 47, 4] |
| | TweetEval[54] | 10K | 1 | Human annotation | Hate speech | - | - |
| Tabular | Credit Card Fraud$^\beta$[55] | 284K | 1 | Human annotation | Credit card fraud | 28 | [56, 57] |
| | Adult$^\beta$[58] | 48K | 1 | Rule-based extraction | Salary | 14 | [30, 21, 22] |
| | Dry Bean[59] | 13K | 1 | Vision system-based annotation | Bean variety | 17 | - |
| | Car Evaluation[60] | 1K | 1 | Hierarchical decision model [60] | Car condition | 6 | [61] |
| | Mushroom$^\beta$[62] | 8K | 1 | - | Mushroom edibility | 22 | [56] |
| | COMPAS$^\beta$[63] | 6K | 1 | - | Recidivism | 28 | [21] |
| Time Series | Crop[64] | 7K / 16K | 1 | Hierarchical k-means tree with dynamic time warping [64] | Crop cover | $46 \times 1$ | - |
| | ElectricDevices[65] | 9K / 7K | 1 | Human annotation | Appliance-type | $96 \times 1$ | - |
| | MIT-BIH[66] | 23K / 4K | 1 | Human expert annotation | Arrhythmia | $256 \times 2$ | - |
| | PenDigits[67] | 7K / 3K | 1 | Human annotations | Handwritten digit | $16 \times 1$ | - |
| | WhaleCalls$^\beta$[68] | 11K / 2K | 1 | - | Whale call | $4,000 \times 1$ | - |

Table 4: **Summary of datasets.** AQuA currently includes a variety of datasets for different classification problems, varying in the number of classes, sources of annotations, and data modalities. All datasets except those marked with $\beta$ are multi-class.

various practical scenarios, we carefully choose datasets with diversity in the following characteristics: (1) *classification problems* (*e.g.*, sentiment classification vs. hate speech detection), (2) *number of classes* (binary vs multi-class classification), (3) *relative prevalence of classes* (e.g., skewed datasets like Credit Card Fraud [55] and balanced ones like IMDb [53]), (4) *sources of annotations* (e.g., human vs rule-based annotation), and (5) *number of annotations per example* (e.g., CIFAR-10N labeled by 3 annotators). Table 4 summarizes the key characteristics of datasets included as a part of AQuA. In particular, to make comparison with prior work easier while maintaining diversity across practical scenarios, we try to include datasets that have been used frequently by prior work (see usage in Table 4). Below we provide a brief description of datasets included in AQuA:

**CIFAR-10N [49]:** CIFAR-10N is a human-annotated dataset built upon the CIFAR-10 dataset, which is a 10-class image dataset consisting of $32 \times 32$ color images, with each class containing a total of 6000 images. The classes are airplanes, cars, birds, cats, deer, dogs, frogs, horses, ships, and trucks, and they are all mutually exclusive. CIFAR-10N enables researchers to evaluate inter-annotator agreement-based metrics, since it contains 3 human-annotated labels per sample obtained from Amazon Mechanical Turk. The training set of the CIFAR-10N datasets consists of a "clean label" along with three human-annotated labels on the training set of CIFAR-10.

**CIFAR-10H [16]:** Like CIFAR-10N, the CIFAR-10H data also comprises of multiple human annotations of the CIFAT-10 data. But unlike, CIFAR-10N, only the test set samples are annotated by crowd workers in Amazon Mechanical Turks. Each data point is annotated by 47 to 63 human annotators, making CIFAR-10H a repository of human perceptual uncertainty on the labels of CIFAR-10's testing data.

**Clothing100K [51, 24]:** Clothing100K is a subset of the Clothing1M dataset, which includes over 1 million clothing images belonging to 14 different classes. The labels of data points are obtained by crawling online shopping websites, and therefore expected to reflect real-world noise. Due to the presence of real-world noise, most recently proposed studies evaluate their methods on Clothing1M or its subsets. To speed up our experiments, we only use a subset of 100,000 samples to train and evaluate models in AQuQ [24].

**NoisyCXR [52]:** NoisyCXR dataset is a multi-class dataset comprising of chest X-rays, with the primary goal of detecting pneumonia in lungs. Like CIFAR-10N and CIFAR-10H, this dataset too comprises of one or more expert-annotated labels. We included NoisyCXR since many data points have more than one expert labels and the dataset presents practical challenges prevalent in deploying machine learning in the real world such as ambiguously labels and vague samples.

**IMDb [53]:** The IMDb dataset consists of 50,000 highly polarized textual movie reviews from IMDb with labels for binary sentiment classification. Each sample is labeled either negative or positive. Using the 10-score rating system on IMDb, the review text is labeled negative when its star rating is $<= 4$, and it is considered positive when the star rating is $>= 7$. Any sample with

scores greater than 4 but less than 7 is considered neither positive nor negative and excluded from the dataset. The training and testing splits contain 25,000 samples each, and each contains an equal number of positive and negative reviews.

**TweetEval [54]:** TweetEval is a multi-task textual benchmark comprising of labels for seven different tasks including topic classification, sentiment analysis, irony detection, hate speech detection, offensive language detection, emoji prediction, and emotion analysis. For our benchmark, we chose the hate speech detection task primarily due to its size (i.e. the number of data points associated with hate speech labels was much larger than some other task), and real-world impact. These data points are obtained from Twitter and focus on the detection of hateful tweets targeting women and immigrants. The dataset contains an even number of training, validation, and testing samples.

**Credit Card Fraud Detection Dataset [55]:** This is a real-world binary classifcation tabular dataset obtained from European credit card holders' transactions in September 2013. We included this dataset due to its highly unbalanced class distribution: only a small fraction of 0.172% of the samples are labeled as fraud. The attribute values for each sample are obtained after principle components analysis transformation to protect users' transaction information. Only the time and amount are not transformed and used as is.

**Adult [58]:** The Adult dataset, also known as the "Census Income" dataset, is a tabular binary class classification dataset used to predict whether or not an individual has an annual salary of $>=$ USD $50,000$. The data is collected and extracted from the 1994 Consensus database under the conditions: `((AAGE>16) && (AGI>100) && (AFNLWGT>1) && (HRSWK>0))`. It contains attributes like age, work class, fnlwgt (the final weight, i.e., the number of people each row represents), education, education number, marital status, occupation, relationship, race, sex, capital gain, capital loss, hours per week, and native country. We included this dataset since it is widely used to evaluate advances in the context of the fairness of machine learning models.

**Dry Bean [59]:** This is a tabular multi-class classification dataset for classifying a sample into one of seven types of beans. It was created by clicking high-resolution images of 13,611 bean grains, and these images were subjected to segmentation and feature extraction, resulting in a total of 16 attributes: 12 based on dimensions and 4 based on shape form.

**Car Evaluation [60]:** This is a tabular multi-class classification dataset for evaluating a car's condition. It has class values "unacceptable", "acceptable", "good" and "very good". It was generated using a hierarchical decision model which evaluated cars based on three intermediate concepts: `TECH`, `PRICE`, and `COMFORT`. These intermediate concepts were further linked to 6 lower level concepts. Owing to this underlying structure, this dataset can be used for testing constructive induction and structure discovery methods.

**Mushroom [62]:** The Mushroom dataset is a tabular binary class classification dataset, created from descriptions of hypothetical records of 23 species of gilled mushrooms belonging to the Lepiota and Agaricus families. These 22 attribute, mushroom records were derived from The Audubon Society Field Guide to North American Mushrooms. Each species was originally labeled as `definitely poisonous`, `definitely edible`, or `unknown edibility`. However, the dataset creators merged the definitely poisonous and unknown edibility classes into one `poisonous` class.

**COMPAS [63]:** The Correctional Offender Management Profiling for Alternative Sanctions (COMPAS) dataset is obtained from pretrial COMPAS algorithm jurisdiction from Broward County Sheriff's Office in Florida to evaluate recidivism in cases in a two-year span. In COMPAS jurisdiction, each defendant receives three scores which include "Risk of Recidivism," "Risk of Violence" and "Risk of Failure to Appear", which are based on the answers in the COMPAS survey [63]. The data was compiled using the person's name, date of birth, and race, which sometimes could be incorrectly labeled and portray a wrong COMPAS score corresponding to the criminal records. Like Adult, the COMPAS dataset is also one of the most commonly used datasets to evaluate the fairness of machine learning models.

**Crop [64]:** The Crop dataset is a multi-class tabular dataset, obtained from the European Space Agency Sentinel-2 and NASA Landsat-8 program to demonstrate the change of landscape through

its pixel over a period of time data. The change is observed through the change in the colors of the geographic coordinate shown in pixels over the time series. The dataset includes "wheat crop", "broad-leaved tree" and "urban" classes. With the given pixels changing over the time series, they can be used to generate land-cover maps with different classes.

**ElectricDevices [65]:**  This is a multi-class time-series dataset for detecting the type of appliance from their electricity usage patterns. The dataset was created from the data recorded as part of a UK government study *Powering the Nation*, conducted with the intention of collecting data about consumers' electricity use within the home to reduce the national carbon footprint. The dataset comprises of electricity readings from 251 households, taken over a month in 2-minute intervals.

**MIT-BIH [66]:**  The Massachusetts Institute of Technology - Beth Israel Hospital (MIT-BIH) dataset is a multi-class dataset comprising of electrocardiograms primarily used to evaluate automated arrhythmia detection algorithms [15]. It is collected from a mixed population of 47 in-patients and out-patients. The analog output of the playback unit was filtered using a bandpass of 0.1–100 Hz and digitized with 360 Hz. Each record is 30 min long and was annotated by a simple QRS detector with revisited domain expert annotations.

**PenDigits [67]:**  This is a multi-class time-series handwritten digit classification dataset. It was created by tracing the pen used by 44 writers to draw digits across a digital screen. Then, the authors re-sampled the data spatially to generate attributes having a constant spatial step and variable time step. The data was further re-sampled to 8 spatial points, where each instance is 2 dimensions of 8 points.

**WhaleCalls [68]:**  The WhaleCalls dataset is a binary class time-series classification dataset for evaluating whether an audio signal is a right whale's up-call. Up-calls are right whale vocalizations in the acoustic range of 60–250Hz. They are often difficult to hear due to increased congestion in the low-frequency band with anthropogenic sounds like piling, naval operations, or ship noise. Thus, detecting right whale up-calls is a critical task, since it further enables maritime navigation technologies.

## A.4   Classification Models Used in our Benchmark

The ultimate goal of label cleaning is to train accurate downstream classifiers, but different studies use different classification models to measure the efficacy of their proposed label cleaning methods. To provide a level playing field for all cleaning methods, we include at least two classification model architectures for each data modality. Specifically, we include ResNet-18 [8], MobileNet [9] and FastViT-T8 [69] for image datasets, `all-distilroberta-v1` [70, 71] and `all-MiniLM-L6-v2` [72] for text datasets, ResNet-1D, PatchTST [73] and LSTM Fully Convolutional Network [74] for time-series datasets, and TabTransformer [75] and a Multi-Layer Perceptron for tabular datasets. While choosing classification models we prioritized *performant* methods with (1) *different architectures* and *inductive biases*, (2) ideally *pre-trained* using different strategies, and (3) *previously-used* either by label cleaning methods or task-relevant papers. We do not use tree-based models in our experiments, even though they are easy to integrate into AQuA, since they are incompatible with some of the label error detection methods like AUM. We provide a brief descriptions of all classification models included in AQuA below.

**ResNet-18 [8]:**  ResNet is a commonly used computer vision architecture aimed at reducing the vanishing gradient problem in deep networks using jumping connections between layers and activating the previous layers. Our benchmark uses ResNet-18, which consists of 18 deep layers with a $7 \times 7$ kernel in the first layer, 4 identical ConvNet layers, and a fully connected layer with softmax activation. Each ConvNet layer has two blocks, each composed of two weight layers. Variants of ResNet are frequently used in the evaluation pipeline of popular label error detection models [23, 24].

**MobileNet [9]:**  MobileNet is a 53-layer deep convolutional neural network (CNN) used for mobile vision applications owing to its low computational intensity. It is implemented on the idea of depth-wise separable convolutions to create a light deep CNN having fewer parameters. Each depth-wise separable convolution is further composed of a depth-wise convolution and a point-wise convolution.

Thus, MobileNet consists of a total of 28 layers, when accounting for the depth-wise and point-wise layers. After each convolutional layer, batch normalization and ReLU activation are applied. We include MobileNet because it has been shown to be performant and light-weight, enabling us to speed up our experiments.

**FastViT-T8 [69]:** FastViT-T8 is a hybrid vision transformer model that achieves state-of-the-art accuracy-latency tradeoff. It is trained using a novel token mixing operator, RepMixer, that uses structural reparameterization for lowering memory access costs by eliminating skip-connections in the network. To reduce latency, FastViT replaces dense $kxk$ convolutions with their factorised versions. The FastViT-T8 model has an expansion ratio less than 4 and a total of 8 FastViT blocks. It consists of a total of 3.6M parameters. We include it in our experiments since it adds a different architecture for evaluation and achieves a good balance between computational cost and accuracy.

**DistilRoBERTa [71, 70]:** We use the `all-distilroberta-v1` model, which is a pre-trained `distilroberta-base` model, further fine-tuned on a 1 billion sentence pairs dataset using a self-supervised contrastive learning objective, where the model is tasked with predicting one sentence out of a randomly sampled set of sentences which can be paired with an input sentence. It was trained to map sentences and paragraphs into 768-dimensional vector space and can be further used for clustering and semantic search. all-distilroberta-v1's ancestor BERT and RoBERTa have been frequently used by studies in natural language processing and detecting labeling errors [39] alike.

**MiniLM-L6 [72]:** We also use the `all-MiniLM-L6-v2` model, which is a pre-trained `MiniLM-L6-H384-uncased` model, further fine-tuned on a 1 billion sentence pairs dataset using a self-supervised contrastive learning objective, where the model is tasked with predicting one sentence out of a randomly sampled set of sentences which can be paired with an input sentence. It was trained to map sentences and paragraphs into 384-dimensional vector space and can be further used for clustering and information retrieval applications. We included this model because it has a different inductive bias in comparison to all-distilroberta-v1 and is one of the fastest open-source pre-trained language models.

**Multi-layer Perceptron:** Multi-layer perceptron is a fully-connected multi-layer feed-forward connection of neurons, producing a set of output from a set of inputs. It typically consists of at least one hidden layer, which is any layer between the input and the output layer. Each layer consists of artificial neurons which apply activation function from the calculated sum from its inputs and forward it to the output. While it is frequently used for image classification, we apply it tabular data in our benchmark as a standard evaluation model to compare cleaning methods.

**TabTransformer [75]:** TabTransformer is a deep data modeling architecture for tabular data built upon self-attention based transformer architecture for supervised and semi-supervised learning. It transforms categorical features into contextualized embeddings, outperforming other deep networks for tabular data while matching the performance of tree-based ensemble methods. The contextualized embeddings enable interpretability compared to context-free embeddings from competing approaches and are robust against noisy and missing data.

**ResNet-1D [8]:** While the ResNet architecture has classically been used in computer vision tasks, one-dimensional convolutional neural networks have been shown to be state-of-the-art from time series classification [89]. In the healthcare domain, specifically in settings where there often are multiple channels of time series data, ResNet-1D can be implemented with channel attention to improve the model's learning efficiency from multi-feature channels.

**PatchTST [73]:** PatchTST is a transformer model designed for multivariate time-series forecasting. It has two key design elements: patching and channel-independence. During patching, we segment the time-series into sub-series to be fed into the transformer as tokens. This aids in local semantic information retention in the embeddings, reduced computation and memory usage for attention maps, and enables the model to learn a longer sequence. Channel independence refers to individual channels containing a univariate time series with the same embedding and transformer weights, and enables PatchTST to surpass the long-term forecasting accuracy compared to state-of-the-art time-series transformer-based models.

**Fully Convolutional Network [74]:** A fully convolutional network (FCN) is a deep learning architecture primarily consisting of convolutional layers, pooling, and upsampling, and is commonly used for semantic segmentation. Since it typically lacks a dense layer, it is quick to train. In an FCN, a 1x1 convolutional layer replaces the conventional fully-connected convolutional layer and dense layers. In particular, we use an LSTM-FCN to evaluate cleaning methods on time-series classification tasks. Like ResNet-1D, FCNs too have been shown to perform well for time-series classification problems [89].

## A.5 Hyperparameters and Hyperparameter Grids

We tuned hyper-parameters of all the classification and cleaning methods till they performed reasonably well on average on all the datasets using hyper-parameter grids in used by prior work and reported in Tables 5 and 6. During training, we reduce the learning rate by a factor of 10 if the loss does not improve for a "patience" number of epochs.

We deliberately did not perform extensive hyper-parameter tuning so as to not overfit to already existing label noise in the original datasets. Also, in practice it is unclear how to tune these cleaning methods well, without explicit knowledge of where the label errors are. We also did not tune hyper-parameters for downstream classifiers so that differences in their performance could be directly attributed to the cleaning methods, rather than differences in their own hyper-parameters.

In the case of SimiFeat and CINCER, we selected hyperparameter grids based on the parameters outlined in the original papers that introduced these methods. However, for AUM, we had to define the hyperparameter grid ourselves, as the authors did not provide specific recommendations in their publication. Notably, Confident Learning did not involve any hyperparameters as part of its configuration.

| Label Error Detection Method | Hyper-parameters | |
|---|---|---|
| AUM | alpha: | $\{\mathbf{0.01}, 0.05, 0.1, 0.15, 0.2\}$ |
| CINCER | threshold: | $\{0.05, 0.1, 0.15, 0.2, \mathbf{0.25}\}$ |
| | inspector: | $\{\mathbf{margin}\}$ |
| | negotiator: | $\{\mathbf{random}\}$ |
| | nfisher radius: | $\{\mathbf{0.1}\}$ |
| Confident Learning | | — |
| SimiFeat | max iter: | $\{600, \mathbf{1000}\}$ |
| | min similarity: | $\{\mathbf{0.45}, 0.5\}$ |
| | Tii offset: | $\{0.1, 1.0, \mathbf{2.5}\}$ |

Table 5: Hyper-parameter grids for label error detection models. The final hyper-parameters chosen for our experiments are in bold. The exhaustive set of hyperparameters for all downstream classification models can be found in `https://github.com/autonlab/aqua/tree/main/aqua/configs/models/cleaning`.

## A.6 Reproducibility and Replicability

**Data cards.** A data card is a CSV file for a given dataset, random seed, noise rate, and noise type, where rows and columns correspond to data points and predictions of cleaning methods, respectively. Each data card also has two additional columns for corrupted (i.e. the static copy) and original labels of data points. All the cleaning methods are evaluated on the same labeling errors. All the data cards from out experiments are uploaded here[7].

**Randomness.** We try to control all randomness in our experiments stemming from `PyTorch`, `random`, `numpy`, and CUDA. All our experiments are run with the random seed 42. For tabular data, we run two independent experiments with random seeds 42 and 43 for the multi-layer perception model.

**Hyper-parameter tuning.** For each cleaning method and downstream classification model, for a given dataset, hyper-parameters were chosen based on model performance on the observed training

---

[7]https://drive.google.com/drive/folders/1RHczHDUUilTOhcPyF5JSDvkO-rhiUKgb

| Model | Hyper-parameters | |
|---|---|---|
| ResNet-18 | `batch size:` | $\{64, 128, \mathbf{256}\}$ |
| | `epochs:` | $\{\mathbf{20}\}$ |
| | `learning rate:` | $\{0.005, \mathbf{0.01}, 0.1\}$ |
| | `momentum:` | $\{\mathbf{0.8}, 0.9\}$ |
| | `weight decay:` | $\{1e-5, \mathbf{1e-4}\}$ |
| MobileNet | `batch size:` | $\{64, 128, \mathbf{256}\}$ |
| | `epochs:` | $\{\mathbf{20}\}$ |
| | `learning rate:` | $\{0.005, \mathbf{0.01}, 0.1\}$ |
| | `momentum:` | $\{0.8, \mathbf{0.9}\}$ |
| | `weight decay:` | $\{1e-5, \mathbf{1e-4}\}$ |
| FastViT-T8 | `batch size:` | $\{64, \mathbf{128}, 256\}$ |
| | `epochs:` | $\{\mathbf{20}\}$ |
| | `learning rate:` | $\{0.005, \mathbf{0.01}, 0.1\}$ |
| | `momentum:` | $\{\mathbf{0.8}, 0.9\}$ |
| | `weight decay:` | $\{1e-5, \mathbf{1e-4}\}$ |
| DistilRoBERTa | `batch size:` | $\{\mathbf{64}, 128\}$ |
| | `epochs:` | $\{1, \mathbf{2}, 3\}$ |
| | `learning rate:` | $\{1e-5, 5e-5, \mathbf{1e-4}\}$ |
| MiniLM-L6 | `batch size:` | $\{\mathbf{64}, 128\}$ |
| | `epochs:` | $\{1, 2, \mathbf{3}\}$ |
| | `learning rate:` | $\{1e-5, 5e-5, \mathbf{1e-4}\}$ |
| Multi-layer Perceptron | `batch size:` | $\{\mathbf{64}\}$ |
| | `dropout rate:` | $\{\mathbf{0.0}, 0.1, 0.2\}$ |
| | `epochs:` | $\{\mathbf{15}, 30\}$ |
| | `learning rate:` | $\{\mathbf{0.001}, 0.005\}$ |
| TabTransformer | `batch size:` | $\{\mathbf{64}\}$ |
| | `momentum:` | $\{0.01, \mathbf{0.02}\}$ |
| | `epochs:` | $\{5, \mathbf{10}, 20\}$ |
| | `learning rate:` | $\{0.005, \mathbf{0.01}, 0.02\}$ |
| | `mask type:` | $\{\mathbf{sparsemax}\}$ |
| ResNet-1D | `batch size:` | $\{\mathbf{32}, 64, 128\}$ |
| | `epochs:` | $\{\mathbf{5}, 10\}$ |
| | `learning rate:` | $\{0.005, \mathbf{0.01}\}$ |
| Fully Convolutional Network | `batch size:` | $\{\mathbf{16}, 32, 64\}$ |
| | `epochs:` | $\{5, \mathbf{10}\}$ |
| | `learning rate:` | $\{0.005, \mathbf{0.01}\}$ |
| PatchTST | `batch size:` | $\{32, 64, \mathbf{128}\}$ |
| | `epochs:` | $\{10, 20, \mathbf{40}, 80\}$ |
| | `learning rate:` | $\{0.00005, \mathbf{0.0001}, 0.0002\}$ |
| | `patch length:` | $\{8, \mathbf{16}, 32\}$ |

Table 6: Hyper-parameter grids for downstream classification models. The final hyper-parameters chosen for our experiments are in **bold**. The exhaustive set of hyperparameters for all downstream classification models can be found in `https://github.com/autonlab/aqua/tree/main/aqua/configs/models/base`.

set, measured using weighted $F_1$ score. Once chosen, hyper-parameters were frozen for all noise experiments (noise type + noise rate). However, this evaluation setup has the following limitations:

- Tuning hyper-parameters based on the observed training set presents an advantage to the baseline method. In the ideal world, we should conduct extensive hyper-parameter tuning in each experiment setting, i.e. for each combination of dataset, noise rate, noise type, and cleaning method. However, that would be prohibitively expensive. Besides, we believe that insensitivity to hyper-parameters would be a hallmark of a good cleaning method.

- Tuning hyper-parameters based on a held-out validation set with no label errors prior to and after label cleaning. But this ideal scenario is contingent on a guaranteed error-free validation set and at least twice as much compute, which are prohibitive assumptions.

There were two primary reasons behind this design decision: (1) Our goal was to identify hyper-parameters that led to reasonable performance on the training set. Fine-grained tuning of hyper-parameters based on any dataset, whether held-out or in-domain, is tricky because the impact of label errors on model evaluation is hard to predict. We believe that evaluating model performance

in the presence of label noise is a hard but important research direction that warrants a dedicated study. (2) Furthermore, it may not be important to pick the "best" model that performs well on a held-out dataset, when in fact most if not all of the considered label cleaning methods utilize these downstream models (primarily trained on the training set) to learn representations of training data points. Once erroneous labels are identified, they are removed and the same model is re-trained on the "cleaned" training data, and their performance is measured on the test data.

## A.7   Synthetic Label Noise

To enable a realistic, multi-faceted and holistic evaluation of label error detection models, we implement **7** popular label noise injection techniques and multiple metrics of predictive performance. Specifically, for single-label datasets, we implement asymmetric [34], class-dependent [76], instance-dependent [33], and uniform [76] noise, and for datasets with labels from multiple annotators, we implement dissenting label, dissenting worker, and crowd majority [39].

**Uniform Noise [76]:**   For this type of noise, each entry in the noise transition matrix, except the diagonal ones, is equal. Specifically, for a noise rate $p \in [0, 1]$,

$$\mathbf{T}_{ij} = \begin{cases} 1 - p, & i = j \\ \frac{p}{M-1}, & \text{otherwise} \end{cases}$$

**Class-dependent Noise [76]:**   In this setting, similar classes have a higher probability of being mislabeled with each other. For any given dataset, we define the noise transition matrix as the confusion matrix derived from of a model that has been trained and evaluated on the dataset's training set.

**Asymmetric Label Noise [34]:**   We generate asymmetric noise by pair-wise flipping, i.e., for dataset with $K$ classes, we randomly flip the observed label $i$ to the next class $(i + 1) \bmod K$.

**Instance-dependent Label Noise [41]:**   Unlike the previous settings, instance-dependent noise depends both on the data features and class labels to introduce realistic noise into a dataset. We follow Algorithm 2 in [41] to generate instance-dependent label noise.

We also implement three kinds of label noise for datasets which comprise of labels from multiple annotators following Chong et al. [39].

**Dissenting Label :**   This approach randomly replaces the final labels with disagreeing labels to simulate a situation of imperfect quality control.

**Dissenting Worker [39]:**   The dissenting worker approach simulates gaps in annotator training by randomly selecting an annotator and replacing the final labels with labels from the given annotator which do not match the final labels. This process is repeated for different annotators till the required noise rate is achieved.

**Crowd Majority [39]:**   The crowd majority approach can introduce systematic errors into a dataset by aggregating all individual annotations to produce a label other than the final label.

## A.8 Additional Results

### A.8.1 Performance of Cleaning Methods Across Different Synthetic Noise Types

| Datasets | Uniform | | | | Asymmetric | | | | Class-dependent | | | | Instance-dependent | | | |
|---|---|---|---|---|---|---|---|---|---|---|---|---|---|---|---|---|
| | AUM | CIN | CON | SIM | AUM | CIN | CON | SIM | AUM | CIN | CON | SIM | AUM | CIN | CON | SIM |
| CIFAR-10 | 84.8 | 80.7 | 17.9 | 89.1 | 85.0 | 80.1 | 18.0 | 88.8 | 98.9 | 88.2 | 18.1 | 97.2 | 78.5 | 77.3 | 18.8 | 83.3 |
| Clothing-100K | 85.2 | 84.2 | 80.6 | 85.4 | 84.9 | 84.0 | 94.3 | 85.2 | 92.4 | 91.2 | 70.2 | 92.7 | 77.9 | 74.0 | 79.1 | 78.0 |
| NoisyCXR | 85.0 | 79.2 | 13.2 | 85.2 | 85.0 | 79.3 | 14.5 | 85.2 | 99.5 | 86.5 | 14.1 | 100 | 78.6 | 76.9 | 16.5 | 78.7 |
| IMDb | 84.8 | 90.8 | 59.7 | 91.6 | 85.0 | 91.2 | 66.1 | 91.8 | 93.4 | 94.8 | 63.4 | 96.3 | 78.6 | 87.3 | 56.9 | 86.2 |
| TweetEval | 85.6 | 86.7 | 62.6 | 87.4 | 84.9 | 86.6 | 56.8 | 87.3 | 70.2 | 70.9 | 63.7 | 71.8 | 78.5 | 78.9 | 59.9 | 81.3 |
| Credit Fraud | 85.0 | 85.3 | 81.5 | 96.1 | 85.0 | 85.2 | 97.7 | 92.3 | 76.6 | 76.6 | 88.5 | 92.9 | 78.6 | 78.7 | 93.4 | 94.5 |
| Adult | 85.0 | 86.1 | 64.2 | 90.7 | 85.0 | 86.3 | 54.9 | 87.8 | 60.6 | 61.1 | 57.7 | 62.8 | 78.3 | 80.3 | 70.4 | 78.4 |
| Dry Bean | 84.8 | 94.6 | 32.0 | 91.8 | 85.1 | 93.7 | 26.3 | 90.6 | 85.8 | 94.4 | 30.6 | 91.9 | 78.5 | 94.0 | 28.1 | 82.2 |
| Car Evaluation | 84.7 | 87.3 | 77.4 | 88.5 | 84.3 | 87.0 | 81.4 | 92.0 | 88.2 | 91.3 | 83.9 | 90.8 | 77.5 | 84.7 | 78.1 | 88.6 |
| Mushrooms | 84.1 | 93.4 | 59.9 | 93.2 | 85.0 | 93.5 | 60.5 | 93.9 | 98.8 | 99.9 | 65.2 | 99.9 | 78.2 | 90.4 | 57.1 | 78.2 |
| COMPAS | 84.9 | 84.9 | 60.4 | 84.4 | 85.0 | 85.9 | 57.0 | 85.2 | 55.7 | 55.7 | 52.7 | 55.6 | 77.9 | 79.6 | 57.0 | 78.1 |
| Crop | 85.3 | 79.8 | 14.2 | 88.6 | 85.2 | 69.0 | 13.2 | 87.4 | 46.5 | 60.3 | 27.9 | 65.1 | 79.2 | 65.5 | 14.9 | 77.8 |
| Electric Devices | 85.4 | 90.4 | 21.4 | 91.9 | 84.9 | 88.0 | 39.3 | 89.3 | 75.9 | 83.2 | 32.1 | 82.1 | 78.9 | 88.0 | 33.4 | 90.7 |
| MIT-BIH | 84.6 | 93.2 | 38.7 | 92.8 | 85.0 | 93.4 | 31.4 | 90.2 | 73.5 | 67.8 | 38.6 | 88.3 | 78.8 | 89.1 | 36.5 | 84.8 |
| PenDigits | 84.4 | 97.3 | 19.5 | 93.6 | 84.9 | 97.1 | 19.6 | 93.5 | 98.2 | 97.6 | 19.3 | 99.0 | 78.9 | 93.7 | 20.2 | 84.0 |
| WhaleCalls | 84.3 | 84.7 | 59.8 | 88.8 | 85.1 | 85.2 | 59.9 | 88.9 | 34.9 | 34.1 | 50.1 | 40.9 | 78.7 | 78.8 | 57.1 | 84.3 |

Table 7: Performance evaluation of cleaning methods to detect erroneous labels across different types of synthetic noise added to the train set in terms of weighted $F_1$, for noise rate $= 0.1$. The classification models used for images, text, tabular, and time series datasets are ResNet-18, `all-distilroberta-v1`, Multi-layer perception (random seed 42), and ResNet-1D, respectively.

| Datasets | Uniform | | | | Asymmetric | | | | Class-dependent | | | | Instance-dependent | | | |
|---|---|---|---|---|---|---|---|---|---|---|---|---|---|---|---|---|
| | AUM | CIN | CON | SIM | AUM | CIN | CON | SIM | AUM | CIN | CON | SIM | AUM | CIN | CON | SIM |
| CIFAR-10 | 84.8 | 81.3 | 18.2 | 88.9 | 84.9 | 83.2 | 17.8 | 88.4 | 90.5 | 84.0 | 17.6 | 91.6 | 78.4 | 80.8 | 18.7 | 80.0 |
| Clothing-100K | 85.2 | 83.1 | 94.4 | 85.4 | 84.9 | 83.0 | 94.3 | 85.2 | 84.8 | 78.1 | 94.2 | 85.0 | 77.9 | 68.7 | 74.4 | 78.0 |
| NoisyCXR | 85.0 | 78.3 | 13.8 | 85.2 | 84.9 | 77.0 | 15.1 | 85.2 | 90.9 | 80.8 | 16.3 | 91.6 | 78.6 | 75.0 | 15.2 | 79.0 |
| IMDb | 84.8 | 80.3 | 59.5 | 90.4 | 84.9 | 81.9 | 59.6 | 90.8 | 88.0 | 83.9 | 60.8 | 92.5 | 79.0 | 81.8 | 57.3 | 85.7 |
| TweetEval | 85.6 | 86.2 | 56.7 | 86.6 | 85.1 | 85.9 | 62.6 | 86.3 | 73.0 | 73.6 | 46.2 | 75.0 | 78.6 | 78.7 | 60.0 | 81.8 |
| Credit Fraud | 85.1 | 85.4 | 92.8 | 91.1 | 85.0 | 85.2 | 67.9 | 85.2 | 79.2 | 79.3 | 93.6 | 94.4 | 78.5 | 78.6 | 65.8 | 86.5 |
| Adult | 84.7 | 85.9 | 83.6 | 85.0 | 85.0 | 86.4 | 74.5 | 88.6 | 64.4 | 65.6 | 58.3 | 64.2 | 78.5 | 80.4 | 70.5 | 85.6 |
| Dry Bean | 84.7 | 95.2 | 46.7 | 91.8 | 84.9 | 95.8 | 39.3 | 85.8 | 87.3 | 94.5 | 38.6 | 88.6 | 78.2 | 92.6 | 40.6 | 83.6 |
| Car Evaluation | 84.4 | 88.5 | 79.6 | 92.0 | 84.8 | 87.7 | 78.6 | 92.3 | 83.0 | 92.2 | 84.3 | 85.5 | 77.0 | 84.4 | 74.8 | 82.2 |
| Mushrooms | 85.5 | 94.0 | 60.0 | 95.4 | 84.8 | 93.8 | 66.6 | 94.1 | 99.4 | 99.9 | 75.0 | 99.6 | 77.7 | 90.5 | 57.7 | 77.7 |
| COMPAS | 84.9 | 85.7 | 66.1 | 83.8 | 84.9 | 85.3 | 77.6 | 83.9 | 55.5 | 55.0 | 53.7 | 55.0 | 77.6 | 79.5 | 57.8 | 77.5 |
| Crop | 84.7 | 86.8 | 15.4 | 91.1 | 85.0 | 78.6 | 12.3 | 88.0 | 35.1 | 61.8 | 37.8 | 56.2 | 78.8 | 77.8 | 15.7 | 81.9 |
| Electric Devices | 84.5 | 86.3 | 35.0 | 91.3 | 85.0 | 84.6 | 39.3 | 89.1 | 6.7 | 52.1 | 70.0 | 47.6 | 79.3 | 82.7 | 39.2 | 83.2 |
| MIT-BIH | 84.8 | 94.2 | 66.8 | 91.9 | 85.0 | 94.2 | 70.1 | 87.8 | 37.5 | 66.7 | 52.4 | 62.3 | 78.8 | 92.2 | 65.0 | 83.1 |
| PenDigits | 84.5 | 97.9 | 18.9 | 93.8 | 84.9 | 98.1 | 20.2 | 94.0 | 5.5 | 4.4 | 80.1 | 58.7 | 77.8 | 96.3 | 20.1 | 81.3 |
| WhaleCalls | 84.9 | 80.6 | 64.9 | 86.7 | 85.0 | 82.1 | 66.5 | 86.9 | 33.8 | 43.2 | 51.4 | 37.0 | 77.7 | 74.7 | 61.7 | 81.1 |

Table 8: Performance evaluation of cleaning methods to detect erroneous labels across different types of synthetic noise added to the train set in terms of weighted $F_1$, for noise rate $= 0.1$. The classification models used for images, text, tabular, and time series datasets are MobileNet-v2, `all-MiniLM-L6-v2`, Multi-layer perception (random seed 43), and Fully convolutional network, respectively.

| Datasets | Uniform | | | | Asymmetric | | | | Class-dependent | | | | Instance-dependent | | | |
|---|---|---|---|---|---|---|---|---|---|---|---|---|---|---|---|---|
| | AUM | CIN | CON | SIM | AUM | CIN | CON | SIM | AUM | CIN | CON | SIM | AUM | CIN | CON | SIM |
| CIFAR-10 | 85.0 | 64.2 | 85.1 | 85.1 | 85.0 | 66.0 | 85.3 | 85.3 | 91.8 | 69.8 | 92.2 | 92.2 | 78.6 | 62.5 | 78.8 | 78.8 |
| Clothing-100K | 85.3 | 85.0 | 85.5 | 89.6 | 85.1 | 61.0 | 85.3 | 89.6 | 49.2 | 41.5 | 48.8 | 63.5 | 78.4 | 77.9 | 78.6 | 84.3 |
| NoisyCXR | 85.1 | 71.5 | 85.4 | 86.4 | 85.1 | 71.1 | 85.3 | 85.7 | 63.6 | 68.6 | 63.4 | 73.5 | 78.6 | 66.3 | 78.7 | 81.1 |
| Credit Fraud | 85.0 | 85.2 | 97.7 | 85.2 | 85.0 | 85.3 | 95.4 | 85.4 | 34.8 | 33.9 | 79.9 | 33.9 | 78.7 | 78.8 | 65.9 | 78.8 |
| Adult | 85.0 | 86.0 | 85.3 | 87.0 | 85.0 | 85.9 | 85.3 | 87.2 | 64.6 | 65.6 | 64.4 | 67.2 | 78.7 | 79.9 | 78.7 | 81.3 |
| Dry Bean | 85.1 | 94.7 | 22.4 | 92.0 | 85.0 | 94.2 | 25.0 | 90.2 | 86.7 | 95.3 | 32.4 | 92.5 | 78.3 | 93.0 | 27.8 | 83.6 |
| Car Evaluation | 85.2 | 94.7 | 82.4 | 87.5 | 84.9 | 95.3 | 78.3 | 88.3 | 69.7 | 82.7 | 83.0 | 73.6 | 79.7 | 93.7 | 77.9 | 89.1 |
| Mushrooms | 84.9 | 93.7 | 67.5 | 93.9 | 85.3 | 85.3 | 85.3 | 85.3 | 99.4 | 100 | 66.4 | 99.7 | 79.4 | 91.1 | 65.1 | 89.0 |
| COMPAS | 85.8 | 86.5 | 60.8 | 85.8 | 85.1 | 85.8 | 60.4 | 85.1 | 55.2 | 57.4 | 50.9 | 60.0 | 79.1 | 81.0 | 59.1 | 79.2 |
| Crop | 85.0 | 85.5 | 8.9 | 53.2 | 85.1 | 85.3 | 9.2 | 53.2 | 3.1 | 0.4 | 91.6 | 67.6 | 78.2 | 78.4 | 9.1 | 43.0 |
| Electric Devices | 84.9 | 85.1 | 34.6 | 67.1 | 85.0 | 85.3 | 34.6 | 64.5 | 6.0 | 3.5 | 77.2 | 61.1 | 78.4 | 78.4 | 33.9 | 57.5 |
| MIT-BIH | 85.2 | 88.9 | 45.2 | 85.5 | 85.1 | 87.7 | 53.7 | 85.3 | 82.5 | 81.7 | 40.4 | 82.7 | 78.7 | 84.9 | 44.1 | 78.8 |
| PenDigits | 85.1 | 85.3 | 17.9 | 54.4 | 85.1 | 85.3 | 17.8 | 56.1 | 7.0 | 1.7 | 82.2 | 64.4 | 79.1 | 79.0 | 20.0 | 48.0 |
| WhaleCalls | 84.2 | 84.3 | 59.2 | 81.6 | 85.1 | 85.3 | 59.3 | 83.9 | 35.0 | 34.1 | 50.2 | 40.3 | 78.6 | 78.8 | 57.2 | 78.0 |

Table 9: Performance evaluation of cleaning methods to detect erroneous labels across different types of synthetic noise added to the train set in terms of weighted $F_1$, for noise rate $= 0.1$. The classification models used for images, tabular, and time series datasets are Fast-ViT-T8, TabTransformer, and PatchTST, respectively.

| Datasets | Uniform | | | | Asymmetric | | | | Class-dependent | | | | Instance-dependent | | | |
|---|---|---|---|---|---|---|---|---|---|---|---|---|---|---|---|---|
| | AUM | CIN | CON | SIM | AUM | CIN | CON | SIM | AUM | CIN | CON | SIM | AUM | CIN | CON | SIM |
| CIFAR-10 | 45.9 | 67.2 | 32.1 | 47.3 | 45.8 | 59.2 | 32.6 | 48.5 | 98.9 | 88.2 | 18.1 | 97.2 | 40.4 | 59.4 | 35.0 | 48.0 |
| Clothing-100K | 45.2 | 36.0 | 59.1 | 44.7 | 45.5 | 36.2 | 24.9 | 45.0 | 92.4 | 91.2 | 70.2 | 92.7 | 40.2 | 31.0 | 58.6 | 39.6 |
| NoisyCXR | 45.7 | 70.8 | 31.6 | 45.1 | 45.7 | 58.1 | 30.8 | 45.0 | 99.5 | 86.5 | 14.1 | 100 | 39.8 | 62.8 | 33.8 | 39.0 |
| IMDb | 45.6 | 44.9 | 50.6 | 47.6 | 45.6 | 45.0 | 50.7 | 47.6 | 93.4 | 94.8 | 63.4 | 96.3 | 40.5 | 39.6 | 50.7 | 43.7 |
| TweetEval | 44.2 | 43.4 | 47.3 | 49.7 | 45.8 | 45.0 | 46.9 | 53.1 | 70.2 | 70.9 | 63.7 | 71.8 | 39.5 | 39.0 | 53.4 | 51.8 |
| Credit Fraud | 45.6 | 45.1 | 40.6 | 45.0 | 45.6 | 45.0 | 60.3 | 45.0 | 76.6 | 76.6 | 88.5 | 92.9 | 40.0 | 39.3 | 50.7 | 39.3 |
| Adult | 45.5 | 45.3 | 55.6 | 45.0 | 45.7 | 45.0 | 45.2 | 45.0 | 60.6 | 61.1 | 57.7 | 62.8 | 40.0 | 39.7 | 55.1 | 39.4 |
| Dry Bean | 45.4 | 80.2 | 35.6 | 50.9 | 45.5 | 68.9 | 38.6 | 50.6 | 85.8 | 94.4 | 30.6 | 91.9 | 41.2 | 64.5 | 32.4 | 51.1 |
| Car Evaluation | 43.8 | 74.8 | 78.4 | 80.3 | 45.6 | 61.0 | 72.1 | 68.9 | 88.2 | 91.3 | 83.9 | 90.8 | 41.7 | 58.4 | 71.4 | 51.4 |
| Mushrooms | 46.7 | 55.2 | 51.2 | 67.6 | 45.5 | 55.6 | 50.1 | 45.0 | 98.8 | 99.9 | 65.2 | 99.9 | 39.9 | 39.6 | 49.4 | 45.1 |
| COMPAS | 46.7 | 46.2 | 52.3 | 46.2 | 45.7 | 45.0 | 53.2 | 51.0 | 55.7 | 55.7 | 52.7 | 55.6 | 40.1 | 39.5 | 53.1 | 53.8 |
| Crop | 46.9 | 63.7 | 27.5 | 64.7 | 45.3 | 56.6 | 29.5 | 52.9 | 46.5 | 60.3 | 27.9 | 65.1 | 39.2 | 41.4 | 34.4 | 59.2 |
| Electric Devices | 46.5 | 78.4 | 36.1 | 69.0 | 45.3 | 62.2 | 36.1 | 61.0 | 75.9 | 83.2 | 32.1 | 82.1 | 40.8 | 39.6 | 19.1 | 60.7 |
| MIT-BIH | 45.8 | 78.1 | 40.3 | 80.7 | 45.5 | 66.9 | 42.9 | 55.3 | 73.5 | 67.8 | 38.6 | 88.3 | 40.7 | 46.2 | 47.0 | 52.0 |
| PenDigits | 46.5 | 91.2 | 33.0 | 56.1 | 45.4 | 63.9 | 33.3 | 53.0 | 98.2 | 97.6 | 19.3 | 99.0 | 40.9 | 59.0 | 24.1 | 66.5 |
| WhaleCalls | 45.5 | 45.0 | 51.3 | 55.3 | 45.4 | 45.0 | 50.6 | 50.1 | 34.9 | 34.1 | 50.1 | 40.9 | 40.4 | 39.4 | 49.4 | 39.5 |

Table 10: Performance evaluation of cleaning methods to detect erroneous labels across different types of synthetic noise added to the train set in terms of weighted $F_1$, for noise rate $= 0.4$. The classification models used for images, text, tabular, and time series datasets are ResNet-18, `all-distilroberta-v1`, Multi-layer perception (random seed 42), and ResNet-1D, respectively.

| Datasets | Uniform | | | | Asymmetric | | | | Class-dependent | | | | Instance-dependent | | | |
|---|---|---|---|---|---|---|---|---|---|---|---|---|---|---|---|---|
| | AUM | CIN | CON | SIM | AUM | CIN | CON | SIM | AUM | CIN | CON | SIM | AUM | CIN | CON | SIM |
| CIFAR-10 | 46.0 | 72.3 | 32.2 | 63.7 | 45.7 | 60.8 | 32.4 | 48.7 | 90.5 | 84.0 | 17.6 | 91.6 | 40.0 | 64.9 | 34.9 | 47.8 |
| Clothing-100K | 45.2 | 33.5 | 78.4 | 44.7 | 45.5 | 39.1 | 69.7 | 45.0 | 84.8 | 78.1 | 94.2 | 85.0 | 40.2 | 35.6 | 82.7 | 39.6 |
| NoisyCXR | 45.7 | 68.5 | 31.4 | 45.1 | 45.6 | 57.0 | 28.8 | 45.2 | 90.9 | 80.8 | 16.3 | 91.6 | 40.4 | 59.9 | 33.5 | 41.2 |
| IMDb | 45.6 | 44.9 | 50.4 | 47.8 | 45.7 | 45.0 | 50.8 | 48.6 | 88.0 | 83.9 | 60.8 | 92.5 | 40.2 | 39.5 | 49.9 | 42.7 |
| TweetEval | 44.2 | 43.4 | 40.7 | 53.1 | 45.6 | 45.0 | 54.3 | 48.2 | 73.0 | 73.6 | 46.2 | 75.0 | 40.8 | 40.0 | 53.8 | 44.1 |
| Credit Fraud | 45.7 | 45.1 | 20.6 | 45.1 | 45.6 | 45.0 | 52.0 | 45.0 | 79.2 | 79.3 | 93.6 | 94.4 | 40.1 | 39.4 | 60.0 | 39.4 |
| Adult | 45.9 | 46.1 | 45.4 | 45.3 | 45.5 | 45.5 | 61.9 | 54.8 | 64.4 | 65.6 | 58.3 | 64.2 | 40.2 | 39.8 | 55.2 | 39.5 |
| Dry Bean | 45.3 | 82.8 | 45.8 | 61.3 | 45.6 | 61.9 | 44.6 | 64.8 | 87.3 | 94.5 | 38.6 | 88.6 | 40.1 | 56.4 | 28.2 | 63.2 |
| Car Evaluation | 43.6 | 73.3 | 77.7 | 73.8 | 45.9 | 58.9 | 76.5 | 61.1 | 83.0 | 92.2 | 84.3 | 85.5 | 41.8 | 56.6 | 72.9 | 51.6 |
| Mushrooms | 46.3 | 55.4 | 49.5 | 55.8 | 45.8 | 51.8 | 50.8 | 68.1 | 99.4 | 99.9 | 75.0 | 99.6 | 39.8 | 39.2 | 50.0 | 49.4 |
| COMPAS | 44.6 | 44.0 | 52.2 | 44.0 | 45.5 | 45.2 | 51.0 | 45.0 | 55.5 | 55.0 | 53.7 | 55.0 | 39.9 | 39.0 | 50.4 | 51.7 |
| Crop | 46.0 | 75.2 | 29.5 | 64.8 | 46.3 | 57.8 | 29.9 | 52.8 | 35.1 | 61.8 | 37.8 | 56.2 | 40.2 | 45.9 | 35.0 | 56.7 |
| Electric Devices | 45.4 | 76.0 | 40.3 | 72.1 | 45.4 | 62.2 | 40.2 | 59.2 | 6.7 | 52.1 | 70.0 | 47.6 | 41.3 | 48.9 | 34.8 | 52.7 |
| MIT-BIH | 44.8 | 80.6 | 54.1 | 62.0 | 45.5 | 57.8 | 49.9 | 48.9 | 37.5 | 66.7 | 52.4 | 62.3 | 40.4 | 54.6 | 41.6 | 64.1 |
| PenDigits | 46.7 | 92.4 | 33.0 | 56.7 | 45.7 | 66.6 | 33.3 | 53.6 | 5.5 | 4.4 | 80.1 | 58.7 | 40.7 | 62.9 | 29.8 | 65.9 |
| WhaleCalls | 46.3 | 52.5 | 54.2 | 50.2 | 45.4 | 52.7 | 54.3 | 51.0 | 33.8 | 43.2 | 51.4 | 37.0 | 39.8 | 48.3 | 49.8 | 43.2 |

Table 11: Performance evaluation of cleaning methods to detect erroneous labels across different types of synthetic noise added to the train set in terms of weighted $F_1$, for noise rate $= 0.4$. The classification models used for images, text, tabular, and time series datasets are MobileNet-v2, `all-MiniLM-L6-v2`, Multi-layer perception (random seed 43), and Fully convolutional network, respectively.

| Datasets | Uniform | | | | Asymmetric | | | | Class-dependent | | | | Instance-dependent | | | |
|---|---|---|---|---|---|---|---|---|---|---|---|---|---|---|---|---|
| | AUM | CIN | CON | SIM | AUM | CIN | CON | SIM | AUM | CIN | CON | SIM | AUM | CIN | CON | SIM |
| CIFAR-10 | 45.4 | 56.2 | 44.6 | 44.6 | 45.7 | 52.4 | 45.0 | 45.0 | 91.8 | 69.8 | 92.2 | 92.2 | 40.6 | 52.7 | 39.7 | 39.7 |
| Clothing-100K | 45.4 | 35.4 | 44.7 | 61.0 | 45.6 | 38.2 | 41.5 | 62.3 | 49.2 | 41.5 | 48.8 | 63.5 | 41.1 | 28.9 | 40.6 | 53.7 |
| NoisyCXR | 45.1 | 60.9 | 44.6 | 58.3 | 45.6 | 56.8 | 45.0 | 54.8 | 63.6 | 68.6 | 63.4 | 73.5 | 40.2 | 53.7 | 39.5 | 52.3 |
| Credit Fraud | 45.9 | 45.2 | 40.6 | 45.2 | 45.6 | 45.0 | 60.4 | 45.0 | 34.8 | 33.9 | 79.9 | 33.9 | 40.0 | 39.2 | 40.2 | 39.2 |
| Adult | 45.7 | 45.0 | 45.0 | 64.7 | 45.8 | 45.6 | 45.0 | 45.2 | 64.6 | 65.6 | 64.4 | 67.2 | 40.8 | 40.0 | 40.0 | 53.7 |
| Dry Bean | 46.2 | 87.6 | 39.8 | 59.9 | 45.7 | 60.8 | 35.9 | 48.9 | 86.7 | 95.3 | 32.4 | 92.5 | 39.4 | 61.4 | 42.0 | 55.3 |
| Car Evaluation | 46.8 | 46.8 | 46.8 | 46.8 | 45.2 | 45.2 | 45.2 | 45.2 | 69.7 | 82.7 | 83.0 | 73.6 | 41.9 | 56.1 | 82.6 | 51.3 |
| Mushrooms | 46.4 | 55.0 | 49.8 | 63.9 | 45.6 | 50.4 | 52.3 | 62.8 | 99.4 | 100 | 66.4 | 99.7 | 40.7 | 39.9 | 50.1 | 39.9 |
| COMPAS | 45.1 | 45.4 | 44.7 | 52.5 | 45.5 | 45.0 | 49.1 | 53.2 | 55.2 | 57.4 | 50.9 | 60.0 | 40.6 | 39.8 | 54.0 | 49.7 |
| Crop | 46.0 | 44.9 | 27.1 | 49.1 | 45.5 | 45.0 | 27.2 | 50.2 | 3.1 | 0.4 | 91.6 | 67.6 | 39.9 | 4.2 | 11.4 | 23.1 |
| Electric Devices | 45.5 | 45.0 | 41.5 | 56.3 | 45.4 | 45.0 | 35.9 | 53.6 | 6.0 | 3.5 | 77.2 | 61.1 | 39.7 | 38.9 | 26.9 | 26.4 |
| MIT-BIH | 45.3 | 70.8 | 47.5 | 44.8 | 45.7 | 56.6 | 37.9 | 45.0 | 82.5 | 81.7 | 40.4 | 82.7 | 39.3 | 57.2 | 43.2 | 38.7 |
| PenDigits | 46.1 | 44.9 | 32.3 | 49.0 | 45.5 | 45.0 | 33.1 | 48.6 | 7.0 | 1.7 | 82.2 | 64.4 | 40.6 | 10.5 | 10.5 | 25.4 |
| WhaleCalls | 45.8 | 45.1 | 50.4 | 47.4 | 45.6 | 45.0 | 50.6 | 47.2 | 35.0 | 34.1 | 50.2 | 40.3 | 40.6 | 39.9 | 50.3 | 43.1 |

Table 12: Performance evaluation of cleaning methods to detect erroneous labels across different types of synthetic noise added to the train set in terms of weighted $F_1$, for noise rate $= 0.4$. The classification models used for images, tabular, and time series datasets are Fast-ViT-T8, TabTransformer, and PatchTST, respectively.

### A.8.2  Impact of Label Noise on Weighted $F_1$ Score

| Datasets | Uniform | | | | Asymmetric | | | | Class-dependent | | | | Instance-dependent | | | |
|---|---|---|---|---|---|---|---|---|---|---|---|---|---|---|---|---|
| | AUM | CIN | CON | SIM | AUM | CIN | CON | SIM | AUM | CIN | CON | SIM | AUM | CIN | CON | SIM |
| CIFAR-10 | 96.5 | 86.9 | 17.9 | 96.5 | 96.5 | 86.8 | 17.9 | 96.6 | 96.6 | 87.1 | 17.8 | 96.2 | 90.0 | 82.9 | 17.5 | 92.7 |
| Clothing-100K | 96.8 | 96.7 | 85.0 | 97.3 | 96.5 | 96.5 | 84.9 | 97.0 | 99.5 | 99.7 | 96.8 | 100 | 90.0 | 89.5 | 94.7 | 90.4 |
| NoisyCXR | 96.4 | 84.8 | 12.8 | 96.9 | 96.5 | 84.3 | 15.9 | 97.0 | 99.4 | 86.6 | 15.6 | 100 | 89.7 | 81.0 | 15.2 | 90.1 |
| IMDb | 96.5 | 97.5 | 65.0 | 96.9 | 96.5 | 96.6 | 64.9 | 96.9 | 95.8 | 95.5 | 64.5 | 96.6 | 89.8 | 92.8 | 61.4 | 94.8 |
| TweetEval | 96.2 | 94.6 | 77.9 | 94.9 | 96.6 | 96.6 | 61.9 | 95.2 | 72.0 | 72.2 | 52.1 | 71.9 | 90.2 | 90.6 | 64.9 | 90.3 |
| Credit Fraud | 96.5 | 97.0 | 99.0 | 99.2 | 96.6 | 97.0 | 99.0 | 97.3 | 76.2 | 76.3 | 88.3 | 85.1 | 89.9 | 90.2 | 96.9 | 94.5 |
| Adult | 96.5 | 95.9 | 90.1 | 95.8 | 96.6 | 95.9 | 77.4 | 95.8 | 63.9 | 64.9 | 57.9 | 63.6 | 90.1 | 90.8 | 74.2 | 90.3 |
| Dry Bean | 96.3 | 96.4 | 63.3 | 97.3 | 96.3 | 96.3 | 56.0 | 97.3 | 88.9 | 95.8 | 42.7 | 95.0 | 89.6 | 95.7 | 54.7 | 91.5 |
| Car Evaluation | 95.0 | 91.3 | 89.7 | 95.0 | 96.7 | 89.5 | 84.8 | 94.6 | 65.7 | 85.9 | 81.6 | 75.3 | 89.5 | 88.1 | 81.9 | 93.3 |
| Mushrooms | 96.9 | 98.8 | 88.0 | 99.9 | 96.5 | 98.6 | 68.4 | 100 | 99.5 | 100 | 100 | 100 | 90.1 | 95.8 | 86.7 | 93.0 |
| COMPAS | 96.2 | 93.4 | 75.7 | 94.9 | 96.7 | 93.6 | 75.8 | 95.3 | 55.9 | 60.8 | 53.7 | 60.5 | 90.3 | 88.8 | 72.3 | 89.7 |
| Crop | 96.8 | 89.4 | 7.8 | 94.4 | 96.5 | 86.3 | 7.9 | 94.3 | 35.1 | 49.4 | 34.5 | 57.0 | 90.0 | 78.5 | 8.3 | 88.8 |
| Electric Devices | 96.6 | 90.5 | 33.7 | 95.8 | 96.5 | 91.7 | 15.2 | 96.8 | 73.5 | 84.4 | 33.0 | 83.6 | 89.6 | 87.4 | 35.5 | 88.8 |
| MIT-BIH | 96.7 | 97.3 | 39.4 | 97.5 | 96.4 | 97.2 | 52.7 | 98.2 | 70.8 | 84.5 | 40.6 | 86.7 | 89.7 | 94.4 | 47.7 | 90.7 |
| PenDigits | 96.6 | 98.0 | 17.9 | 98.5 | 96.6 | 97.2 | 17.4 | 98.5 | 97.3 | 97.4 | 18.4 | 98.8 | 90.2 | 96.8 | 18.0 | 92.8 |
| WhaleCalls | 96.2 | 96.7 | 64.9 | 95.8 | 96.5 | 97.0 | 73.6 | 95.9 | 46.9 | 55.0 | 50.2 | 56.0 | 90.0 | 90.3 | 69.0 | 92.5 |

Table 13: Performance evaluation of cleaning methods to detect erroneous labels across different types of synthetic noise added to the train set in terms of weighted $F_1$, for noise rate $= 0.02$. The classification models used for images, text, tabular, and time series datasets are ResNet-18, `all-distilroberta-v1`, Multi-layer perception (random seed 42), and ResNet-1D, respectively.

| Datasets | Uniform | | | | Asymmetric | | | | Class-dependent | | | | Instance-dependent | | | |
|---|---|---|---|---|---|---|---|---|---|---|---|---|---|---|---|---|
| | AUM | CIN | CON | SIM | AUM | CIN | CON | SIM | AUM | CIN | CON | SIM | AUM | CIN | CON | SIM |
| CIFAR-10 | 74.8 | 85.4 | 65.1 | 77.9 | 83.8 | 76.2 | 83.4 | 81.5 | 90.6 | 84.0 | 17.6 | 91.6 | 74.7 | 79.8 | 69.3 | 77.4 |
| Clothing-100K | 89.9 | 79.0 | 64.6 | 83.1 | 82.8 | 80.1 | 67.1 | 74.7 | 84.9 | 78.1 | 94.2 | 85.0 | 88.2 | 98.3 | 56.3 | 87.0 |
| NoisyCXR | 91.6 | 81.4 | 58.5 | 73.0 | 78.3 | 85.3 | 22.8 | 68.9 | 90.9 | 80.9 | 16.3 | 91.7 | 75.8 | 80.9 | 66.8 | 97.8 |
| IMDb | 96.5 | 81.4 | 65.0 | 96.4 | 96.5 | 86.2 | 64.9 | 96.3 | 94.0 | 93.1 | 63.5 | 95.4 | 90.1 | 80.1 | 61.8 | 94.1 |
| TweetEval | 96.2 | 96.7 | 67.7 | 94.6 | 96.6 | 97.0 | 61.8 | 95.1 | 56.8 | 46.3 | 42.3 | 55.6 | 89.7 | 90.1 | 82.3 | 89.8 |
| Credit Fraud | 96.6 | 97.0 | 87.1 | 97.7 | 96.6 | 97.0 | 87.1 | 97.5 | 77.0 | 77.1 | 92.9 | 86.6 | 89.8 | 90.2 | 96.9 | 98.4 |
| Adult | 96.4 | 95.8 | 89.9 | 96.9 | 96.5 | 95.9 | 60.2 | 97.5 | 64.5 | 65.6 | 67.9 | 64.3 | 89.8 | 90.3 | 83.9 | 92.5 |
| Dry Bean | 96.7 | 96.5 | 64.6 | 97.3 | 96.4 | 96.4 | 53.1 | 97.3 | 88.9 | 95.7 | 39.8 | 94.9 | 89.3 | 95.4 | 39.5 | 93.9 |
| Car Evaluation | 97.4 | 97.6 | 81.2 | 97.3 | 96.1 | 98.4 | 81.0 | 96.0 | 78.4 | 86.7 | 81.6 | 81.9 | 91.2 | 90.4 | 87.1 | 92.1 |
| Mushrooms | 96.2 | 98.4 | 68.7 | 97.6 | 96.5 | 98.6 | 92.6 | 98.2 | 99.5 | 100 | 100 | 100 | 89.6 | 96.0 | 86.2 | 98.1 |
| COMPAS | 96.6 | 93.9 | 82.0 | 96.7 | 96.7 | 93.7 | 87.5 | 94.5 | 55.2 | 58.9 | 56.4 | 57.2 | 89.7 | 88.7 | 82.5 | 89.3 |
| Crop | 96.8 | 88.0 | 7.9 | 94.0 | 96.1 | 86.5 | 7.8 | 94.3 | 54.3 | 72.7 | 20.1 | 70.9 | 89.7 | 85.8 | 9.5 | 90.4 |
| Electric Devices | 96.3 | 91.1 | 40.1 | 96.7 | 96.6 | 91.4 | 39.6 | 96.7 | 83.4 | 88.7 | 33.8 | 88.3 | 90.2 | 89.0 | 38.8 | 92.6 |
| MIT-BIH | 96.8 | 97.7 | 73.6 | 97.9 | 96.6 | 97.6 | 70.2 | 97.4 | 73.0 | 82.3 | 64.6 | 89.6 | 89.8 | 95.2 | 70.2 | 94.1 |
| PenDigits | 96.2 | 96.9 | 17.4 | 99.4 | 96.6 | 97.4 | 18.1 | 99.4 | 94.0 | 97.5 | 17.8 | 98.4 | 90.4 | 95.6 | 17.9 | 94.2 |
| WhaleCalls | 96.8 | 88.5 | 72.8 | 95.4 | 96.6 | 90.0 | 76.2 | 95.7 | 91.5 | 90.2 | 67.8 | 91.0 | 90.2 | 81.0 | 76.0 | 91.1 |

Table 14: Performance evaluation of cleaning methods to detect erroneous labels across different types of synthetic noise added to the train set in terms of weighted $F_1$, for noise rate $= 0.02$. The classification models used for images, text, tabular, and time series datasets are MobileNet-v2, `all-MiniLM-L6-v2`, Multi-layer perception (random seed 43), and Fully convolutional network, respectively.

| Datasets | Uniform | | | | Asymmetric | | | | Class-dependent | | | | Instance-dependent | | | |
|---|---|---|---|---|---|---|---|---|---|---|---|---|---|---|---|---|
| | AUM | CIN | CON | SIM | AUM | CIN | CON | SIM | AUM | CIN | CON | SIM | AUM | CIN | CON | SIM |
| CIFAR-10 | 96.6 | 72.5 | 97.0 | 97.0 | 96.4 | 72.3 | 97.0 | 97.0 | 91.8 | 69.8 | 92.2 | 92.2 | 90.3 | 68.4 | 90.6 | 90.6 |
| Clothing-100K | 96.7 | 97.0 | 97.1 | 97.1 | 96.6 | 97.0 | 97.0 | 97.2 | 49.2 | 41.5 | 48.8 | 63.5 | 90.4 | 82.0 | 90.7 | 93.2 |
| NoisyCXR | 96.6 | 73.5 | 97.1 | 94.8 | 96.6 | 74.1 | 97.0 | 94.4 | 63.6 | 68.6 | 63.4 | 73.5 | 89.8 | 71.4 | 90.2 | 89.8 |
| Credit Fraud | 96.6 | 97.0 | 99.8 | 97.6 | 96.6 | 97.0 | 99.8 | 97.5 | 34.8 | 33.9 | 79.9 | 33.9 | 89.8 | 90.1 | 98.4 | 90.3 |
| Adult | 96.6 | 95.8 | 97.0 | 95.6 | 97.0 | 95.9 | 97.0 | 95.3 | 64.6 | 65.6 | 64.4 | 67.2 | 89.9 | 90.2 | 90.2 | 90.9 |
| Dry Bean | 96.4 | 96.1 | 28.5 | 97.1 | 96.6 | 95.9 | 32.4 | 97.0 | 86.7 | 95.3 | 32.4 | 92.5 | 90.4 | 95.0 | 28.6 | 93.3 |
| Car Evaluation | 96.5 | 96.9 | 83.5 | 95.9 | 96.4 | 98.2 | 83.6 | 95.6 | 69.7 | 82.7 | 83.0 | 73.6 | 90.1 | 96.2 | 79.6 | 93.0 |
| Mushrooms | 96.6 | 98.7 | 69.6 | 99.3 | 96.5 | 98.5 | 69.7 | 99.6 | 99.4 | 100 | 66.4 | 99.7 | 90.0 | 96.2 | 74.4 | 97.9 |
| COMPAS | 96.7 | 93.9 | 74.3 | 95.1 | 96.6 | 93.6 | 69.7 | 94.9 | 55.2 | 57.4 | 50.9 | 60.0 | 90.1 | 88.6 | 62.9 | 88.7 |
| Crop | 96.3 | 97.2 | 7.8 | 57.5 | 96.6 | 97.0 | 8.0 | 57.4 | 3.1 | 0.4 | 91.6 | 67.6 | 89.9 | 90.7 | 8.3 | 53.6 |
| Electric Devices | 96.5 | 97.1 | 31.7 | 73.2 | 96.6 | 97.0 | 37.0 | 71.4 | 6.0 | 3.5 | 77.2 | 61.1 | 90.3 | 90.6 | 32.4 | 69.9 |
| MIT-BIH | 96.4 | 94.9 | 41.4 | 96.8 | 96.5 | 95.4 | 52.7 | 97.0 | 82.5 | 81.7 | 40.4 | 82.7 | 89.7 | 91.4 | 37.2 | 90.0 |
| PenDigits | 96.3 | 96.8 | 17.7 | 58.5 | 96.6 | 97.0 | 17.6 | 57.7 | 7.0 | 1.7 | 82.2 | 64.4 | 90.0 | 90.4 | 17.4 | 54.5 |
| WhaleCalls | 96.4 | 96.8 | 64.8 | 94.5 | 96.5 | 97.0 | 64.9 | 94.6 | 35.0 | 34.1 | 50.2 | 40.3 | 90.1 | 90.4 | 61.7 | 87.4 |

Table 15: Performance evaluation of cleaning methods to detect erroneous labels across different types of synthetic noise added to the train set in terms of weighted $F_1$, for noise rate $= 0.02$. The classification models used for images, tabular, and time series datasets are Fast-ViT-T8, TabTransformer, and PatchTST, respectively.

### A.8.3 Critical Difference Diagrams

To compare cleaning methods and downstream classifiers across multiple datasets, we follow the recommendations of Demšar [80]. First, we use the Friedman test [90] to evaluate whether a statistically significant difference exists between classifiers' performance. Then, for classifiers with

| Datasets | No Noise Injected | | | | | Uniform | | | | | Asymmetric | | | | | Class-dependent | | | | | Instance-dependent | | | | |
|---|---|---|---|---|---|---|---|---|---|---|---|---|---|---|---|---|---|---|---|---|---|---|---|---|---|
| | NON | AUM | CIN | CON | SIM | NON | AUM | CIN | CON | SIM | NON | AUM | CIN | CON | SIM | NON | AUM | CIN | CON | SIM | NON | AUM | CIN | CON | SIM |
| CIFAR-10 | 81.1 | 81.1 | 80.4 | 24.0 | 79.4 | 74.4 | 73.3 | 78.0 | 18.6 | 75.7 | 74.7 | 74.2 | 77.0 | 12.5 | 75.4 | 80.5 | 80.6 | 79.9 | 27.0 | 80.3 | 72.6 | 71.2 | 76.4 | 12.5 | 73.4 |
| Clothing-100K | 90.9 | 91.0 | 91.1 | 90.9 | 90.9 | 87.8 | 86.5 | 81.5 | 89.9 | 88.5 | 90.0 | 89.9 | 89.8 | 90.9 | 89.9 | 90.0 | 89.6 | 88.7 | 90.3 | 90.3 | 87.6 | 77.4 | 86.6 | 86.2 | 84.7 |
| NoisyCXR | 65.4 | 65.3 | 64.7 | 10.4 | 64.5 | 61.6 | 61.0 | 63.3 | 7.3 | 61.8 | 61.3 | 62.1 | 63.8 | 9.5 | 61.5 | 65.0 | 65.7 | 65.0 | 7.3 | 65.8 | 59.4 | 59.3 | 62.0 | 13.0 | 59.1 |
| IMDb | 89.1 | 90.5 | 93.1 | 80.0 | 92.1 | 92.5 | 92.0 | 92.3 | 80.5 | 89.4 | 91.3 | 90.4 | 87.8 | 88.9 | 92.2 | 92.4 | 92.3 | 91.1 | 91.3 | 89.6 | 90.0 | 91.6 | 89.1 | 86.9 | 79.2 |
| TweetEval | 82.1 | 80.7 | 81.9 | 60.4 | 81.8 | 82.5 | 76.7 | 80.0 | 78.9 | 82.0 | 81.5 | 66.9 | 81.8 | 79.6 | 81.4 | 81.6 | 82.3 | 77.9 | 66.7 | 76.8 | 79.6 | 78.0 | 78.5 | 80.0 | 78.7 |
| Credit Fraud | 100 | 99.9 | 100 | 100 | 100 | 100 | 99.9 | 100 | 100 | 99.9 | 100 | 100 | 99.9 | 99.7 | 100 | 100 | 99.9 | 99.9 | 99.9 | 100 | 99.9 | 100 | 99.7 | 99.9 | 99.9 |
| Adult | 84.6 | 84.2 | 84.5 | 81.4 | 84.3 | 84.1 | 83.9 | 84.0 | 80.6 | 84.4 | 84.3 | 84.2 | 84.0 | 83.8 | 84.1 | 81.0 | 82.3 | 82.5 | 83.4 | 81.0 | 84.2 | 84.0 | 84.1 | 72.8 | 83.7 |
| Dry Bean | 91.6 | 91.0 | 90.5 | 32.3 | 91.4 | 89.2 | 91.1 | 90.7 | 28.7 | 86.2 | 84.0 | 91.2 | 91.2 | 48.6 | 89.7 | 92.3 | 85.5 | 90.3 | 26.0 | 79.1 | 90.6 | 88.4 | 90.4 | 33.8 | 90.1 |
| Car Evaluation | 93.9 | 89.9 | 85.4 | 57.6 | 87.8 | 83.7 | 81.5 | 74.4 | 57.6 | 67.0 | 85.8 | 82.0 | 75.4 | 63.2 | 64.4 | 89.2 | 88.0 | 71.2 | 57.6 | 92.1 | 80.1 | 79.8 | 60.6 | 57.6 | 74.6 |
| Mushrooms | 100 | 100 | 99.3 | 99.3 | 99.3 | 99.1 | 98.7 | 99.1 | 98.6 | 99.7 | 98.8 | 99.8 | 99.3 | 98.3 | 99.0 | 99.7 | 100 | 100 | 97.0 | 99.3 | 99.7 | 99.1 | 99.8 | 97.1 | 99.9 |
| COMPAS | 67.5 | 67.1 | 64.5 | 60.4 | 66.0 | 66.2 | 67.3 | 67.3 | 62.1 | 65.7 | 64.7 | 66.2 | 65.5 | 30.0 | 65.2 | 66.8 | 67.6 | 65.5 | 61.4 | 68.2 | 66.1 | 65.6 | 38.5 | 60.5 | 68.6 |
| Crop | 52.7 | 50.7 | 47.8 | 2.2 | 60.1 | 51.4 | 57.0 | 52.7 | 6.8 | 49.7 | 46.2 | 46.2 | 45.2 | 3.3 | 56.1 | 49.8 | 47.9 | 41.6 | 12.7 | 40.9 | 53.7 | 51.2 | 41.0 | 4.1 | 53.1 |
| Electric Devices | 61.8 | 65.8 | 67.6 | 31.5 | 64.1 | 64.8 | 65.5 | 65.2 | 23.9 | 50.4 | 61.6 | 63.6 | 61.2 | 30.4 | 61.1 | 53.2 | 57.4 | 53.1 | 38.4 | 52.4 | 62.4 | 51.9 | 64.1 | 43.0 | 58.2 |
| MIT-BIH | 65.6 | 44.1 | 88.4 | 58.3 | 68.6 | 86.3 | 54.0 | 88.1 | 86.4 | 78.2 | 60.1 | 75.5 | 85.9 | 6.0 | 56.4 | 79.1 | 75.4 | 84.6 | 70.7 | 80.2 | 83.2 | 87.2 | 85.8 | 63.5 | 75.4 |
| PenDigits | 95.8 | 93.7 | 95.5 | 28.0 | 95.0 | 95.4 | 96.0 | 96.1 | 33.5 | 95.8 | 91.9 | 92.6 | 93.7 | 22.6 | 95.7 | 94.0 | 95.5 | 96.8 | 32.6 | 95.3 | 93.4 | 72.0 | 93.8 | 34.0 | 95.1 |
| WhaleCalls | 75.1 | 33.3 | 34.2 | 46.4 | 33.3 | 36.7 | 33.3 | 33.3 | 48.1 | 33.5 | 33.4 | 33.5 | 39.3 | 34.3 | 37.6 | 33.3 | 32.2 | 33.3 | 33.3 | 33.3 | 33.3 | 33.3 | 79.8 | 33.3 | 33.3 |

Table 16: Impact of label noise and each cleaning method on weighted $F_1$ score of a downstream model for each modality on the test set for noise rate $= 0.1$. The classification models used for images, text, tabular, and time series datasets are ResNet-18, `all-distilroberta-v1`, Multi-layer perception (random seed 42), and ResNet-1D, respectively.

| Datasets | No Noise Injected | | | | | Uniform | | | | | Asymmetric | | | | | Class-dependent | | | | | Instance-dependent | | | | |
|---|---|---|---|---|---|---|---|---|---|---|---|---|---|---|---|---|---|---|---|---|---|---|---|---|---|
| | NON | AUM | CIN | CON | SIM | NON | AUM | CIN | CON | SIM | NON | AUM | CIN | CON | SIM | NON | AUM | CIN | CON | SIM | NON | AUM | CIN | CON | SIM |
| CIFAR-10 | 80.3 | 79.9 | 80.1 | 52.6 | 80.3 | 75.1 | 77.4 | 68.3 | 43.6 | 69.6 | 75.7 | 72.2 | 67.7 | 52.4 | 75.9 | 74.9 | 76.5 | 77.0 | 56.9 | 63.4 | 73.0 | 75.8 | 74.3 | 47.9 | 71.9 |
| Clothing-100K | 91.0 | 90.4 | 90.3 | 90.9 | 90.4 | 89.6 | 85.2 | 90.1 | 91.0 | 89.0 | 88.8 | 88.9 | 89.8 | 90.9 | 84.6 | 80.1 | 71.2 | 71.7 | 90.8 | 84.0 | 64.8 | 74.6 | 76.6 | 78.7 | 80.0 |
| NoisyCXR | 63.4 | 65.0 | 65.3 | 19.5 | 63.2 | 60.0 | 58.9 | 64.8 | 12.3 | 58.5 | 60.1 | 60.3 | 63.9 | 3.6 | 59.7 | 61.6 | 60.1 | 65.8 | 8.5 | 61.2 | 57.5 | 56.9 | 56.9 | 13.0 | 56.3 |
| IMDb | 80.7 | 84.4 | 85.2 | 59.2 | 88.4 | 78.3 | 65.0 | 84.7 | 86.8 | 77.4 | 78.4 | 73.9 | 82.2 | 85.5 | 83.2 | 81.8 | 77.6 | 87.0 | 79.7 | 84.5 | 78.1 | 71.9 | 81.9 | 66.4 | 76.1 |
| TweetEval | 65.0 | 66.4 | 72.2 | 69.7 | 71.7 | 71.9 | 61.1 | 80.6 | 66.8 | 77.5 | 61.4 | 61.7 | 72.3 | 68.6 | 76.1 | 72.2 | 77.9 | 79.4 | 36.1 | 79.0 | 64.0 | 71.5 | 73.9 | 69.2 | 71.8 |
| Credit Fraud | 100 | 100 | 99.9 | 100 | 100 | 100 | 100 | 100 | 99.9 | 100 | 100 | 100 | 99.9 | 100 | 100 | 100 | 99.9 | 100 | 100 | 99.9 | 100 | 99.9 | 99.9 | 100 | 99.9 |
| Adult | 84.3 | 84.4 | 84.1 | 76.4 | 84.3 | 83.9 | 84.1 | 84.0 | 78.0 | 84.2 | 84.0 | 84.1 | 84.1 | 78.7 | 84.0 | 82.8 | 82.5 | 83.7 | 83.5 | 83.1 | 84.2 | 83.4 | 84.2 | 68.9 | 83.8 |
| Dry Bean | 92.1 | 91.2 | 91.1 | 78.9 | 90.5 | 82.2 | 90.7 | 91.3 | 82.5 | 38.3 | 83.8 | 84.2 | 91.6 | 64.5 | 86.4 | 90.8 | 91.2 | 89.0 | 17.6 | 91.3 | 91.5 | 85.1 | 90.6 | 62.3 | 90.4 |
| Car Evaluation | 91.8 | 89.9 | 77.7 | 57.6 | 89.8 | 82.5 | 81.3 | 78.9 | 57.6 | 86.2 | 82.6 | 83.6 | 60.4 | 57.6 | 82.3 | 90.6 | 86.1 | 87.3 | 57.6 | 82.7 | 80.7 | 80.0 | 75.4 | 59.2 | 80.2 |
| Mushrooms | 99.3 | 100 | 99.3 | 99.8 | 100 | 100 | 99.1 | 99.0 | 97.0 | 99.6 | 98.8 | 98.6 | 99.1 | 98.2 | 100 | 99.1 | 100 | 98.1 | 98.7 | 100 | 99.5 | 99.6 | 98.7 | 87.1 | 99.1 |
| COMPAS | 66.7 | 66.7 | 66.3 | 67.1 | 66.5 | 66.9 | 68.2 | 65.6 | 66.4 | 67.5 | 65.7 | 68.0 | 67.4 | 38.4 | 66.3 | 28.6 | 66.2 | 64.6 | 28.4 | 66.7 | 65.4 | 65.4 | 66.2 | 38.5 | 65.0 |
| Crop | 64.0 | 64.8 | 58.2 | 22.5 | 52.8 | 62.2 | 61.9 | 63.7 | 27.6 | 62.8 | 63.0 | 67.0 | 61.1 | 29.9 | 46.9 | 45.2 | 46.1 | 42.9 | 14.2 | 46.7 | 58.1 | 63.1 | 60.0 | 21.4 | 62.2 |
| Electric Devices | 64.5 | 68.6 | 66.9 | 48.3 | 66.4 | 61.2 | 65.3 | 66.1 | 53.8 | 65.3 | 64.8 | 58.8 | 57.5 | 54.3 | 61.2 | 15.2 | 4.6 | 16.0 | 11.3 | 14.9 | 62.6 | 62.2 | 62.9 | 49.7 | 60.9 |
| MIT-BIH | 86.3 | 85.5 | 85.3 | 86.6 | 84.2 | 85.5 | 85.5 | 85.5 | 85.4 | 84.2 | 85.6 | 85.9 | 86.1 | 85.9 | 85.9 | 85.7 | 85.5 | 81.8 | 81.6 | 84.0 | 85.0 | 85.8 | 84.5 | 85.3 | 84.2 |
| PenDigits | 96.6 | 97.8 | 95.3 | 88.6 | 96.5 | 97.7 | 97.3 | 97.4 | 64.6 | 95.3 | 93.1 | 95.2 | 97.2 | 58.7 | 83.7 | 5.7 | 14.8 | 12.7 | 6.5 | 10.7 | 83.7 | 93.8 | 95.0 | 47.5 | 95.6 |
| WhaleCalls | 96.1 | 36.1 | 85.0 | 78.3 | 92.4 | 85.7 | 45.3 | 79.3 | 34.9 | 86.0 | 84.1 | 83.5 | 86.0 | 69.3 | 84.5 | 71.4 | 33.3 | 33.3 | 33.3 | 33.3 | 50.0 | 47.3 | 50.5 | 50.4 | 82.8 |

Table 17: Impact of label noise and each cleaning method on weighted $F_1$ score of a downstream model for each modality on the test set for noise rate $= 0.1$. The classification models used for images, text, tabular, and time series datasets are MobileNet-v2, `all-MiniLM-L6-v2`, Multi-layer perception (random seed 43), and Fully convolutional network, respectively.

| Datasets | No Noise Injected | | | | | Uniform | | | | | Asymmetric | | | | | Class-dependent | | | | | Instance-dependent | | | | |
|---|---|---|---|---|---|---|---|---|---|---|---|---|---|---|---|---|---|---|---|---|---|---|---|---|---|
| | NON | AUM | CIN | CON | SIM | NON | AUM | CIN | CON | SIM | NON | AUM | CIN | CON | SIM | NON | AUM | CIN | CON | SIM | NON | AUM | CIN | CON | SIM |
| CIFAR-10 | 61.5 | 61.4 | 58.4 | 61.5 | 60.6 | 52.0 | 53.3 | 53.9 | 53.5 | 52.9 | 52.7 | 54.9 | 53.1 | 52.6 | 54.2 | 57.7 | 58.2 | 54.8 | 56.9 | 58.6 | 50.4 | 49.5 | 52.4 | 48.5 | 49.0 |
| Clothing-100K | 90.9 | 90.6 | 90.2 | 90.7 | 90.8 | 81.6 | 69.3 | 88.9 | 87.1 | 85.7 | 84.4 | 88.1 | 85.1 | 87.0 | 90.8 | 72.1 | 88.5 | 84.8 | 70.1 | 83.7 | 74.7 | 84.8 | 36.9 | 83.3 | 87.2 |
| NoisyCXR | 39.7 | 49.3 | 40.4 | 45.5 | 43.3 | 40.8 | 39.2 | 37.4 | 39.8 | 37.5 | 40.0 | 37.8 | 40.7 | 41.8 | 39.8 | 35.7 | 38.7 | 36.3 | 38.3 | 34.5 | 37.3 | 37.8 | 35.9 | 39.7 | 36.9 |
| Credit Fraud | 99.9 | 99.9 | 99.7 | 99.9 | 99.9 | 99.9 | 99.9 | 99.7 | 99.7 | 99.9 | 99.9 | 99.9 | 99.9 | 99.9 | 99.9 | 98.9 | 0.0 | 99.4 | 99.9 | 0.0 | 99.8 | 99.6 | 99.9 | 99.7 | 99.9 |
| Adult | 83.1 | 83.2 | 83.4 | 83.5 | 83.4 | 82.1 | 83.5 | 82.9 | 83.5 | 83.6 | 82.5 | 82.1 | 82.4 | 82.9 | 83.4 | 80.4 | 81.5 | 82.6 | 82.7 | 75.8 | 83.3 | 81.4 | 82.9 | 82.7 | 83.3 |
| Dry Bean | 91.8 | 90.3 | 91.8 | 44.7 | 90.0 | 92.1 | 91.5 | 92.4 | 35.6 | 91.3 | 92.4 | 92.0 | 91.5 | 46.3 | 91.1 | 91.6 | 91.9 | 91.6 | 52.6 | 92.1 | 91.5 | 91.8 | 88.6 | 25.9 | 92.1 |
| Car Evaluation | 95.2 | 97.7 | 97.0 | 57.6 | 97.7 | 84.0 | 87.0 | 85.6 | 83.3 | 89.3 | 91.2 | 89.2 | 91.8 | 62.7 | 91.0 | 86.2 | 83.2 | 77.7 | 57.6 | 83.2 | 86.5 | 89.6 | 88.8 | 57.6 | 95.6 |
| Mushrooms | 100 | 99.8 | 100 | 100 | 100 | 99.3 | 99.8 | 99.2 | 97.9 | 99.8 | 97.9 | 97.9 | 99.9 | 99.6 | 98.6 | 99.8 | 100 | 99.9 | 99.7 | 100 | 98.9 | 99.9 | 99.9 | 92.0 | 99.8 |
| COMPAS | 67.7 | 68.3 | 67.0 | 61.3 | 67.0 | 68.5 | 68.0 | 66.3 | 62.4 | 67.5 | 67.1 | 66.8 | 66.8 | 58.8 | 67.2 | 68.0 | 63.0 | 61.4 | 36.8 | 65.5 | 65.2 | 66.8 | 66.3 | 38.5 | 62.8 |
| Crop | 0.3 | 0.3 | 0.3 | 0.3 | 0.3 | 0.3 | 0.3 | 0.3 | 0.3 | 0.3 | 0.3 | 0.3 | 0.3 | 0.3 | 0.3 | 0.3 | 0.3 | 0.3 | 0.3 | 0.3 | 0.3 | 0.3 | 0.3 | 0.3 | 0.3 |
| Electric Devices | 9.5 | 9.5 | 9.5 | 9.5 | 9.5 | 9.5 | 9.5 | 9.5 | 9.5 | 9.5 | 9.5 | 9.5 | 9.5 | 9.5 | 9.5 | 9.5 | 9.5 | 10.3 | 10.3 | 9.5 | 10.3 | 9.5 | 9.5 | 9.5 | 10.3 |
| MIT-BIH | 66.2 | 65.8 | 69.8 | 22.2 | 64.7 | 55.8 | 58.9 | 64.4 | 22.3 | 58.0 | 69.2 | 66.2 | 67.7 | 22.2 | 66.7 | 51.6 | 56.3 | 59.1 | 35.2 | 66.3 | 67.0 | 65.7 | 59.9 | 33.7 | 66.5 |
| PenDigits | 2.0 | 2.0 | 2.0 | 2.0 | 2.0 | 2.0 | 2.0 | 2.0 | 1.7 | 2.0 | 2.0 | 2.0 | 2.0 | 2.0 | 2.0 | 1.7 | 1.7 | 1.7 | 1.7 | 1.7 | 1.7 | 1.7 | 1.7 | 2.0 | 1.7 |
| WhaleCalls | 33.3 | 33.3 | 33.3 | 33.3 | 33.3 | 33.3 | 33.3 | 33.3 | 33.3 | 33.3 | 33.3 | 33.3 | 33.3 | 33.3 | 33.3 | 33.3 | 33.3 | 33.3 | 33.3 | 33.3 | 33.3 | 33.3 | 33.3 | 33.3 | 33.3 |

Table 18: Impact of label noise and each cleaning method on weighted $F_1$ score of a downstream model for each modality on the test set for noise rate $= 0.1$. The classification models used for images, tabular, and time series datasets are Fast-ViT-T8, TabTransformer, and PatchTST, respectively.

significantly different performance, we conduct pairwise post-hoc analysis recommended by Benavoli et al. [81] where the average rank comparison is replaced with the Wilcoxon signed-rank test [91] with Holm's alpha correction [92]. The thick horizontal line in a critical difference diagram shows models that are not significantly different in performance.

| Datasets | No Noise Injected | | | | | Uniform | | | | | Asymmetric | | | | | Class-dependent | | | | | Instance-dependent | | | | |
|---|---|---|---|---|---|---|---|---|---|---|---|---|---|---|---|---|---|---|---|---|---|---|---|---|---|
| | NON | AUM | CIN | CON | SIM | NON | AUM | CIN | CON | SIM | NON | AUM | CIN | CON | SIM | NON | AUM | CIN | CON | SIM | NON | AUM | CIN | CON | SIM |
| CIFAR-10 | 81.1 | 81.0 | 80.4 | 24.0 | 79.4 | 50.4 | 50.6 | 64.7 | 8.5 | 53.7 | 50.1 | 50.2 | 53.2 | 19.4 | 50.4 | 80.4 | 80.6 | 79.9 | 26.9 | 80.3 | 47.9 | 46.2 | 51.0 | 16.3 | 49.2 |
| Clothing-100K | 90.9 | 91.0 | 91.1 | 90.9 | 90.9 | 61.3 | 62.3 | 57.7 | 79.7 | 67.2 | 70.9 | 60.8 | 63.8 | 28.9 | 70.9 | 90.0 | 89.6 | 88.7 | 90.2 | 90.3 | 82.4 | 64.3 | 50.4 | 73.8 | 70.3 |
| NoisyCXR | 65.4 | 65.3 | 64.7 | 10.4 | 64.5 | 44.2 | 44.1 | 56.9 | 10.1 | 43.4 | 41.1 | 39.8 | 45.5 | 9.5 | 40.6 | 65.0 | 65.7 | 65.0 | 7.2 | 65.8 | 40.5 | 40.4 | 47.6 | 7.4 | 39.3 |
| IMDb | 89.1 | 90.5 | 93.1 | 80.0 | 92.1 | 33.3 | 33.3 | 33.3 | 33.3 | 33.3 | 33.3 | 33.3 | 77.8 | 33.3 | 66.6 | 92.4 | 92.2 | 91.1 | 91.3 | 89.6 | 33.3 | 33.3 | 33.3 | 33.3 | 33.3 |
| TweetEval | 82.1 | 80.6 | 81.9 | 60.4 | 81.8 | 60.4 | 74.0 | 60.4 | 23.5 | 60.4 | 60.4 | 60.4 | 60.4 | 12.1 | 60.4 | 81.6 | 82.3 | 77.9 | 66.7 | 76.8 | 60.4 | 60.4 | 64.8 | 1.6 | 60.4 |
| Credit Fraud | 100 | 100 | 99.9 | 99.9 | 100 | 100 | 100 | 100 | 99.9 | 100 | 100 | 99.9 | 100 | 100 | 99.9 | 99.9 | 100 | 100 | 99.7 | 100 | 99.9 | 99.9 | 100 | 99.9 | 99.9 |
| Adult | 84.2 | 84.3 | 84.1 | 76.4 | 84.3 | 83.9 | 84.1 | 84.0 | 78.0 | 84.2 | 84.0 | 84.1 | 84.1 | 78.7 | 84.0 | 82.7 | 82.5 | 83.7 | 83.5 | 83.1 | 84.2 | 83.3 | 84.2 | 68.9 | 83.8 |
| Dry Bean | 92.1 | 91.2 | 91.1 | 78.9 | 90.5 | 82.1 | 90.7 | 91.3 | 82.5 | 38.3 | 83.8 | 84.2 | 91.6 | 64.5 | 86.4 | 90.8 | 91.2 | 88.9 | 17.6 | 91.2 | 91.5 | 85.1 | 90.6 | 62.3 | 90.4 |
| Car Evaluation | 91.8 | 90.0 | 77.7 | 57.6 | 89.8 | 82.5 | 81.3 | 78.9 | 57.6 | 86.2 | 82.6 | 83.6 | 60.4 | 57.6 | 82.2 | 90.6 | 86.0 | 87.3 | 57.6 | 82.6 | 80.7 | 80.0 | 75.4 | 59.2 | 80.1 |
| Mushrooms | 99.3 | 100 | 99.3 | 99.7 | 100 | 100 | 99.1 | 99.0 | 97.0 | 99.6 | 98.8 | 98.6 | 99.1 | 98.2 | 100 | 99.1 | 100 | 98.1 | 98.6 | 100 | 99.5 | 99.5 | 98.7 | 87.1 | 99.1 |
| COMPAS | 66.7 | 66.7 | 66.3 | 67.1 | 66.5 | 66.9 | 68.1 | 65.6 | 66.3 | 67.5 | 65.7 | 68.0 | 67.4 | 38.3 | 66.3 | 28.6 | 66.2 | 64.6 | 28.4 | 66.6 | 65.4 | 65.4 | 66.2 | 38.4 | 65.0 |
| Crop | 52.7 | 50.7 | 47.8 | 2.2 | 60.0 | 18.3 | 41.4 | 41.2 | 3.8 | 47.3 | 39.5 | 38.4 | 30.4 | 6.1 | 37.3 | 49.7 | 47.9 | 41.6 | 12.7 | 40.9 | 23.9 | 23.0 | 12.9 | 1.6 | 38.5 |
| Electric Devices | 61.8 | 65.8 | 67.6 | 31.5 | 64.1 | 54.8 | 53.7 | 58.0 | 24.6 | 54.8 | 51.9 | 53.7 | 55.6 | 27.1 | 50.4 | 53.2 | 57.4 | 53.1 | 38.4 | 52.4 | 39.6 | 34.9 | 43.1 | 1.7 | 52.6 |
| MIT-BIH | 65.6 | 44.0 | 88.4 | 58.3 | 68.6 | 79.6 | 83.7 | 89.9 | 40.5 | 77.7 | 69.8 | 70.7 | 33.0 | 41.9 | 60.8 | 79.1 | 75.4 | 84.6 | 70.7 | 80.2 | 56.7 | 58.8 | 54.1 | 67.9 | 71.8 |
| PenDigits | 95.7 | 95.6 | 95.5 | 27.9 | 94.9 | 89.8 | 94.5 | 91.3 | 25.4 | 92.1 | 72.3 | 80.7 | 68.5 | 22.4 | 85.9 | 99.1 | 99.5 | 96.8 | 32.6 | 95.3 | 75.0 | 80.7 | 64.8 | 1.6 | 77.8 |
| WhaleCalls | 75.1 | 33.3 | 34.1 | 46.4 | 33.3 | 33.3 | 33.3 | 74.5 | 31.2 | 68.3 | 33.3 | 77.5 | 33.3 | 33.3 | 33.3 | 33.3 | 32.2 | 33.3 | 33.3 | 33.3 | 33.3 | 65.8 | 71.3 | 33.8 | 33.3 |

Table 19: Impact of label noise and each cleaning method on weighted $F_1$ score of a downstream model for each modality on the test set for noise rate $= 0.4$. The classification models used for images, text, tabular, and time series datasets are ResNet-18, `all-distilroberta-v1`, Multi-layer perception (random seed 42), and ResNet-1D, respectively.

| Datasets | No Noise Injected | | | | | Uniform | | | | | Asymmetric | | | | | Class-dependent | | | | | Instance-dependent | | | | |
|---|---|---|---|---|---|---|---|---|---|---|---|---|---|---|---|---|---|---|---|---|---|---|---|---|---|
| | NON | AUM | CIN | CON | SIM | NON | AUM | CIN | CON | SIM | NON | AUM | CIN | CON | SIM | NON | AUM | CIN | CON | SIM | NON | AUM | CIN | CON | SIM |
| CIFAR-10 | 80.3 | 79.9 | 80.1 | 52.6 | 80.3 | 64.1 | 55.7 | 65.4 | 29.4 | 65.9 | 13.3 | 63.9 | 58.5 | 29.5 | 59.5 | 74.9 | 76.6 | 77.0 | 56.9 | 63.4 | 35.3 | 57.8 | 59.9 | 22.7 | 60.4 |
| Clothing-100K | 91.0 | 90.5 | 90.3 | 90.9 | 90.5 | 70.0 | 62.4 | 79.2 | 77.7 | 59.1 | 73.8 | 69.7 | 47.3 | 88.3 | 64.9 | 80.1 | 71.3 | 71.8 | 90.9 | 84.0 | 63.6 | 28.0 | 69.1 | 70.4 | 63.1 |
| NoisyCXR | 63.4 | 65.0 | 65.4 | 19.6 | 63.3 | 44.9 | 45.4 | 53.0 | 8.6 | 42.4 | 43.1 | 38.9 | 43.5 | 9.6 | 38.9 | 61.7 | 60.2 | 65.9 | 8.5 | 61.2 | 36.1 | 36.8 | 40.2 | 8.8 | 40.8 |
| IMDb | 80.8 | 84.5 | 85.2 | 59.2 | 88.4 | 33.3 | 33.3 | 33.3 | 33.3 | 43.3 | 77.3 | 33.3 | 45.3 | 33.3 | 63.5 | 81.8 | 77.6 | 87.0 | 79.8 | 84.6 | 33.3 | 33.3 | 33.3 | 33.3 | 33.3 |
| TweetEval | 65.0 | 66.5 | 72.3 | 69.7 | 71.8 | 69.9 | 60.7 | 74.2 | 12.2 | 60.4 | 60.4 | 73.0 | 60.4 | 60.4 | 60.4 | 72.2 | 77.9 | 79.5 | 36.1 | 79.0 | 60.4 | 60.4 | 59.9 | 60.4 | 12.2 |
| Credit Fraud | 100 | 100 | 99.9 | 99.9 | 100 | 99.8 | 99.8 | 99.7 | 0.1 | 99.7 | 99.8 | 99.9 | 99.8 | 99.8 | 99.8 | 99.9 | 100 | 99.9 | 99.7 | 100 | 99.7 | 99 | 99.7 | 99.7 | 99.1 |
| Adult | 84.3 | 84.4 | 84.1 | 76.4 | 84.3 | 81.7 | 82.0 | 81.3 | 35.3 | 82.0 | 82.1 | 80.3 | 80.6 | 66.5 | 80.3 | 82.8 | 82.6 | 83.7 | 83.6 | 83.1 | 77.0 | 79.5 | 79.8 | 66.2 | 77.2 |
| Dry Bean | 92.1 | 91.2 | 91.2 | 78.9 | 90.6 | 90.3 | 89.6 | 89.1 | 54.5 | 85.6 | 78.3 | 78.5 | 80.7 | 28.9 | 75.9 | 90.8 | 91.2 | 89.0 | 17.7 | 91.3 | 65.3 | 67.9 | 60.3 | 3.5 | 78.3 |
| Car Evaluation | 91.9 | 89.9 | 77.7 | 57.6 | 89.8 | 75.6 | 60.8 | 60.3 | 57.6 | 57.6 | 80.3 | 76.3 | 61.9 | 57.6 | 74.2 | 90.6 | 86.1 | 87.3 | 57.6 | 82.7 | 67.7 | 63.9 | 63.4 | 57.6 | 59.8 |
| Mushrooms | 99.3 | 100 | 99.3 | 99.8 | 100 | 96.5 | 95.2 | 95.5 | 64.4 | 94.8 | 95.7 | 96.3 | 95.5 | 31.8 | 96.6 | 99.1 | 100 | 98.1 | 98.7 | 100 | 86.0 | 88.8 | 88.3 | 31.4 | 85.2 |
| COMPAS | 66.7 | 66.7 | 66.4 | 67.1 | 66.5 | 59.9 | 28.4 | 59.3 | 54.8 | 59.8 | 64.8 | 60.0 | 62.1 | 28.4 | 65.9 | 28.7 | 66.3 | 64.6 | 28.4 | 66.7 | 60.5 | 64.8 | 62.4 | 60.0 | 59.1 |
| Crop | 64.1 | 64.9 | 58.3 | 22.5 | 52.9 | 60.5 | 59.0 | 61.2 | 16.5 | 56.6 | 41.0 | 46.8 | 50.5 | 15.3 | 44.8 | 45.3 | 46.2 | 42.9 | 14.3 | 46.8 | 35.6 | 3.9 | 37.8 | 7.0 | 40.0 |
| Electric Devices | 64.6 | 68.6 | 66.9 | 48.3 | 66.4 | 50.5 | 46.9 | 53.5 | 39.0 | 49.9 | 47.3 | 43.8 | 42.6 | 37.2 | 51.8 | 15.2 | 4.6 | 16.0 | 11.4 | 14.9 | 38.0 | 38.1 | 33.3 | 21.0 | 40.9 |
| MIT-BIH | 86.3 | 85.5 | 85.4 | 86.6 | 84.2 | 86.3 | 84.9 | 85.4 | 84.7 | 87.0 | 85.8 | 80.7 | 83.3 | 75.4 | 83.1 | 85.7 | 85.5 | 81.9 | 81.7 | 84.0 | 43.0 | 55.7 | 75.8 | 48.6 | 78.2 |
| PenDigits | 96.6 | 97.8 | 95.3 | 88.7 | 96.6 | 92.1 | 97.1 | 96.6 | 34.7 | 95.7 | 69.8 | 69.2 | 63.2 | 34.1 | 69.9 | 5.7 | 14.9 | 12.7 | 6.6 | 10.7 | 85.6 | 63.0 | 73.0 | 6.5 | 72.4 |
| WhaleCalls | 96.1 | 36.1 | 85.0 | 78.3 | 92.5 | 58.2 | 59.2 | 54.6 | 58.5 | 60.8 | 59.3 | 60.2 | 55.7 | 58.6 | 61.6 | 44.8 | 50.0 | 47.3 | 50.5 | 50.4 | 57.2 | 56.1 | 51.3 | 50.6 | 53.6 |

Table 20: Impact of label noise and each cleaning method on weighted $F_1$ score of a downstream model for each modality on the test set for noise rate $= 0.4$. The classification models used for images, text, tabular, and time series datasets are MobileNet-v2, `all-MiniLM-L6-v2`, Multi-layer perception (random seed 43), and Fully convolutional network, respectively.

| Datasets | No Noise Injected | | | | | Uniform | | | | | Asymmetric | | | | | Class-dependent | | | | | Instance-dependent | | | | |
|---|---|---|---|---|---|---|---|---|---|---|---|---|---|---|---|---|---|---|---|---|---|---|---|---|---|
| | NON | AUM | CIN | CON | SIM | NON | AUM | CIN | CON | SIM | NON | AUM | CIN | CON | SIM | NON | AUM | CIN | CON | SIM | NON | AUM | CIN | CON | SIM |
| CIFAR-10 | 61.5 | 61.4 | 58.4 | 61.5 | 60.6 | 34.5 | 33.5 | 37.0 | 34.7 | 32.8 | 36.0 | 34.7 | 34.7 | 35.6 | 34.8 | 57.7 | 58.2 | 54.8 | 56.9 | 58.6 | 31.5 | 31.9 | 31.5 | 31.3 | 32.5 |
| Clothing-100K | 90.9 | 90.6 | 90.2 | 90.7 | 90.8 | 80.4 | 77.6 | 78.9 | 70.9 | 87.5 | 69.2 | 65.1 | 90.3 | 26.7 | 87.3 | 72.1 | 88.5 | 84.8 | 70.1 | 83.7 | 68.2 | 72.9 | 64.3 | 69.7 | 80.0 |
| NoisyCXR | 31.4 | 32.0 | 32.7 | 45.5 | 43.3 | 23.7 | 24.3 | 23.5 | 24.5 | 17.0 | 27.5 | 29.5 | 27.4 | 27.8 | 23.4 | 35.7 | 38.7 | 36.3 | 38.3 | 34.5 | 22.1 | 23.4 | 24.8 | 24.5 | 22.9 |
| Credit Fraud | 99.9 | 99.9 | 99.7 | 99.7 | 99.9 | 99.7 | 99.7 | 99.7 | 99.7 | 99.7 | 99.8 | 99.7 | 99.7 | 99.7 | 99.7 | 98.9 | 0.0 | 99.4 | 99.9 | 0.0 | 99.7 | 99.7 | 99.7 | 0.0 | 99.7 |
| Adult | 83.1 | 83.2 | 83.4 | 83.5 | 83.4 | 71.8 | 70.4 | 78.7 | 76.6 | 66.2 | 80.9 | 80.1 | 80.1 | 68.6 | 79.7 | 80.4 | 81.5 | 82.6 | 82.7 | 75.8 | 76.8 | 66.1 | 66.6 | 69.6 | 66.1 |
| Dry Bean | 91.8 | 90.3 | 91.8 | 44.7 | 90.0 | 92.1 | 88.9 | 91.2 | 50.0 | 90.7 | 72.9 | 76.5 | 81.7 | 49.4 | 80.3 | 91.6 | 91.9 | 91.6 | 52.6 | 92.1 | 79.7 | 58.6 | 63.4 | 50.2 | 71.7 |
| Car Evaluation | 95.2 | 97.7 | 97.0 | 57.6 | 97.7 | 75.0 | 77.2 | 78.4 | 79.1 | 68.7 | 81.0 | 83.9 | 81.3 | 76.9 | 84.6 | 86.2 | 83.2 | 77.7 | 57.6 | 83.2 | 75.5 | 73.5 | 77.1 | 57.6 | 74.6 |
| Mushrooms | 100 | 99.8 | 100 | 100 | 100 | 91.0 | 91.9 | 88.1 | 48.5 | 95.5 | 90.9 | 86.3 | 92.2 | 85.3 | 75.1 | 99.8 | 100 | 99.9 | 99.7 | 100 | 76.0 | 86.5 | 82.4 | 37.0 | 80.4 |
| COMPAS | 67.7 | 68.3 | 67.0 | 61.3 | 67.0 | 59.6 | 60.3 | 63.2 | 62.3 | 58.7 | 63.6 | 52.8 | 61.6 | 60.0 | 59.6 | 68.0 | 63.0 | 61.4 | 36.8 | 65.5 | 28.4 | 35.1 | 54.7 | 38.5 | 47.8 |
| Crop | 0.3 | 0.3 | 0.3 | 0.3 | 0.3 | 0.3 | 0.3 | 0.3 | 0.3 | 0.3 | 0.3 | 0.3 | 0.3 | 0.3 | 0.3 | 0.3 | 0.3 | 0.3 | 0.3 | 0.3 | 0.3 | 0.3 | 0.3 | 0.3 | 0.3 |
| Electric Devices | 9.5 | 9.5 | 9.5 | 9.5 | 9.5 | 9.5 | 10.3 | 9.5 | 9.5 | 9.5 | 9.5 | 9.5 | 9.5 | 9.5 | 9.5 | 9.5 | 9.5 | 10.3 | 10.3 | 9.5 | 1.4 | 1.4 | 1.4 | 1.4 | 1.4 |
| MIT-BIH | 66.2 | 65.8 | 69.8 | 22.2 | 64.7 | 42.8 | 40.9 | 60.7 | 33.2 | 50.9 | 43.5 | 43.8 | 44.8 | 22.7 | 46.8 | 51.6 | 56.3 | 59.1 | 35.2 | 66.3 | 38.7 | 40.1 | 55.7 | 38.9 | 38.5 |
| PenDigits | 2.0 | 2.0 | 2.0 | 2.0 | 2.0 | 2.0 | 2.0 | 2.0 | 2.0 | 2.0 | 1.7 | 2.0 | 2.0 | 2.0 | 2.0 | 1.7 | 1.7 | 1.7 | 1.7 | 1.7 | 2.0 | 2.0 | 2.0 | 2.0 | 2.0 |
| WhaleCalls | 33.3 | 33.3 | 33.3 | 33.3 | 33.3 | 33.3 | 33.3 | 33.3 | 33.3 | 33.3 | 33.3 | 33.3 | 33.3 | 33.3 | 33.3 | 33.3 | 33.3 | 33.3 | 33.3 | 33.3 | 33.3 | 33.3 | 33.3 | 33.3 | 33.3 |

Table 21: Impact of label noise and each cleaning method on weighted $F_1$ score of a downstream model for each modality on the test set for noise rate $= 0.4$. The classification models used for images, tabular, and time series datasets are Fast-ViT-T8, TabTransformer, and PatchTST, respectively.

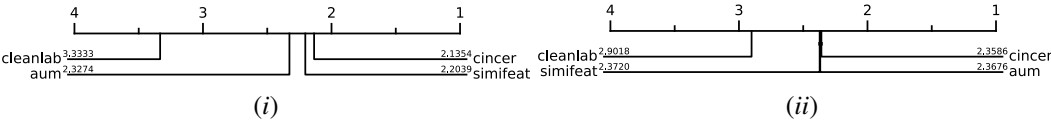

|  | (i) | | (ii) |
|---|---|---|---|

Figure 6: Ranking of cleaning methods across all datasets, base classification models, synthetic noise types, noise rates, random seeds in terms of (i) their ability to identify labeling errors measured using weighted $F_1$, (ii) the weighted $F_1$ of downstream models trained on their cleaned data

### A.8.4 Effects of Cleaning Methods on Data Distribution

| Datasets | No Noise Injected | | | | | Uniform | | | | | Asymmetric | | | | | Class-dependent | | | | | Instance-dependent | | | | |
|---|---|---|---|---|---|---|---|---|---|---|---|---|---|---|---|---|---|---|---|---|---|---|---|---|---|
| | NON | AUM | CIN | CON | SIM | NON | AUM | CIN | CON | SIM | NON | AUM | CIN | CON | SIM | NON | AUM | CIN | CON | SIM | NON | AUM | CIN | CON | SIM |
| CIFAR-10 | 81.1 | 81.1 | 80.5 | 24.0 | 79.4 | 80.1 | 79.8 | 79.6 | 20.2 | 79.9 | 78.3 | 79.0 | 78.1 | 18.9 | 80.0 | 80.3 | 79.9 | 78.9 | 20.6 | 80.7 | 77.5 | 74.5 | 78.0 | 12.0 | 39.2 |
| Clothing-100K | 90.9 | 91.0 | 91.1 | 90.9 | 91.0 | 90.8 | 90.9 | 90.6 | 90.9 | 90.9 | 90.9 | 90.9 | 90.9 | 90.9 | 90.9 | 90.9 | 91.0 | 91.1 | 90.9 | 90.9 | 90.3 | 90.2 | 90.4 | 90.9 | 90.6 |
| NoisyCXR | 65.5 | 65.3 | 64.7 | 10.4 | 64.6 | 63.7 | 64.3 | 64.8 | 10.3 | 64.3 | 64.3 | 64.3 | 65.0 | 11.8 | 64.5 | 65.3 | 65.3 | 65.3 | 12.8 | 65.4 | 62.5 | 62.5 | 63.8 | 11.0 | 62.3 |
| IMDb | 80.8 | 84.5 | 85.2 | 59.2 | 88.4 | 85.7 | 86.1 | 74.1 | 61.2 | 89.0 | 78.7 | 83.1 | 76.1 | 68.6 | 87.0 | 80.9 | 87.2 | 85.5 | 73.1 | 83.4 | 66.7 | 51.9 | 86.3 | 77.8 | 80.8 |
| TweetEval | 65.0 | 66.5 | 72.3 | 69.7 | 71.8 | 61.2 | 79.7 | 67.2 | 66.1 | 78.5 | 81.9 | 69.7 | 73.3 | 73.3 | 73.8 | 74.6 | 47.2 | 26.6 | 12.2 | 12.2 | 66.2 | 66.8 | 77.1 | 78.1 | 74.3 |
| Credit Fraud | 100 | 100 | 99.9 | 99.9 | 100 | 100 | 100 | 100 | 99.9 | 99.9 | 100 | 100 | 100 | 99.9 | 100 | 100 | 100 | 99.9 | 99.9 | 100 | 100 | 100 | 99.9 | 100 | 99.9 |
| Adult | 84.3 | 84.4 | 84.1 | 76.4 | 84.3 | 84.1 | 84.3 | 84.2 | 79.7 | 84.1 | 84.0 | 84.4 | 84.4 | 84.0 | 84.0 | 82.8 | 82.7 | 82.9 | 72.3 | 83.0 | 84.4 | 84.1 | 84.4 | 73.3 | 84.0 |
| Dry Bean | 92.1 | 91.2 | 91.2 | 78.9 | 90.6 | 92.7 | 93.1 | 92.6 | 91.0 | 93.1 | 92.6 | 92.8 | 92.7 | 64.5 | 92.9 | 92.9 | 92.7 | 92.7 | 17.4 | 93.0 | 93.2 | 92.6 | 92.8 | 14.9 | 93.2 |
| Car Evaluation | 91.9 | 89.9 | 77.7 | 57.6 | 89.8 | 87.4 | 92.2 | 87.5 | 57.6 | 90.6 | 89.5 | 83.6 | 89.9 | 57.6 | 85.4 | 86.0 | 79.5 | 59.1 | 57.6 | 87.7 | 89.9 | 84.2 | 78.9 | 73.8 | 86.8 |
| Mushrooms | 99.3 | 100 | 99.3 | 99.8 | 100 | 100 | 100 | 100 | 99.4 | 100 | 100 | 100 | 100 | 100 | 100 | 100 | 100 | 100 | 99.3 | 100 | 99.9 | 99.8 | 100 | 99.9 | 100 |
| COMPAS | 66.7 | 66.7 | 66.4 | 67.1 | 66.5 | 67.9 | 66.3 | 68.9 | 55.1 | 68.7 | 69.6 | 67.5 | 69.1 | 62.2 | 68.5 | 66.3 | 67.6 | 65.3 | 33.7 | 64.7 | 68.0 | 68.3 | 68.5 | 62.4 | 68.3 |
| Crop | 64.1 | 64.9 | 58.3 | 22.5 | 52.9 | 64.5 | 58.7 | 55.2 | 9.1 | 64.6 | 63.5 | 44.3 | 57.2 | 5.2 | 64.8 | 58.1 | 51.4 | 58.7 | 5.2 | 52.7 | 61.8 | 61.5 | 63.1 | 11.0 | 62.0 |
| Electric Devices | 64.6 | 68.6 | 66.9 | 48.3 | 66.4 | 67.1 | 59.1 | 68.2 | 54.2 | 66.0 | 65.0 | 66.8 | 66.0 | 53.5 | 68.0 | 66.9 | 66.4 | 63.8 | 54.0 | 68.1 | 67.0 | 55.3 | 64.4 | 50.9 | 65.5 |
| MIT-BIH | 86.3 | 85.5 | 85.4 | 86.6 | 84.2 | 86.2 | 85.5 | 86.2 | 84.2 | 87.1 | 82.6 | 84.4 | 84.0 | 85.3 | 82.2 | 86.3 | 85.9 | 85.5 | 85.0 | 85.1 | 85.5 | 86.2 | 84.6 | 85.9 | 85.0 |
| PenDigits | 96.6 | 97.8 | 95.3 | 88.7 | 96.6 | 88.2 | 96.5 | 97.0 | 23.8 | 98.4 | 95.9 | 93.2 | 98.1 | 37.0 | 97.5 | 94.8 | 97.9 | 96.7 | 15.7 | 97.3 | 97.7 | 97.3 | 93.6 | 24.0 | 96.3 |
| WhaleCalls | 96.1 | 36.1 | 85.0 | 78.3 | 92.5 | 90.7 | 95.7 | 82.3 | 82.4 | 63.8 | 95.8 | 94.4 | 84.4 | 78.4 | 74.6 | 92.7 | 91.8 | 86.7 | 73.5 | 80.8 | 87.9 | 89.2 | 81.2 | 75.7 | 91.4 |

Table 22: Impact of label noise and each cleaning method on weighted $F_1$ score of a downstream model for each modality on the test set for noise rate = 0.02. The classification models used for images, text, tabular, and time series datasets are ResNet-18, `all-distilroberta-v1`, Multi-layer perception (random seed 42), and ResNet-1D, respectively.

| Datasets | No Noise Injected | | | | | Uniform | | | | | Asymmetric | | | | | Class-dependent | | | | | Instance-dependent | | | | |
|---|---|---|---|---|---|---|---|---|---|---|---|---|---|---|---|---|---|---|---|---|---|---|---|---|---|
| | NON | AUM | CIN | CON | SIM | NON | AUM | CIN | CON | SIM | NON | AUM | CIN | CON | SIM | NON | AUM | CIN | CON | SIM | NON | AUM | CIN | CON | SIM |
| CIFAR-10 | 80.3 | 79.9 | 80.1 | 52.6 | 80.3 | 79.6 | 80.1 | 80.6 | 64.8 | 81.3 | 81.0 | 80.5 | 80.3 | 63.9 | 79.8 | 74.9 | 76.6 | 77.0 | 56.9 | 63.4 | 76.1 | 78.9 | 79.4 | 59.6 | 79.4 |
| Clothing-100K | 91.0 | 90.5 | 90.3 | 90.9 | 90.5 | 90.6 | 90.9 | 90.9 | 90.9 | 90.5 | 90.9 | 90.5 | 90.9 | 90.9 | 90.5 | 80.1 | 71.3 | 71.8 | 90.9 | 89.3 | 88.2 | 89.3 | 90.1 | 90.9 | 89.3 |
| NoisyCXR | 63.4 | 65.0 | 65.4 | 19.6 | 63.3 | 62.6 | 62.8 | 65.3 | 13.2 | 63.2 | 63.1 | 63.2 | 64.2 | 2.1 | 62.4 | 61.7 | 60.2 | 65.9 | 8.5 | 61.2 | 61.3 | 60.2 | 61.8 | 12.4 | 61.5 |
| IMDb | 89.1 | 90.6 | 93.1 | 80.0 | 92.2 | 92.4 | 83.4 | 91.2 | 91.5 | 92.3 | 90.5 | 92.2 | 91.4 | 88.8 | 90.9 | 93.0 | 85.9 | 92.6 | 91.2 | 92.6 | 90.3 | 91.7 | 91.1 | 88.3 | 84.4 |
| TweetEval | 82.1 | 80.7 | 81.9 | 60.4 | 81.8 | 81.8 | 82.1 | 82.9 | 75.1 | 80.1 | 81.8 | 78.8 | 83.2 | 81.9 | 82.2 | 84.0 | 80.9 | 82.1 | 80.8 | 80.5 | 80.1 | 79.5 | 80.3 | 81.1 | 81.2 |
| Credit Fraud | 99.9 | 99.9 | 99.9 | 100 | 99.9 | 100 | 99.9 | 100 | 99.9 | 100 | 100 | 100 | 99.9 | 99.7 | 99.9 | 100 | 99.9 | 99.9 | 100 | 99.9 | 100 | 99.9 | 100 | 99.9 | 100 |
| Adult | 84.6 | 84.2 | 84.5 | 81.5 | 84.4 | 84.4 | 84.1 | 84.1 | 79.9 | 84.3 | 84.1 | 84.3 | 84.4 | 76.0 | 84.3 | 83.0 | 83.3 | 83.2 | 82.4 | 83.0 | 84.1 | 84.1 | 84.2 | 74.8 | 84.2 |
| Dry Bean | 91.6 | 91.0 | 90.5 | 32.4 | 91.4 | 91.9 | 91.1 | 91.3 | 74.3 | 92.2 | 91.3 | 91.8 | 91.9 | 73.3 | 92.4 | 90.4 | 91.1 | 90.1 | 13.8 | 91.8 | 91.2 | 91.2 | 91.1 | 59.1 | 88.8 |
| Car Evaluation | 93.9 | 89.9 | 85.5 | 57.6 | 87.8 | 86.0 | 86.6 | 84.0 | 80.8 | 87.4 | 85.0 | 87.1 | 64.0 | 57.6 | 82.3 | 76.9 | 73.1 | 76.6 | 57.6 | 77.7 | 85.6 | 85.5 | 64.2 | 57.6 | 82.3 |
| Mushrooms | 100 | 100 | 99.3 | 99.7 | 99.3 | 99.9 | 100 | 99.6 | 100 | 99.9 | 99.9 | 99.8 | 100 | 75.6 | 100 | 99.0 | 100 | 100 | 99.9 | 100 | 99.9 | 99.9 | 100 | 99.9 | 100 |
| COMPAS | 67.5 | 67.1 | 64.5 | 60.4 | 66.1 | 66.4 | 65.7 | 68.1 | 38.5 | 65.8 | 66.5 | 66.8 | 66.9 | 39.4 | 66.9 | 65.9 | 63.1 | 65.6 | 28.4 | 66.3 | 68.5 | 65.9 | 67.2 | 38.5 | 68.0 |
| Crop | 52.8 | 50.7 | 47.9 | 2.2 | 60.1 | 40.3 | 55.9 | 50.9 | 1.0 | 58.8 | 52.3 | 39.7 | 49.7 | 4.0 | 49.5 | 41.8 | 39.8 | 36.5 | 5.6 | 30.8 | 15.3 | 46.0 | 53.4 | 4.7 | 54.0 |
| Electric Devices | 61.8 | 65.8 | 67.6 | 31.5 | 64.1 | 49.6 | 64.0 | 63.6 | 21.4 | 64.4 | 63.0 | 56.7 | 60.7 | 10.3 | 63.2 | 46.0 | 59.1 | 55.9 | 33.8 | 54.5 | 64.0 | 65.8 | 56.1 | 21.0 | 64.1 |
| MIT-BIH | 65.7 | 44.1 | 88.5 | 58.4 | 68.6 | 75.0 | 73.1 | 87.2 | 65.9 | 84.5 | 78.2 | 46.4 | 65.7 | 43.2 | 77.2 | 82.6 | 88.5 | 73.6 | 16.6 | 88.9 | 55.1 | 67.4 | 83.3 | 22.5 | 83.6 |
| PenDigits | 95.8 | 95.7 | 95.6 | 28.0 | 95.0 | 94.7 | 94.8 | 95.4 | 33.9 | 95.0 | 99.3 | 95.1 | 93.9 | 24.8 | 92.0 | 96.2 | 94.4 | 96.7 | 38.5 | 93.4 | 91.6 | 95.1 | 93.0 | 13.9 | 94.9 |
| WhaleCalls | 75.1 | 33.3 | 34.2 | 46.4 | 33.3 | 33.6 | 33.3 | 34.6 | 33.3 | 40.7 | 33.4 | 33.6 | 33.4 | 33.3 | 33.4 | 33.4 | 33.3 | 77.1 | 33.3 | 33.3 | 42.9 | 33.5 | 71.1 | 33.3 | 33.4 |

Table 23: Impact of label noise and each cleaning method on weighted $F_1$ score of a downstream model for each modality on the test set for noise rate = 0.02. The classification models used for images, text, tabular, and time series datasets are MobileNet-v2, `all-MiniLM-L6-v2`, Multi-layer perception (random seed 43), and Fully convolutional network, respectively.

| Datasets | No Noise Injected | | | | | Uniform | | | | | Asymmetric | | | | | Class-dependent | | | | | Instance-dependent | | | | |
|---|---|---|---|---|---|---|---|---|---|---|---|---|---|---|---|---|---|---|---|---|---|---|---|---|---|
| | NON | AUM | CIN | CON | SIM | NON | AUM | CIN | CON | SIM | NON | AUM | CIN | CON | SIM | NON | AUM | CIN | CON | SIM | NON | AUM | CIN | CON | SIM |
| CIFAR-10 | 61.5 | 61.4 | 58.4 | 61.5 | 60.6 | 58.8 | 59.5 | 57.5 | 58.4 | 58.5 | 60.9 | 58.5 | 57.5 | 59.7 | 58.1 | 57.7 | 58.2 | 54.8 | 56.9 | 58.6 | 55.2 | 55.6 | 56.2 | 54.5 | 55.8 |
| Clothing-100K | 90.9 | 90.6 | 90.2 | 90.7 | 90.8 | 90.5 | 90.1 | 90.8 | 90.8 | 90.3 | 89.1 | 90.8 | 90.5 | 88.0 | 89.5 | 72.1 | 88.5 | 84.8 | 70.1 | 83.7 | 89.9 | 89.5 | 80.8 | 90.1 | 88.8 |
| NoisyCXR | 46.2 | 36.2 | 46.6 | 45.5 | 43.3 | 44.6 | 44.3 | 40.5 | 45.6 | 45.7 | 45.4 | 43.3 | 41.2 | 45.8 | 40.4 | 35.7 | 38.7 | 36.3 | 38.3 | 34.5 | 40.4 | 43.0 | 38.8 | 39.7 | 40.9 |
| Credit Fraud | 99.9 | 99.9 | 99.7 | 99.9 | 99.9 | 100 | 99.9 | 99.9 | 99.7 | 99.9 | 99.7 | 99.9 | 99.9 | 99.8 | 100 | 98.9 | 0.0 | 99.4 | 99.9 | 0.0 | 99.9 | 99.8 | 99.9 | 99.5 | 99.7 |
| Adult | 83.1 | 83.2 | 83.4 | 83.5 | 83.4 | 82.6 | 82.6 | 83.4 | 83.5 | 83.7 | 83.8 | 83.5 | 82.8 | 81.8 | 82.9 | 80.4 | 81.5 | 82.6 | 82.7 | 75.8 | 83.8 | 78.9 | 83.1 | 80.7 | 83.6 |
| Dry Bean | 91.8 | 90.3 | 91.8 | 44.7 | 90.0 | 92.9 | 91.3 | 92.1 | 45.9 | 91.9 | 91.3 | 91.9 | 91.1 | 56.0 | 91.6 | 91.6 | 91.9 | 91.6 | 52.6 | 92.1 | 91.2 | 91.9 | 92.7 | 48.1 | 92.2 |
| Car Evaluation | 95.2 | 97.7 | 97.0 | 57.6 | 97.7 | 97.2 | 94.7 | 91.9 | 67.2 | 96.6 | 96.5 | 97.1 | 82.2 | 73.3 | 97.0 | 86.2 | 83.2 | 77.7 | 57.6 | 83.2 | 96.5 | 96.6 | 94.1 | 61.7 | 90.5 |
| Mushrooms | 100 | 99.8 | 100 | 100 | 100 | 99.8 | 100 | 100 | 98.4 | 99.4 | 99.8 | 100 | 100 | 99.7 | 100 | 99.8 | 100 | 99.9 | 99.7 | 100 | 100 | 98.7 | 99.9 | 99.7 | 99.7 |
| COMPAS | 67.7 | 68.3 | 67.0 | 61.3 | 67.0 | 67.8 | 66.2 | 68.8 | 59.8 | 66.1 | 66.7 | 63.7 | 69.0 | 66.0 | 64.3 | 68.0 | 63.0 | 61.4 | 36.8 | 65.5 | 66.4 | 69.1 | 66.3 | 62.3 | 68.6 |
| Crop | 0.3 | 0.3 | 0.3 | 0.3 | 0.3 | 0.3 | 0.3 | 0.3 | 0.3 | 0.3 | 0.3 | 0.3 | 0.3 | 0.3 | 0.3 | 0.3 | 0.3 | 0.3 | 0.3 | 0.3 | 0.3 | 0.3 | 0.3 | 0.3 | 0.3 |
| Electric Devices | 9.5 | 9.5 | 9.5 | 9.5 | 9.5 | 9.5 | 9.5 | 9.5 | 9.5 | 9.5 | 9.5 | 9.5 | 9.5 | 10.3 | 9.5 | 9.5 | 9.5 | 10.3 | 10.3 | 9.5 | 9.5 | 9.5 | 9.5 | 9.5 | 9.5 |
| MIT-BIH | 66.2 | 65.8 | 69.8 | 22.2 | 64.7 | 60.9 | 64.3 | 73.0 | 52.3 | 64.7 | 67.0 | 61.9 | 72.3 | 34.0 | 66.9 | 51.6 | 56.3 | 59.1 | 35.2 | 66.3 | 58.4 | 65.9 | 71.5 | 23.3 | 60.2 |
| PenDigits | 2.0 | 2.0 | 2.0 | 2.0 | 2.0 | 2.0 | 2.0 | 1.7 | 1.7 | 2.0 | 2.0 | 2.0 | 2.0 | 2.0 | 2.0 | 1.7 | 1.7 | 1.7 | 1.7 | 1.7 | 2.0 | 2.0 | 2.0 | 2.0 | 2.0 |
| WhaleCalls | 33.3 | 33.3 | 33.3 | 33.3 | 33.3 | 33.3 | 33.3 | 33.3 | 33.3 | 33.3 | 33.3 | 33.3 | 33.3 | 33.3 | 33.3 | 33.3 | 33.3 | 33.3 | 33.3 | 33.3 | 33.3 | 33.3 | 33.3 | 33.3 | 33.3 |

Table 24: Impact of label noise and each cleaning method on weighted $F_1$ score of a downstream model for each modality on the test set for noise rate = 0.02. The classification models used for images, tabular, and time series datasets are Fast-ViT-T8, TabTransformer, and PatchTST, respectively.

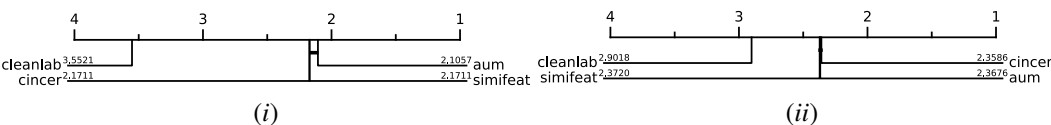

Figure 7: Ranking of cleaning methods across all datasets, base classification models, synthetic noise types, noise rates, random seeds in terms of (*i*) their ability to identify labeling errors measured using *accuracy*, (*ii*) the *accuracy* of downstream models trained on their cleaned data.

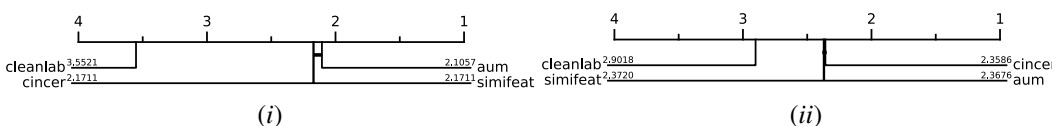

Figure 8: Ranking of synthetic noise types by their ability to impact the (*i*) performance of cleaning methods, (*ii*) *weighted* $F_1$ of downstream models trained on cleaned datasets.

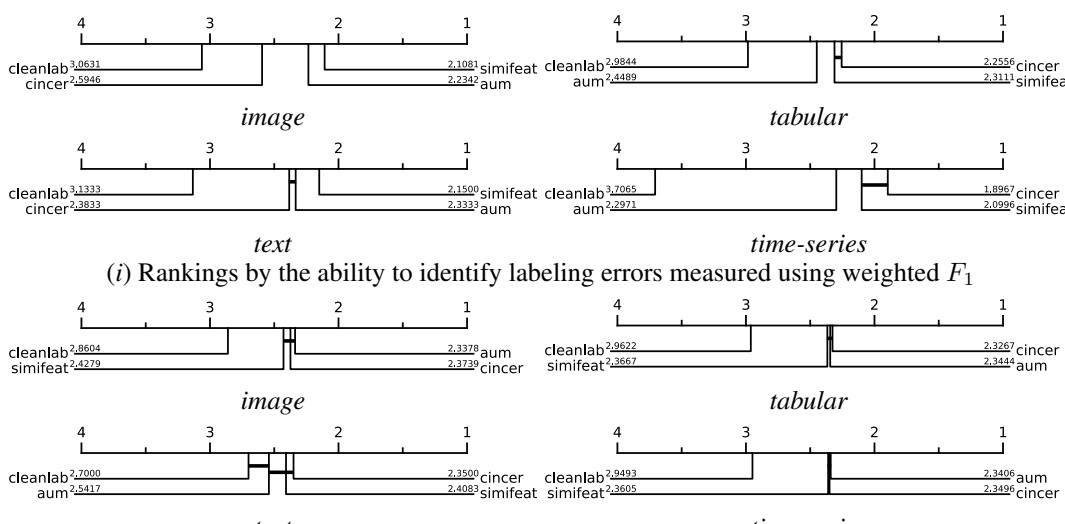

(*i*) Rankings by the ability to identify labeling errors measured using weighted $F_1$

(*ii*) Rankings by the weighted $F_1$ of downstream models trained on their cleaned data

Figure 9: Rankings of cleaning methods segmented by data modality.

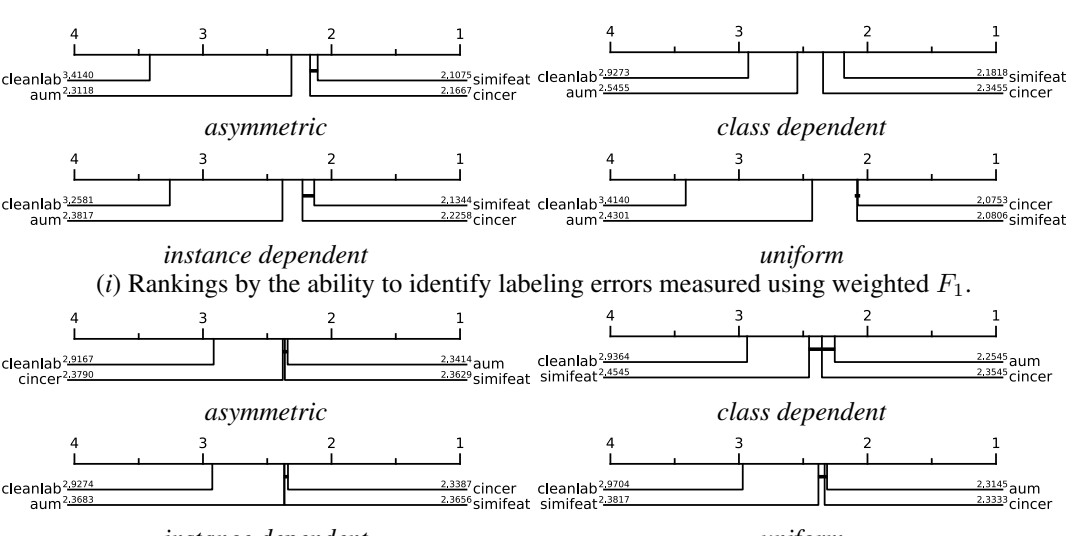

(*i*) Rankings by the ability to identify labeling errors measured using weighted $F_1$.

(*ii*) Rankings by the weighted $F_1$ of downstream models trained on their cleaned data.

Figure 10: Rankings of cleaning methods segmented by synthetic noise type.

## B   Sources and Licenses

All experimentation datacards to reproduce results can be found here.

| Datasets | AUM | CIN | CON | SIM |
|---|---|---|---|---|
| **Average** | 0.011 | 0.133 | 0.574 | 0.061 |
| **Std Dev** | 0.001 | 0.115 | 0.248 | 0.038 |
| **Median** | 0.011 | 0.095 | 0.478 | 0.057 |

Table 25: Proportion of data points cleaned by each cleaning method, averaged over noise type, noise rate, random seed(s), and downstream model architecture and datasets. Confident Learning removes 57% of training data points on average, explaining its poor performance. All other methods remove <13% data points. It is probably correct that for confident learning the downstream models are not seeing enough data or trained long enough for models to converge. But we believe that this might be a problem of the cleaning method, more than the experiment design.

| Datasets | AUM | CIN | CON | SIM |
|---|---|---|---|---|
| **Average** | 0.002 | 0.011 | 0.047 | 0.003 |
| **Std Dev** | 0.003 | 0.011 | 0.036 | 0.004 |
| **Median** | 0.000 | 0.008 | 0.053 | 0.001 |

Table 26: Difference in proportion of data points belonging to the minority class before and after label cleaning, averaged over noise type, noise rate, random seed(s), and downstream model architecture and datasets. Barring Confident Learning, the other cleaning methods do not have a major impact on class imbalance.

| Cleaning Methods and Datasets | Reference | License | Source |
|---|---|---|---|
| SimiFeat | [34] | CC BY-NC 4.0 | Link |
| AUM | [24] | MIT | Link |
| CINCER | [30] | MIT | Link |
| Confident Learning | [23] | GNU AGPL v3.0 | Link |
| CIFAR-10N | [49] | CC BY-NC 4.0 | Link |
| CIFAR-10H | [16] | CC BY-NC-SA 4.0 | Link |
| Clothing-100K | [51, 24] | Non-commercial research and educational purposes | Link 1, Link 2[7] |
| NoisyCXR | [52] | Unrestricted use [8] | Link |
| IMDb | [53] | MIT | Link |
| TweetEval | [54] | MIT | Link |
| Credit Card Fraud Detection | [55] | DbCL v1.0 | Link |
| Adult | [58] | CC BY-NC 4.0 | Link |
| Dry Bean | [59] | CC BY-NC 4.0 | Link |
| Car Evaluation | [60] | CC BY-NC 4.0 | Link |
| Mushroom | [62] | CC BY-NC 4.0 | Link |
| COMPAS | [63] | DbCL v1.0 | Link |
| Crop | [64] | GNU GPL v3.0 | Link |
| ElectricDevices | [65] | GNU GPL v3.0 | Link |
| MIT-BIH | [66] | ODC-By v1.0 | Link |
| PenDigits | [67] | CC BY-NC 4.0 | Link |
| WhaleCalls | [68] | Copyright © 2011 by Cornell University and Cornell Research Foundation, Inc. [9] | Link |

Table 27: Licenses for cleaning methods and datasets.

---

[7] Dataset can be downloaded by contacting `tong.xiao.work@gmail.com`

[8] We acknowledge the NIH Clinical Center (`clinicalcenter.nih.gov`) and National Library of Medicine `www.nlm.nih.gov`) for providing this dataset.

[9] Data courtesy of and copyrighted by Cornell University and the Cornell Research Foundation.