# OpenReview forum: "AQuA: A Benchmarking Tool for Label Quality Assessment"
_NeurIPS.cc/2023/Track/Datasets_and_Benchmarks — NeurIPS 2023 Datasets and Benchmarks Poster_

### Official Review · Reviewer_RN59 · 2023-07-20
**Good work**

**Rating:** 6
**Confidence:** 4
**Correctness:** Yes
**Clarity:** Yes

**Strengths:**

The submission has several strengths. Firstly, AQuA is a significant contribution to the research community as it provides a standardized evaluation framework, benchmark datasets, and evaluation metrics for assessing the performance of label error detection models. This will enable researchers to compare the performance of different models and identify the most effective methods for detecting and correcting labeling errors in their data.

Secondly, the proposed design space and benchmark design are well thought out and provide a comprehensive approach to evaluating label error detection models. The authors have conducted a large-scale experiment (>1000 unique experiments) to demonstrate the efficacy of AQuA in benchmarking machine learning models in the presence of label noise.

Thirdly, the paper discusses the limitations of the proposed approach and provides recommendations for future work. This demonstrates the authors' commitment to improving the quality of research in this area and ensuring that the proposed approach is continually refined and improved.

Finally, the authors have discussed the potential negative societal impacts of their work and have ensured that their paper conforms to ethics review guidelines. This demonstrates their commitment to ethical and responsible research practices.

Overall, the submission is of high quality and has significant relevance to the broader research community. The proposed approach has the potential to improve the accuracy and reliability of machine learning models, which has important implications for a wide range of applications, including healthcare, finance, and transportation.

**Additional Feedback:**

no

**Documentation:**

Yes

**Limitations:**

The authors have discussed the limitations of their work and have provided recommendations for future work. They have also discussed the potential negative societal impacts of their work and have ensured that their paper conforms to ethics review guidelines. However, there are some areas where the authors could provide more detailed recommendations for mitigating the risks associated with their work.

For example, the authors could discuss how to ensure that label error detection models are not used to perpetuate existing biases in the data. They could also discuss how to ensure that the models are not used to discriminate against certain groups of people. Additionally, the authors could provide more detailed recommendations for ensuring that the benchmark datasets are representative of the broader population and are not biased in any way.

Overall, the authors have done a good job of addressing the limitations and potential negative societal impact of their work. However, there is always room for improvement, and the authors could provide more detailed recommendations for mitigating the risks associated with their work.

**Opportunities For Improvement:**

While the submission has several strengths, there are also some limitations that could be addressed in future work.

Firstly, the benchmark datasets included in AQuA are limited to only four data modalities: image, text, time-series, and tabular. While these are common data modalities, there are other types of data that could be included in future versions of AQuA. For example, audio data is becoming increasingly important in many applications, and it would be valuable to include audio datasets in AQuA.

Secondly, the paper does not provide a detailed analysis of the performance of different label error detection models on the benchmark datasets. While the authors have conducted a large-scale experiment to demonstrate the efficacy of AQuA, a more detailed analysis of the results would be valuable. For example, it would be useful to know which models perform best on different types of datasets and under different conditions.

Thirdly, the paper does not discuss the potential biases that may be present in the benchmark datasets. It is possible that the datasets included in AQuA are biased in some way, which could affect the performance of label error detection models. Future work could address this limitation by carefully selecting datasets that are representative of the broader population.

Finally, while the authors have discussed the potential negative societal impacts of their work, they could have provided more detailed recommendations for mitigating these risks. For example, they could have discussed how to ensure that label error detection models are not used to perpetuate existing biases in the data.

Overall, while the submission is of high quality, there are opportunities for improvement in future work. Addressing these limitations could further enhance the value of AQuA as a benchmarking tool for label quality assessment in machine learning.

**Relation To Prior Work:**

Yes

**Summary And Contributions:**

The submission introduces AQuA, a benchmarking tool for label quality assessment in machine learning. AQuA provides a comprehensive environment for evaluating methods that detect and correct labeling errors in data. The paper proposes a design space and benchmark design to aid in the development and evaluation of label error identification methods. The contributions of AQuA include a standardized evaluation framework, a set of benchmark datasets, and a suite of evaluation metrics for assessing the performance of label error detection models. The paper also discusses the limitations of the proposed approach and provides recommendations for future work.

---

> ### Author Response · Authors · 2023-08-25
> **Thanks for your feedback and time! Summary of changes and clarifications.**
>
> Dear Reviewer RN59,
>
> Thank you so much for your time and feedback! We are making several changes to the paper based on your feedback:
>
> 1. **Variety of Benchmark Datasets**: Based on your suggestions, we are implementing the [AudioSet](https://research.google.com/audioset/dataset/index.html) dataset into AQuA. While we agree that AQuA is only limited to four data modalities, we would like to stress that ours is the first study to include multiple datasets from 4 prevalent modalities. A majority of previous studies conducted experiments on a small number of datasets in a single (image) modality.
> 2. **Detailed Analysis of Experimental Results**: Due to space constraints, we are adding further insights from our experiments in the appendix of the paper. For example, we are analyzing the performance of cleaning methods in low and high noise settings, the impact of class imbalance on detecting labeling errors, etc.
> 3. **Limitations, Biases and Societal Impacts**: We completely agree with you! Based on your suggestions, we are adding a longer discussion on the limitations of our work as well as avenues for future work. This would include some of the limitations of our experimental settings and benchmark design as pointed out by the reviewers. We would also discuss how label error detection models might perpetuate existing biases and impact the fairness of models. In fact, we included the Adult dataset, which is frequently used in the fairness literature, in AQuA to evaluate the impact of label errors on the fairness of models.

---

### Official Review · Reviewer_AcXH · 2023-07-21
**An interesting benchmark with broad range of application**

**Rating:** 6
**Confidence:** 4
**Correctness:** Yes
**Clarity:** Very Clear

**Strengths:**

Unified Evaluation. this benchmark provides a standardized and objective evaluation platform for various label error detection models.

The motivation of this paper is very clear.  The paper addresses the prevalent issue of erroneous labels in widely-used datasets, which has significant implications for model generalization and evaluation. By focusing on real-world challenges, the paper's contributions have practical relevance for ML practitioners and researchers.

I appreciate the effort of the authors for adding a section of Design Space for Label Error Detection Models. The introduction of a design space allows for concrete delineation of design choices for label error detection models. This framework facilitates the understanding and comparison of various approaches, aiding in the selection of appropriate methods to improve label quality.

The benchmark contains datasets from multiple domains and includes baselines from recent works. The wide range of datasets and state-of-the-art baselines make this benchmark a valuable resource for researchers and practitioners in the field.





**Additional Feedback:**

Some additional questions:

Does the label imbalance issue have a large effect on the performance of different noisy label detection methods? It would be better to give more explanation on this issue.

What is the relation between this work and so-called *weakly-supervised learning*, since the label derived from weak supervision are also noisy?

**Documentation:**

Good

**Limitations:**

Yes

**Opportunities For Improvement:**

It seems that the backbone model used in this work is not very strong. For example, have the authors consider to use Vision Transformer for images, or RoBERTa-Large for text? I am not sure about the performance of these baselines when using a stronger neural model.

The task selected in this paper is not overly difficult. For example, the text/tabular datasets are very easy. These might somehow explain why the model w/o a label cleaning module can also achieve good performance. It would be better to add more challenging datasets with more classes.

It is somehow not clear how to perform hyperparameter selection. Are additional clean labels required for this step?

It would be interesting to explore how to denoise the noisy label generated from a foundation model such as CLIP.




**Relation To Prior Work:**

Yes

**Summary And Contributions:**

The paper introduces AQuA, a benchmarking environment to evaluate methods for handling label noise in machine learning. It addresses the impact of erroneous labels on model generalization and provides a design space for label error detection models, enabling objective evaluation and aiding ML practitioners in improving label quality.

In my opinion, this work makes a good contribution to the field, as it enable objective evaluation and assist ML practitioners in improving label quality for more reliable machine learning outcomes in the presence of mislabeled data.

---

> ### Author Response · Authors · 2023-08-25
> **Thank you so much for your time and feedback!**
>
> Dear Review AcXH,
>
> Thank you so much for your time and feedback!
>
> ### Opportunities for Improvement
> 1. **Downstream classifiers**: Based on your and reviewer ICgR's feedback, we have expanded the set of downstream classifiers to include state-of-the-art transformer-based models: TabTransformer [1] for tabular data, PatchTST [2] for time-series data, and FastViT-T8 [3] for image data. We are currently running experiments using these models.
> 2. **Difficulty of classification problems**: We agree that some of the classification problems might be easy. Our choice of datasets was partly based on a survey of over 50 papers in this field and frequently used datasets therein. However, we would also like to point out harder classification problems with real-world applications that we have included in the benchmark: e.g., Arrhythmia classification using ECG data on the MIT-BIH dataset, Pneumonia classification using Chest X-rays on the NoisyCXR dataset. Finally, we firmly believe that AQuA's modular design will enable researchers to seamlessly include more datasets in the benchmark.
> 3. **Hyper-parameter tuning**: For SimiFeat and CINCER, we picked hyper-parameter grids from the papers which first proposed these methods. For AUM, we picked the hyper-parameter grid ourselves since the authors did not provide any recommendations. Finally, confident learning did not have any hyper-parameters (see [cleanlab](https://github.com/cleanlab/cleanlab)). For downstream classifiers, we borrowed hyper-parameters from their original implementations. We did not tune hyper-parameters for downstream classifiers so that differences in their performance could be directly attributed to the cleaning methods, rather than differences in their own hyper-parameters. For each cleaning method and downstream classification model, for a given dataset,  hyper-parameters were chosen based on model performance on the observed training set. Once chosen, hyper-parameters were frozen for all noise experiments (noise type + noise rate).
> 4. Noisy labels generated by CLIP: We believe that we can use methods in AQuA off-the-shelf to detect noisy labels originating from foundation models such as CLIP. However, there may be better ways to identify labeling errors by leveraging their structure and model weights. Such methods are currently beyond the scope of our work.
>
> ### Questions
> 1. **Impact of label imbalance**: This is an interesting question. We are currently examining differences between the performance of cleaning methods in imbalanced and balanced datasets and will share our findings in the paper.
> 2. **Relation with Weakly Supervised Learning**: AQuA serves two purposes: (1) as a benchmarking tool to evaluate methods that identify labeling errors, (2) and generally as a tool to identify labeling errors in a dataset and choose an appropriate cleaning method. Weakly supervised learning is a class of methods that learn from imperfect and weak sources of supervision to label datasets (see [1] and [2] as examples). The labels arising from these methods are indeed noisy. Methods in AQuA can therefore be used to clean datasets labeled using weakly supervised methods.
>
> **References**
>
> [1] Zhang, Jieyu, et al. "A survey on programmatic weak supervision." arXiv preprint arXiv:2202.05433 (2022).
>
> [2] Goswami, Mononito, Benedikt Boecking, and Artur Dubrawski. "Weak supervision for affordable modeling of electrocardiogram data." AMIA Annual Symposium Proceedings. Vol. 2021. American Medical Informatics Association, 2021.

---

> > ### Comment · Reviewer_AcXH · 2023-08-31
> >
> > Thanks for your response! I have no further questions and would like to keep my current score.

---

### Official Review · Reviewer_Q72y · 2023-07-25
**AQuA benchmark proposed for rigorous evaluation of methods dealing with Label Quality**

**Rating:** 7
**Confidence:** 2
**Clarity:** Yes

**Strengths:**

1. **Comprehensive Problem Addressing:** The paper tackles the significant issue of labeling errors in widely used ML datasets, emphasizing their impact on model generalization and evaluation.

2. **Novel Benchmark Framework:** The introduction of the Automated Label Quality Assessment (AQuA) benchmark provides a valuable contribution to the field, enabling rigorous evaluation of methods dealing with label noise.

3. **Practical Relevance:** The paper's practical implications are noteworthy, as it empowers ML practitioners to select appropriate label cleaning tools for improving model performance when faced with mislabeled data.


**Additional Feedback:**

None

**Correctness:**

Yes, the evaluation methods and experiment design appropriate and performed correctly

**Documentation:**

Yes

**Limitations:**

I am not familiar with this domain, but I think it could be better if the author discuss more about the label noise in other settings such as multimodal and regression.

**Opportunities For Improvement:**

it is better to provide some use cases that helps the reader the better understanding the tool's ability.

**Relation To Prior Work:**

Yes

**Summary And Contributions:**


This paper addresses the critical issue of labeling errors in commonly used datasets for training and evaluating machine learning models. It highlights the impact of erroneous labels on model generalization and evaluation, while pointing out the lack of a comprehensive benchmark to evaluate existing methods. The proposed Automated Label Quality Assessment (AQuA) benchmark framework aims to fill this gap, offering a rigorous evaluation of machine learning methods dealing with label noise. The paper demonstrates the necessity of this benchmark through a large-scale experiment and emphasizes its potential to aid ML practitioners in selecting appropriate label cleaning tools for improved model performance when dealing with mislabeled data.

---

> ### Author Response · Authors · 2023-08-25
> **Thank you so much for your time and feedback! Summary of changes and clarifications.**
>
> Dear Reviewer Q72y,
>
> Thank you for your comments and feedback on our paper. Based on your recommendations, we are incorporating the following changes into our paper and AQUA’s documentation.
>
> **Better documentation**: Based on your suggestions, we are making a Jupyter notebook that would walk users through all of AQuA’s functionalities, from loading a dataset, identifying labeling errors using a cleaning method, to training a classifier and evaluating it. This will help researchers better understand how to best use AQuA.
>
> **Noise in Other Settings**: Currently, we are considering label noise in multi-class single-label classification problems. While other types of noise are beyond the scope of this work, we agree that multi-annotator, multi-class multi-label, and noise in regression problems are exciting avenues of future work. We believe that AQuA’s modular design will enable researchers to experiment with both multi-annotator and multi-class multi-label classification problems easily.

---

### Official Review · Reviewer_iCgr · 2023-07-28
**Perhaps the devil is in the details about how the experiments were run**

**Rating:** 7
**Confidence:** 4
**Correctness:** 1. There is no mention of a learning …

**Strengths:**

The authors perform experiments with a sufficiently large number of datasets, and across a range of noise injection methods. They do not just measure the ability of the cleaning methods to identify samples with noisy labels, but also measure the impact of the cleaning methods on the performance of downstream models trained from scratch on cleaned versions of the datasets.

By combining the results from two noise levels and two architectures, the main results tables show results with n=4 (though these are not like-for-like repetitions), which is acceptable

**Additional Feedback:**

IMHO, AQuA is a poor choice of name for the benchmark.
> To bridge this gap, we propose the **A**utomated *L*abel **Qu**ality **A**ssessment AQuA

You dropped the most important letter!! It's a benchmark for evaluating label noise detection algorithms, the letter L should be in the name, surely. Is "Automated" supposed to refer to a property of the label noise detection methods which are being evaluated? In this case, the name doesn't scan correctly as there is a word missing. Is "Automated" referring to a property of the benchmark? If it is the later, I'm not sure your codebase has a sufficiently high level of automation to justify this. It is clear that you've coined a backronym (for some reason, targeting a word unrelated to the subject matter), presumably for memorability? But you could have re-ordered the words slightly to make LAQuA and said it is pronounced like lacquer. Or used one of many other words with the letter L in them.

---

In future, please don't resubmit the main paper as part of the supp. materials upload. It adds confusion since it is unclear if something has changed in the main text between the two.

**Clarity:**

It is generally well written.

Table 4 in the appendix is an unmodified duplicate of Table 1.

### Typographic corrections

- L48: "Assessment AQuA" missing comma
- L268: "1280 experiments" I think should instead be 1260 experiments since $5×(4×2+1)×(11×2+6) = 1260$
- L359: [11] "Bert" incorrect casing
- L474: [42] has no journal/venue, not even arXiv.
- L541: [59] DOI field should point to a DOI not a URL of a DOI.
- L565: [66] "mit-bih arrhythmia database" incorrect casing
- L571: [68] "uea" incorrect casing
- L579: [70] "bert" incorrect casing
- L584: [72] "Lstm" incorrect casing
- L870: replace hyphen with em-dash
- L873: "0.1-100 Hz" replace hyphen with en-dash
- L883: "60Hz-250Hz" replace hyphen with en-dash
- L911: "and ~can~ is"
- L964: "observe**d**"

**Documentation:**

The documentation for the code is rather wanting. The github README merely repeats the results given in the paper. The documentation needs to clearly outline all the steps I need to reproduce the experimental results, and to evaluate my own label noise detection algorithm.

Ideally, I would like to have the code structured in such a way that I can write my own label noise detection algorithm in a particular format --- a function that accepts these inputs in this order and produces these outputs --- that I can pass to the benchmarking code and have it run the full suite on the method. From what I can tell from briefly looking through the code, it relies on me having certain directory structures and having files named in a way that matches the name of the methods, so it does not appear to be suitably portable.

I'd also like to have the authors provide a single script that I run once which downloads all the data needed to apply the benchmark (i.e. the contents of all 17 datasets). I should not have to jump through hoops to obtain each of the datasets individually; e.g. following a chain of links to find the download, untar it and find the appropriate dataset within its outputs, email someone to obtain a download link, reformat the dataset structure from the downloaded structure to the structure needed to run the benchmark code.

**Ethics:**

The (Right)WhaleCalls and ElectricDevices datasets are (according to the AQuA paper) not available under a permissive licence. Their use is likely a copyright violation and not legally compliant. Unless the authors can present evidence that these datasets are distributed under an appropriate license, their use is not appropriate and they should be removed from the benchmark entirely.

**Limitations:**

To their credit, the authors do run a large number of experiments. However, there is no analysis or discussion of which experiments can be grouped together to increase their statistical power and which can not. Given that the conclusions about which cleaner performs the best changes depending on whether only image or only text data is considered, does an overall comparison make sense? The samples for the comparison are each dataset+architecture run. In which case, removing some text datasets and adding some new image datasets and architectures would change the distribution of the modalities in the overall data and hence the final conclusions?

The authors use three image datasets which have multiple annotators, but do not make use of the multiple annotations. This limitation is acknowledged in the paper, and the authors say it is left as future work. Confusingly though, the appendix (L957 etc) refers to methods that were not used in the main paper, nor the appendix.
> for datasets with labels from multiple annotators, we implement dissenting label, dissenting worker, and crowd majority...

**Opportunities For Improvement:**

1. L270:
> To ensure that all classes, especially those in minority are weighted equally during evaluation, and due to lack of space, we report the F1 weighted by the support of each class.

This appears to be a contradiction. The macro weighting (not used in the paper) would ensure classes in the minority are weighted the same as classes in the majority. Please justify the use of the support weighting used, or change the results to use macro-weighting instead of support-weighting if that is what you meant to do.


2. L831:
> It contains attributes like age, work class, fnlwgt, education, education number, marital status, occupation, relationship, race, sex,

fnlwgt is not a word and does not mean anything to me. Please describe it instead of merely stating the column name.


3. L983:
> All experimentation datacards to reproduce results can be found here.

This link points to an empty GDrive folder.


4. The main paper spends quite a while motivating the premise, but does not have space to define the 4 synthetic noise injection methods. I think the noise methods used are sufficiently important to include them in the main text.


### Experiments, Datasets and Models

5. The models used do not span the diversity of architectures available for the modalities considered. It would be great to expand the set of architectures applied to the image domain to include ViT-small, which is more different from ResNet-18 than MobileNetV2. The time-series architectures were ResNet and LSTM+FCN, which did not include an RNN-only architecture, nor a transformer-based architecture. Only an MLP was deployed on tabular data, but other NN models for tabular data do exist --- this could be extended with a ResNet and/or the FT-Transformer architecture [(Gorishniy, 2021)](https://arxiv.org/abs/2106.11959) --- but since the task of label cleaning is suitably general, I do not see a reason why tree-based methods such as XGBoost and Random Forest should not be considered, since these are usually more performant on tabular data than MLPs [(Grinsztajn, 2022)](https://arxiv.org/abs/2207.08815), and are also adversely affected by label noise.

6. The paper says three synthetic noise rates (0%, 10% and 40%) are considered, but as one of the rates is 0% really only two noise rates are considered. I think the benchmark would be improved by considering more noise rates, and I think that the benchmark should include an evaluation that has a more realistic level of noise. Most datasets used in practice have noise rates well below 10%. I think adding a 2% noise rate, which is approximately equal logarithmic spacing below 10% as 40% is above it, would make sense, i.e. noise values {2%, 10%, 40%}, but the authors may have better suggestions. (This would be in addition to the "0% synthetic noise level" that, as far as I can tell, is used solely for the "No Noise Injected" results in Table 2.) Adding more noise levels would enable the benchmark to answer the question of whether some cleaners are better with low noise rates, and other cleaners better with higher noise rates.

7. For the model training paradigm, there is no mention of data augmentation. It is okay to not use data augmentation (if that is the experimental paradigm you are using), but in that case you should explicitly state that there is no data augmentation to make sure it is clear. It should be noted that vision models are essentially always trained with data augmentation to reduce overfitting, so deviating from the standard experimental paradigm may impact the applicability of the experimental findings.

8. The authors share the grid used in the hyperparameter search, but not the hyperparameters which were discovered and used in the rest of the experiments. This needs to be added, to ensure replicability.

9. There is no discussion of the number of seeds used for the initial state of the networks, nor which seeds are used, or whether they were held the same between experiments with different cleaners, architectures, etc.

10. There is no discussion of the seeds used for the noise injection methods, whether they were held the same between experiments with different cleaners, architectures, etc.

11. If the noise injection is done in the same way for each experiment, would it not be wise for the authors to distribute a static copy of these noised labels?

12. L959:
> Class-dependent Noise [73]: In this setting, similar classes have a higher probability of being
mislabeled with each other. To this end, for a given dataset, we set the noise transition matrix to be
the confusion matrix of a neural network trained and evaluated on the train and test set of the dataset,
respectively.

This seems overly adversarial. The confusion matrix should surely be evaluated on a held-out validation set instead of the test set that performance will be evaluated on.


### Figures and statistical tests

13. Figure 5: I've never seen a "Critical Difference Diagram" before and I can't imagine I'm the only reader that is unfamiliar with them. Some more description of what is going on in the diagram needs to be given. I assume the thick horizontal bar show groups of items which are not statistically significantly different to each other (as mentioned on L275)? But there are no descriptions of the statistical test being performed in the paper. This is important because, as far as I can tell, the statistical tests being used in these figures are the only tests being used to support the experimental conclusions. Please describe the statistical tests being used to conclude that one cleaning method is better than the others.


### Tables

14. Table 2 and Table 3 show the datasets ordered differently to Table 1, which is confusing for the reader. Please keep the ordering consistent.

15. In Tables 2, 3, 7-14, the F1 scores should be right-aligned instead of centered so that performances less than 10% are more legible. (You can include a period after 100 so it is aligned better without needing to add space for the tenth-place of 100.0, or manually the cells containing 100 centered.)

16. In Tables 2, 3, 7-14, I think it would make sense for the noise categories to be sorted by increasing complexity (as they are described in the appendix) instead of alphabetically. That is to say, Uniform, Asym, Class-dep, Instance-dep.

17. Tables 7-14 do not have the cell highlighting used in Tables 2, 3. Please add this to improve readability (I would assume the highlighting is automatically generated and not manually implemented, so I see no reason not to do it again). Furthermore, it would be helpful to indicate which model performances are not statistically higher than chance level. From inspecting the tables, it seems there are some but it is hard to be sure because I am not aware of the number of classes in each dataset.

18. In Tables 7-14, the caption describes the models used in the order images, text, tabular, and time
series; but this is not the order in which the categories are shown in the table (see point above about the dataset ordering being different to Table 1.

19. Tables 3, and 11-14, do not use the whole width of the page and could have the categories added as an extra column to the left.

20. Tables 8 and 9 are missing horizontal bars separating the categories.

21. The order of Table 2 and Table 3 should be swapped. Table 3 is the ability of the cleaner to detect erroneous labels, whilst Table 2 is the performance of a downstream model after removing the samples identified as erroneous. Since Table 3 measures something which comes earlier in the pipeline than Table 2, it logically should appear before Table 2. This would make it easier for a reader to follow the paper.

22. The difference between the contents of Table 2 and Table 3 is currently not made very clear and should be highlighted better. "Performance of cleaning methods" is a rather vague description that could apply to either set of results.

23. Table 1 could be improved in several ways.
- Columns Modality, Classification Task,  should be left-aligned.
- Number of train and test samples should be two separate columns (and right-aligned). You could save space by writing the numbers to only thousands precision (50k  | 10k \\ 0k | 10k \\ ... 7k | 3k \\ 11k | 2k) removing a bunch of digits. I am perplexed as to how some of the datasets only have one number for both train and test. Please explain why this is, or fix the table to show the actual train and test set sizes separately.
- "# Annotations" is unclear (it's number of annotators per sample, but could be read as the total number of labels). The CIFAR-10H and NoisyCXR values should be written in the same format as each other, e.g. 47--63 and 1--XX.
- A detailed description of the label source is not necessary within this figure. Enough detail to describe the methodology completely is not possible in this space. Leave that to the appendix. The important thing to convey in the table is only whether the annotation source is human or automated. This can be done with a binary check/cross column, which will take up much less space.
- For Sample Size, the default spacing around the times symbols is wider than you need and taking up precious space in the table. Reduce the spacing by adding negative space either side of them with `\!`, i.e. `\!\times\!`.
- There number of classes should be added as a column in the table (removing the need for the beta symbol).
- It would be great to have a measure of class imbalance added to the table as well. The ratio of the size of the largest class to the smallest class, or an entropy-based measurement such as the ratio of the entropy of the class distribution to a uniform distribution over that number of classes would work.

24. There's no standard deviation/error bars shown, though this might not be meaningful to add when the repetitions being averaged over in the tables are dissimilar to each other.

**Relation To Prior Work:**

I think so.

**Summary And Contributions:**

AQuA is a benchmarking tool for methods aiming to detect noisy labels in datasets. The benchmark task is to identify synthetic noise injected into 16 existing datasets spanning several modalities (image, text, tabular, and time series) under four different noise injection strategies (asymmetric, class-dependent, instance-dependent, and uniform) at two different rates (10%, 40%). The authors measure the performance of four existing label noise detection methods on these tasks, and additionally, for each label noise detection method, train models from scratch with the samples detected as noised removed.

**Edit:** The authors have addressed the majority of my concerns, and I have raised my score accordingly.

---

> ### Author Response · Authors · 2023-08-25
> **Thank you so much for your feedback! Summary of changes and clarifications (1/N)**
>
> Dear Reviewer iCgr,
> Thank you so much for your feedback. We have made several changes based on your suggestions:
>
> ### Experiments, Datasets and Models
> 1. **Downstream classifiers**: We have expanded the set of downstream classifiers to include state-of-the-art transformer-based models: TabTransformer [1] for tabular data, PatchTST [2] for time-series data, and FastViT-T8 [3] for image data. We are currently running experiments using these models. While tree-based models can be easily integrated into AQuA, they are incompatible with a few label error detection methods e.g. AUM.
> 2. **Low noise rate setting**: We agree with your assessment regarding 2% noise rate and are in the process of integrating this into our results. We have almost completed running experiments on all models and datasets for 2% noise rate.
> 3. **Data augmentation**: We preprocess all datasets based on prior work, but we are not using any data augmentation during training. We have explicitly stated this in the paper.
> 4. **Replicability and reproducibility**: We have updated the paper to make the following clearer.
> (A) **Publicly available data cards**: We have uploaded all the data cards [here](https://drive.google.com/drive/folders/1RHczHDUUilTOhcPyF5JSDvkO-rhiUKgb). A data card is a CSV file for a given dataset, random seed, noise rate, and noise type, where rows and columns correspond to data points and predictions of cleaning methods, respectively. Each data card also has two additional columns for corrupted (i.e. the static copy) and original labels of data points. All the cleaning methods are evaluated on the same labeling errors.
> (B) **Random seeds**: We try to control all randomness in our experiments stemming from PyTorch, random, numpy, and CUDA ([PyTorch reproducibility guide](https://pytorch.org/docs/stable/notes/randomness.html)). All our experiments are run with the random seed 42. For tabular data, we run two independent experiments with random seeds 42 and 43.
> 5. **Hyper-parameter tuning**: All hyper-parameters were tuned on the training set. We have now also included chosen hyper-parameters in addition to hyper-parameter grids to ensure replicability. Also, see hyper-parameter tuning under the Correctness section.
> 6. **Class-dependent noise**: In our experiments, we estimate the confusion matrix used to add class-dependent label noise to the training set. We have now clarified this in the paper.
> 7. **F1 weighted by support**: During evaluation, we weigh classes by their support, since macro weighting weighs all classes equally and does not account for class imbalance (see [scikit-learn](https://scikit-learn.org/stable/modules/generated/sklearn.metrics.f1_score.html)). We have rephrased the paper to make this clearer.
>
> **References**
>
> [1] Huang, Xin, et al. "Tabtransformer: Tabular data modeling using contextual embeddings." arXiv preprint arXiv:2012.06678 (2020).
>
> [2] Nie, Yuqi, et al. "A Time Series is Worth 64 Words: Long-term Forecasting with Transformers." The Eleventh International Conference on Learning Representations. 2022.
>
> [3] Vasu, Pavan Kumar Anasosalu, et al. "FastViT: A Fast Hybrid Vision Transformer using Structural Reparameterization." arXiv preprint arXiv:2303.14189 (2023).

---

> > ### Author Response · Authors · 2023-08-25
> > **Summary of changes and clarifications (2/N)**
> >
> > ### Figures and statistical tests
> >
> > Thanks for your suggestion. We have now included the following description of the critical difference diagram in the paper:
> > To compare cleaning methods and downstream classifiers across multiple datasets, we follow the recommendations of [4]. First, we use the Friedman test [5] to evaluate whether a statistically significant difference exists between classifiers’ performance. Then, for classifiers with significantly different performance, we conduct pairwise post-hoc analysis recommended by [6] where the average rank comparison is replaced with the Wilcoxon signed-rank test [7] with Holm’s alpha correction [8]. The thick horizontal line in a critical difference diagram shows models that are not significantly different in performance.
> >
> > ### Limitations
> > **Multi-annotator noise**: Based on our future plans, we have implemented three multi-annotator label noise injection methods. Since experiments under these noise settings will involve significant work, we leave them as future work.
> >
> > ### Tables and Typographical corrections
> > Thank you so much for pointing these out. We have now corrected all typographical errors and made all suggested changes to all tables.
> >
> > ### Documentation
> > Thank you for your suggestions. Our goal is to improve the documentation to include step-by-step guides to implement support for new (1) datasets, (2) cleaning methods, (3) downstream classifiers, (4) noise injection methods, and (5) evaluation metrics. We are making a Jupyter notebook that illustrates the entire AQuA process end-to-end, i.e., loading a dataset, identifying labeling errors using a cleaning method, training a classifier, and evaluating it. Besides, we plan to make most datasets available for download using a single API call. We are, however, excluding datasets that we cannot freely distribute e.g. Clothing1M/100K from this exercise. We also plan to make AQuA pip installable in the near future and provide a graphical user interface for less technical users.
> >
> > ### Ethics
> > The ElectricDevices dataset is a part of the UCR-UEA time-series classification archive, which makes all its datasets available under the [GNU General Public License v3.0](https://github.com/time-series-machine-learning/tsml-repo/blob/master/LICENSE). While the RightWhaleCalls dataset is copyrighted by Cornell University and the Cornell Research Foundation, Inc., we have permission to copy, modify, and distribute the data for non-profit research, and non-profit commercial purposes (see [here](https://www.kaggle.com/competitions/whale-detection-challenge/rules#data-license:~:text=Data%20Fusion-,Data,-License)).
> >
> > ### Additional Feedback
> > Thanks for your suggestions on the name of the benchmark! We are considering two alternative names: (1) AQuA, short for **A**nnotation **Qu**ality **A**ssessment, and (2) IdLE, short for **Id**entifying **L**abel **E**rrors
> >
> > **References**
> >
> > [4] Demšar, Janez. "Statistical comparisons of classifiers over multiple data sets." The Journal of Machine Learning Research 7 (2006): 1-30.
> >
> > [5] Friedman, Milton. "A comparison of alternative tests of significance for the problem of m rankings." The annals of mathematical statistics 11.1 (1940): 86-92.
> >
> > [6] Benavoli, Alessio, Giorgio Corani, and Francesca Mangili. "Should we really use post-hoc tests based on mean-ranks?." The Journal of Machine Learning Research 17.1 (2016): 152-161.
> >
> > [7] Wilcoxon, Frank. "Individual comparisons by ranking methods." Breakthroughs in Statistics: Methodology and Distribution. New York, NY: Springer New York, 1992. 196-202.
> >
> > [8] Holm, Sture. "A simple sequentially rejective multiple test procedure." Scandinavian journal of statistics (1979): 65-70.

---

> > > ### Author Response · Authors · 2023-08-25
> > > **Summary of changes and clarifications (3/N)**
> > >
> > > ### Correctness
> > >
> > > **Hyper-parameter tuning:**
> > > 1. For SimiFeat and CINCER, we picked hyper-parameter grids from the papers which first proposed these methods. For AUM, we picked the hyper-parameter grid ourselves since the authors did not provide any recommendations. Finally, confident learning did not have any hyper-parameters (see [Cleanlab](https://github.com/cleanlab/cleanlab)).
> > > 2. For downstream classifiers, we borrowed hyper-parameters from their original implementations. We did not tune hyper-parameters for downstream classifiers so that differences in their performance could be directly attributed to the cleaning methods, rather than differences in their own hyper-parameters.
> > > 3. For each cleaning method and downstream classification model, for a given dataset,  hyper-parameters were chosen based on model performance on the observed training set. Once chosen, hyper-parameters were frozen for all noise experiments (noise type + noise rate). Thus, we believe that our results would be reflective of performance in the wild. However, we acknowledge the following limitations with the current setup:
> > > 4. *Limitation 1*: We agree with the reviewer that tuning hyper-parameters based on the observed training set presents an advantage to the baseline method. In the ideal world, we should conduct extensive hyper-parameter tuning in each experiment setting, i.e. for each combination of dataset, noise rate, noise type, and cleaning method. However, that would be prohibitively expensive. Besides, we believe that insensitivity to hyper-parameters would be a hallmark of a good cleaning method.
> > > 5. *Limitation 2*: Tuning hyper-parameters based on a held-out validation set would be ideal.
> > >
> > > **Number of epochs and batch size**: We agree that tuning the number of epochs and batch size is not ideal, and instead fixing the amount of computational resources spent in training is a better experimental setting. However, we believe that a fixed number of epochs is a good proxy for the “constant training” setting that you are suggesting, under the assumption that the number of predicted mislabeled data points is not significantly different for different cleaning methods. With that said, we are running a small number of experiments in this fixed training setting to test this.
> > >
> > > **Learning rate scheduler**: We are using Reduce LR on Plateau. We will consider the OneCycle learning rate schedule when running future experiments.

---

> > > > ### Comment · Reviewer_iCgr · 2023-08-28
> > > > **Awaiting updated manuscript to review changes**
> > > >
> > > > Thanks to the authors for their comprehensive response! Their statements indicate they have addressed the vast majority of my concerns. However, the copy of the paper that's available to download for me is still the original version as the paper, without any of the updates the authors indicated.
> > > >
> > > > If the authors could upload an updated copy of the main paper and supplementary materials, that would be appreciated so I can confirm everything has been addressed as stated, and see if there is any further follow-up needed. If the authors could upload the revised manuscript with added/adjusted text in a different colour, and/or indicate in OpenReview which line numbers I should look at, that would be appreciated as otherwise I am unlikely to find their changes.

---

> > > > > ### Author Response · Authors · 2023-08-28
> > > > > **Unfortunately, revisions are not allowed before camera ready submission**
> > > > >
> > > > > Dear Reviewer iCgr,
> > > > >
> > > > > We really appreciate your time, effort and responsiveness. We were making changes and highlighting them in the hope that we would be able to submit a revision. However, we realized that we cannot submit a revision prior to the camera ready version of the paper.
> > > > >
> > > > > Thanks for your understanding,
> > > > >
> > > > > Authors
> > > > >
> > > > > ---
> > > > >
> > > > > According to the [NeurIPS 2023 FAQs](https://nips.cc/Conferences/2023/PaperInformation/NeurIPS-FAQ):
> > > > > > **Can we upload a revision of our paper during the rebuttal/discussion period?** No revisions are allowed until the camera-ready stage.
> > > > >
> > > > > > **Can we upload a revision of the supplementary materials during the rebuttal/discussion period?** No. You may revise it for the camera-ready stage.
> > > > >
> > > > > > **Can the rebuttal include new results?** Yes, however your original submission will serve as the basis for the reviewers' (and ACs') acceptance recommendations. The rebuttals should serve only to clarify the reviewers' and ACs' questions during the discussion period.
> > > > >
> > > > > > **Can we include an anonymous link in the author rebuttal?** No. Do not use links in any part of the response. The only exception is if the reviewers asked for code, in which case you can send an anonymized link to the AC in an Official Comment (make sure all linked files are anonymized).

---

> > > > > > ### Comment · Reviewer_iCgr · 2023-08-28
> > > > > > **Unfortunately, revisions are not allowed before camera ready submission?**
> > > > > >
> > > > > > Huh. Weird that they have changed the format from last year. I take it you have submitted an Author Rebuttal PDF, which I can only access after the rebuttal/discussion period has ended?
> > > > > >
> > > > > > Although it contradicts the submission guidelines, I note that it is actually possible in practice to submit a revised copy of the paper during the rebuttal/discussion period and it is immediately available to reviewers... Another paper I am reviewing has done this, and a paper I am author on has done this, and I do not anticipate anyone complaining about this. Given that the interface permits it, it is easy not to realize it is not what you are supposed to do. I think most people have not read the guidelines in detail and so are not aware the rebuttal format has changed this year.
> > > > > >
> > > > > > *From rebuttal*
> > > > > > > Experiments, Datasets and Models
> > > > > > >
> > > > > > > 5. **Hyper-parameter tuning:** All hyper-parameters were tuned on the training set.
> > > > > >
> > > > > > Can you explain what you mean by this? You tuned the hparams to optimize the training accuracy, and not generalization performance?
> > > > > >
> > > > > > *From rebuttal*
> > > > > > > **Number of epochs and batch size:** We agree that tuning the number of epochs and batch size is not ideal, and instead fixing the amount of computational resources spent in training is a better experimental setting. However, we believe that a fixed number of epochs is a good proxy for the “constant training” setting that you are suggesting, under the assumption that the number of predicted mislabeled data points is not significantly different for different cleaning methods. With that said, we are running a small number of experiments in this fixed training setting to test this.
> > > > > >
> > > > > > Based on the results you report, I got the impression that this assumption is not correct, as some models collapsed down to chance level performance when they were trained on the "cleaned" data that came out of some cleaners on some synthetic noise methods. It occurred to me that this could be caused by some models being too overzealous in their sample removal, resulting in models being trained for very few steps. But it would be easy to check, and I daresay may be worthwhile to report in the appendix, the fraction of samples removed by each cleaner to confirm the numbers are comparable.
> > > > > >
> > > > > > The other thing that could be going "situationally wrong" is that some cleaners may be removing more samples from one class than another. It may be worthwhile to analyze the class imbalance induced by different cleaners, and if it is significant in some cases then consider training models with a rebalanced sampler or class weighting to address the imbalance they induce so the model's prior matches that of the test distribution.
> > > > > >
> > > > > > *From review*
> > > > > > > the IMDb and TweetEval models sometimes do better when noise is added to the data, even when no noise cleaning method has been applied, especially when adding class-dependent noise (see Table 2). This noise profile is essentially adding label smoothing to the training objective and reducing overfitting. But somehow all kinds of noise improve the performance of all-distilroberta-v1 on IMDb (Table 7).
> > > > > >
> > > > > > Do the authors have any thoughts they can share on why adding noise is increasing performance for these datasets/models? Do they agree with me that this indicates the experimental paradigm needs to be refined for their text models, and if so do they have thoughts on how to refine it?

---

> > > > > > > ### Author Response · Authors · 2023-08-29
> > > > > > > **Thanks for your suggestions! Preliminary results of experiments you suggested and some insights (1/N)**
> > > > > > >
> > > > > > > Dear Reviewer iCgr,
> > > > > > > Thank you for your continued engagement to make our manuscript better. We are very appreciative of your time and suggestions.
> > > > > > >
> > > > > > > ### Hyper-parameter Tuning
> > > > > > > > 5. Hyper-parameter tuning: All hyper-parameters were tuned on the training set.
> > > > > > >
> > > > > > > Yes, we tuned hyper-parameters to optimize the weighted $F_1$ score on the training set. There were primarily 2 reasons behind this design decision:
> > > > > > > (1) Our goal was to identify hyper-parameters that led to *reasonable performance* on the training set. Fine-grained tuning of hyper-parameters based on any dataset, whether held-out or in-domain, is tricky because the impact of label errors on model evaluation is hard to predict. We believe that evaluating model performance in the presence of label noise is a hard but important research direction that deserves a dedicated study.
> > > > > > > (2) Besides, we wonder how important it is to pick "the best" model that performs well on a held-out dataset, when in fact most if not all of the considered label cleaning methods utilize these downstream models (primarily trained on the training set) to learn representations of training data points. Once erroneous labels are identified, they are removed and the same model is re-trained on the "cleaned" training data, and their performance is measured on the test data.
> > > > > > > With that said, we do believe that in the ideal scenario, we should tune hyper-parameters on held-out data with no label errors prior to and after label cleaning. But I hope you agree that this ideal scenario is contingent on a guaranteed error-free validation set and at least twice as much compute, which are prohibitive assumptions.

---

> > > > > > > > ### Author Response · Authors · 2023-08-29
> > > > > > > > **Thanks for your suggestions! Preliminary results of experiments you suggested and some insights (2/N)**
> > > > > > > >
> > > > > > > > ### Distribution of Data Points Prior to and Post Label Cleaning
> > > > > > > > Thank you so much for recommending two interesting experiments, which helped us delve deeper into the performance of the cleaning methods. We conducted 2 experiments for which we will add results detailed results in the appendix.
> > > > > > > >
> > > > > > > > #### What is the average proportion of data points cleaned by each cleaning method?
> > > > > > > > We report results averaged over noise type, noise rate, random seed(s), and downstream model architecture and datasets. Some key takeaways: (1) Confident Learning removes 57% of training data points on average explaining its poor performance. Note that confident learning had no hyper-parameters to be tuned, so the poor performance cannot be attributed to them. (2) All other methods remove <13% data points.
> > > > > > > >
> > > > > > > > |     Datasets     |  AUM  | CINCER | Confident Learning | SimiFeat |
> > > > > > > > |:----------------:|:-----:|:------:|:--------:|:--------:|
> > > > > > > > |    **Average**   | 0.011 |  0.133 |   0.574  |   0.061  |
> > > > > > > > |    **Std Dev**   | 0.001 |  0.115 |   0.248  |   0.038  |
> > > > > > > > |    **Median**    | 0.011 |  0.095 |   0.478  |   0.057  |
> > > > > > > >
> > > > > > > > #### Do the methods create a significant class imbalance?
> > > > > > > > We measured imbalance using the proportion of data points belonging to the minority class (i.e. the class which has the minimum # of data points). We report the difference between the proportion of data points prior to and after label cleaning, averaged over noise type, noise rate, random seed(s), and downstream model architecture and datasets. The major takeaway is that barring confident learning, none of the other methods significantly change the proportion of data points in the minority class.
> > > > > > > >
> > > > > > > > |     Datasets     |  AUM  | CINCER | Confident Learning | SimiFeat |
> > > > > > > > |:----------------:|:-----:|:------:|:--------:|:--------:|
> > > > > > > > | **Average**   | 0.002 |  0.011 |   0.047  |   0.003  |
> > > > > > > > | **Std Dev**   | 0.003 |  0.011 |   0.036  |   0.004  |
> > > > > > > > | **Median**    | 0.000 |  0.008 |   0.053  |   0.001  |
> > > > > > > >
> > > > > > > > In summary, it is probably correct that for confident learning the downstream models are not seeing enough data or trained long enough for models to converge. But we believe that this might be a problem of the cleaning method, more than the experiment design.
> > > > > > > >
> > > > > > > > ### Performance Comparison Before and After Label Cleaning
> > > > > > > > > Do the authors have any thoughts they can share on why adding noise is increasing performance for these datasets/models? Do they agree with me that this indicates the experimental paradigm needs to be refined for their text models, and if so do they have thoughts on how to refine it?
> > > > > > > >
> > > > > > > > We agree with you that in the context of class-dependent or uniform noise, label noise serves as regularization to prevent models from overfitting. But we have some additional thoughts:
> > > > > > > >
> > > > > > > > 1. This is a phenomenon that is *not specific to text*, but happens for other modalities, datasets, and noise types too. E.g. in Table 7, see Electric Devices (time-series) under uniform noise, MIT-BIH (time-series), and Dry Bean (tabular) for class-dependent noise. The same is not true in Table 8.
> > > > > > > > 2. The text datasets are binary class classification problems that are known to have pre-existing labeling errors. Therefore, sometimes flipping the label might actually "correct" it.
> > > > > > > > 3. We would also like to draw your attention to IMDb in Table 7 and 8 as an example. When we clean data without adding additional noise, the results are sometimes better than when we add synthetic noise (e.g. class dependent), which supports point (2).
> > > > > > > > 4. We **do not make conclusions based on isolated datasets or models** for this very reason -- instead, we use statistical tests conducted across multiple different experimental settings to draw conclusions.
> > > > > > > > 5. Finally, deep learning optimization is highly non-convex, so adding some noise might help the model reach the global minima, through a different route in the loss space.
> > > > > > > >
> > > > > > > > ### Name for Benchmark
> > > > > > > > We are very very thankful to you for your thoughtful suggestions! We agree with you and will certainly consider your suggestions.

---

> > > > > > > > > ### Comment · Reviewer_iCgr · 2023-08-31
> > > > > > > > >
> > > > > > > > > Thanks to the authors for their response. I'm glad to hear the number of samples removed and class imbalance induced by the cleaners was straight-forward to compute and offered some new insights.
> > > > > > > > >
> > > > > > > > > I am happy to increase my score for the paper, on the assumption that the changes/additions the authors already mentioned have been implemented. The authors may have already added these final items we discussed as well, but in case they haven't already then I'd like to request they add to the paper:
> > > > > > > > >
> > > > > > > > > 1. If there is a significant difference from using constant number of steps for training instead of constant number of epochs (either improving the Confident Learning performance where lots of samples were removed, or decreasing the performance of other cleaned models by reducing early-stopping), this is added to and described in the paper (and at the author's discretion potentially replaces the by-epoch results table, relegating the by-epoch results to the appendix, depending on the significance and implications of the difference in results)
> > > > > > > > >
> > > > > > > > > 2. The amount of samples removed by the various cleaning methods is described and discussed.
> > > > > > > > >
> > > > > > > > > 3. The phenomenon of noise sometimes improving results is mentioned in the discussion.
> > > > > > > > >
> > > > > > > > > 4. The fact that the hparam search was done against the training performance is mentioned in the paper in an appropriate place.
> > > > > > > > >
> > > > > > > > > 5. That models were trained with Reduce LR on Plateau (which was presumably monitored on the train set too?) is similarly mentioned.
> > > > > > > > >
> > > > > > > > > 6. The main paper mentions (in methods or discussion?) that the models being trained are potentially in an overfitting paradigm, since there's very limited regularization (no data augmentation, maybe some w.d. and/or dropout) and the hparams were selected for on the training performance. [I note that the hparam search included ranges of dropout and w.d. values for some models, but if you are optimizing the training perf, I don't see why the hparam search wouldn't choose the smallest options for each of these.] The authors might gain further insights into the impact of the cleaners on downstream model performance if they check the train perf (and compare it with test) to see how much overfitting is taking place.

---

> > > ### Comment · Reviewer_iCgr · 2023-08-29
> > > **Naming suggestions**
> > >
> > > > Thanks for your suggestions on the name of the benchmark! We are considering two alternative names: (1) AQuA, short for **A**nnotation **Qu**ality **A**ssessment, and (2) IdLE, short for **Id**entifying **L**abel **E**rrors
> > >
> > > Identifying labels errors is the function of the detectors (the label cleaners) that are being assessed by the benchmark, so that doesn't seem appropriate to me. I think you ideally need an extra level around it. Annotation Quality Assessment is better (the best option of the three you've mentioned IMHO), but still missing the extra level. The tool is for assessing things which assess labels (meta-assessment) - what you're doing is actually Annotation Quality Assessment Assessment! (And that might be a viable name too? initialized as AQuAA or $AQuA^2$.)
> > >
> > > I think the letters you have to work with give you a large scope for potential names, whether they are just pronouncable acronyms or existing words. Here are a few examples:
> > >
> > > - **A**nnotation **Qu**ality **A**ssessor **A**ssessment (AQuAA)
> > > - **Qu**antifying **A**utomated **D**etection of **Mi**s**L**abelled **D**ata (QuADMiLD)
> > > - **As**sessing **K**leaning of **Mi**slabelled **Da**ta**S**ets (AsKMiDaS)
> > > - **A**ssessing **De**tection of **L**abel **E**rrors (ADeLE)
> > > - **L**abel **E**rror **D**etector **G**rading and **E**valuation **S**uite (LEDGES)
> > > - **G**rading **L**abel **A**nnotation **N**oise **D**etectors (GLAND)
> > > - **G**rading **I**dentifiers for **L**abelling **E**rrors (GILE)
> > > - **D**ata **A**nnotation **N**oise **C**leaning **E**valuation **S**uite (DANCES)
> > > - **E**rroneous **L**abel **D**etection **E**valuation with **R**areification (ELDER)
> > > - **B**enchmarking **L**abel **E**rror **D**etectors (BLED)
> > > - **B**enchmarking **A**nnotation **N**oise **D**etectors (BAND)
> > > - **B**enchmarking **L**abel **A**nnotation **N**oise **D**etectors (BLAND)
> > > - **B**enchmarking **L**abel **A**nnotation **N**oise **C**leaning (BLANC)
> > > - **B**enchmarking **L**abel **A**nnotation **N**oise **C**orrection **H**euristics (BLANCH)
> > > - **B**enchmarking **L**abel **E**rror Detection and **A**nnotation **C**orrection **H**euristics (BLEACH)
> > > - **B**enchmarking **E**rroneous **L**abel **C**orrection **H**euristics (BELCH)
> > > - **Be**nchmark for **D**etectors of **A**nnotation **E**rrors (BeDAE)

---

### Decision · Program_Chairs · 2023-09-22

**Decision:**

Accept (Poster)

**Comment:**

This paper's scores are 7-7-6-6, where reviewers are generally positive. The paper introduces the Automated Label Quality Assessment (AQuA) benchmark, a tool designed for evaluating label quality in machine learning datasets. Reviewers acknowledge that the benchmark addresses the issue of label errors and their impact on model generalization and offers a standard framework for assessing label error detection methods. Reviewers have generally expressed positive opinions about the paper's contributions, practical relevance, and the clear motivation behind the research. During rebuttal, authors did a good job in addressing reviewer questions such as hyperparameter tuning, the use of stronger baseline models, etc. Considering all these aspects, AC recommends to accept this paper.